# Oxygen evolution reaction on IrO$_2$(110) is governed by Walden-type mechanisms

Muhammad Usama[1,2], Samad Razzaq[1], Christof Hättig[2,3],
Stephan N. Steinmann [4] & Kai S. Exner [1,2,5]

Oxygen evolution reaction (OER) is a key process for sustainable energy, although renewable sources require the use of proton exchange membrane electrolyzers, with IrO$_2$-based materials being the gold standard under anodic polarization conditions. However, even for the (110) facet of a single-crystalline IrO$_2$ model electrode, the reaction mechanism is not settled yet due to contradictory reports in literature. In the present manuscript, we disentangle the conflicting results of previous theoretical studies in the density functional theory approximation. We demonstrate that dissimilar reaction mechanisms and limiting steps for the OER over IrO$_2$(110) are obtained for different active surface configurations present on the IrO$_2$ electrode. In contrast to previous studies, we factor Walden-type mechanisms, in which the formation of the product O$_2$ and adsorption of the reactant H$_2$O occur simultaneously, into the analysis of the elementary steps. Combining free-energy diagrams along the reaction coordinate and Bader charge analysis of the active site, we elucidate why mononuclear- or bifunctional-Walden pathways excel the traditional OER mechanisms for the OER over IrO$_2$(110). Our computational methodology to identify the reaction mechanism and limiting step of proton-coupled electron transfer steps is widely applicable to electrochemical processes in the field of energy conversion and storage.

Oxygen evolution reaction (OER) – 2 H$_2$O → O$_2$ + 4 H$^+$ + 4 e$^-$, $U^0_{OER}$ = 1.23 V $vs.$ reversible hydrogen electrode (RHE) – is encountered with the bottleneck in proton exchange membrane (PEM) electrolyzers to produce the energy vector gaseous hydrogen[1–5]. Despite tremendous efforts to develop cost-effective alternatives, hitherto, only IrO$_2$-based catalysts can withstand the harsh acidic reaction conditions in PEM electrolyzers[6–9]. While the OER over IrO$_2$ has been extensively studied by both experimental and theoretical approaches, so far, there is no consensus on the elementary reaction steps that govern the rate of this reaction[10–15]. In the present work, we reinvestigate the OER on the (110) facet of IrO$_2$ as this is a suitable model system to better understand the factors limiting the OER at the atomic level.

In the early work of Rossmeisl and coworkers, canonical DFT calculations in the realm of the computational hydrogen electrode approach (CHE) were applied to model the OER over a single-crystalline IrO$_2$(110) model electrode by means of the mononuclear mechanism, containing the subsequent formation of the *OH, *O, and *OOH adsorbate[16,17] (cf. supporting information (SI), section 3.1). The authors demonstrated that the formation of the *OOH adsorbate on IrO$_2$(110) is limiting the rate in the thermodynamic picture of the potential-determining step (PDS); note that the PDS is not necessarily related with the rate-determining step (RDS) governing the kinetics of the reaction[18,19]. Later, Ping and Goddard investigated the kinetics of the OER over IrO$_2$(110) by grand-canonical DFT calculations, concluding that the formation of the *OOH adsorbate by a chemical step via a

[1]University Duisburg-Essen, Faculty of Chemistry, Theoretical Catalysis and Electrochemistry, Universitätsstraße 5, Essen, Germany. [2]Cluster of Excellence RESOLV, Bochum, Germany. [3]Lehrstuhl für Theoretische Chemie, Ruhr-Universität Bochum, D- Bochum, Germany. [4]CNRS, ENS de Lyon, LCH, UMR 5182, Lyon cedex 07, France. [5]Center for Nanointegration (CENIDE) Duisburg-Essen, Duisburg, Germany. ✉e-mail: kai.exner@uni-due.de

bifunctional mechanism (cf. SI, section 3.3) corresponds to the RDS[20], which is in line with the study of Jones and coworkers in a recent communication[21]. On the other hand, Exner and Over concluded, based on the combination of DFT calculations and experimental Tafel slope analyses, that the decomposition of the *OOH adsorbate rather than its formation is rate determining for typical OER conditions[22]. Ha and Larsen pointed out that, besides the mononuclear mechanism favored by Rossmeisl and coworkers, also a binuclear description (cf. SI, section 3.5) can be operative for the (110) facet over $IrO_2$[23]. Finally, Binninger and Doublet suggested another OER pathway, which consists of the chemical recombination of the two outermost oxygen atoms in two adjacent *OO intermediates[24] (cf. oxide pathway in the SI, section 3.4). The authors demonstrated that the oxide pathway is energetically preferred over the mononuclear description, indicating that all the previous studies may have missed an important pathway in the analysis of free-energy diagrams along the reaction coordinate[25,26].

While the above survey summarizes selected studies from the vibrant field of the OER over $IrO_2$[27,28], highlighting that even experimental groups have made efforts to identify limiting reaction steps by a dedicated modeling procedure of cyclic voltammetrograms[29–32], we trace the observed discrepancy of the discussed DFT studies to the following reasons:

a)  the starting structure for the modeling of the elementary steps largely varies between the different works[16,17,20–24], ranging from hydroxylated to oxygen-covered surfaces. Most of these works consider only one or two different surface configurations, though real-world catalysts, as observed under operando conditions, may show a plethora of different surface patterns on the electrode surface. To underpin this point, we have compiled the data from previous DFT studies in Table S33 (cf. SI, section 13): the thermodynamically stable surface structure of $IrO_2$(110) depends on the exchange correlation function and the solvation description used in the analysis.

b)  in the different works, only selected pathways have been studied by means of DFT so that the conclusions made are sensitive if the set of considered reaction mechanisms is extended. This finding particularly refers to DFT-based kinetic studies where only selected transition states were calculated, which may result in an incorrect determination of the RDS if energetically unfavorable transition states were excluded from the analysis.

In the present article, we shed new light on the elementary steps of the OER over $IrO_2$(110) by connecting DFT calculations with descriptor-based analyses using the activity measure $G_{max}(U)$[33,34]. Despite its high stability under anodic polarization conditions, $IrO_2$(110) reveals potential-induced pitting corrosion, although there is no uniform thinning of the $IrO_2$(110) layer on the atomic scale[35]. Therefore, it is justified to model the elementary steps on the (110) facet, which can be seen as a suitable model system for $IrO_2$-based materials in the OER[36–38]. We consider a variety of different thermodynamically stable surface configurations as the starting point for the description of the elementary reaction steps, and we extend the mechanistic evaluation beyond the traditional reaction mechanisms in the OER (cf. SI, section 3). To this end, we factor Walden-type pathways[39] (cf. SI, section 8) with simultaneous bond-breaking and bond-making events in our dedicated analysis of the elementary reaction steps.

Surprisingly, we observe that, independent of the $IrO_2$ surface configuration, the Walden pathways reveal higher electrocatalytic activity than the traditional mechanisms, which we trace to a favorable charge distribution of the active Ir site during the catalytic cycle. Our work highlights the central role of the Walden inversion beyond molecular chemistry in that concurrent desorption and adsorption steps, albeit so far frequently overlooked in the field of heterogeneous electrocatalysis[40], may govern the complex proton-coupled electron transfer processes at electrified solid/ liquid interfaces.

## Results
### Traditional reaction mechanisms

We model the OER over a $(2 \times 1)$ $IrO_2$(110) surface by DFT calculations; all computational details can be found in section 1 of the SI. The stoichiometric $(2 \times 1)$ $IrO_2$(110) surface, abbreviated $2O_{br} + 2*_{cus}$ (cf. SI, Figure S1), contains two dissimilar Ir surface atoms with a different chemical environment, namely 'cus' and 'bridge' sites[41]. There is a consensus in the literature that the cus sites are the catalytically active centers for the OER or other surface reactions whereas the bridge sites are mainly spectators. Note that the bridge sites can still be involved in the OER, as oxygen atoms at the Ir bridge site can serve as a Brønsted base by accepting a proton during the elementary steps of the OER (cf. SI, section 3.2 and 3.3)[42,43]. To clearly rule out the Ir bridge sites as active sites in the OER, we have compared their activity to that of the Ir cus sites in section 14 of the SI.

To gain insight into the surface configuration of $IrO_2$ during the OER, we apply the concept of surface Pourbaix diagrams to identify thermodynamically stable structure under anodic conditions[44–47]. A detailed discussion of the Pourbaix approach is provided in section 2 of the SI. Figure 1a indicates the DFT-based Pourbaix diagram of a single-crystalline $IrO_2$(110) electrode as a function of applied electrode potential and pH. Please note that the pH independence of the surface configurations in the Pourbaix diagram is a direct result of the CHE method and therefore differs from experimental Pourbaix diagrams, where boundary lines typically exhibit a pH dependence. We note that the pH dependence of the boundary lines can be resolved using DFT calculations in a grand canonical framework[48,49], which we have used herein to benchmark the application of the CHE method for the modeling of mechanistic pathways (cf. SI, section 6.7).

Figure 1a reveals that depending on the applied electrode potential, different hydroxylated, oxygen-covered, and OOH-covered surfaces are energetically preferred. The observed surface configurations are consistent with previous DFT-based studies on $IrO_2$(110) (cf. SI, section 13). We select four different surface configurations (cf. Figure 1b–e) with comparable energetics under typical OER conditions ($U > 1.23$ V vs. RHE) as the starting point for our mechanistic analyses. Initially, we consider five different reaction mechanisms, namely the mononuclear, bifunctional I, bifunctional II, oxide, and binuclear descriptions (cf. SI, section 3)[50–53]. These pathways are summarized in Fig. 2 by a network of elementary steps using the example of the fully oxygen-covered surface (cf. Figure 1d).

To assess the electrocatalytic activity of the different mechanisms over the various surface configurations, we employ the descriptor $G_{max}(U)$, which relies on the notion of a span model[54,55]: $G_{max}(U)$ indicates the largest free-energy span from the intermediate with the smallest to the intermediate with the highest free energy at a given electrode potential in the free-energy landscape. The peculiarity of this descriptor refers to the fact that it contains a measure for sensitivity[56,57], based on the benchmarking with experimentally obtained transition-state free energies[22,58]: only if the $G_{max}(U)$ values of two mechanisms differ by at least 0.20 eV, we infer that the mechanism with the smaller $G_{max}(U)$ value is unambiguously preferred. This allows the screening of mechanistic pathways of proton-coupled electron transfer steps on the level of thermodynamic considerations, focusing on the free energies of the intermediate species. We note that the application of $G_{max}(U)$ as the activity descriptor in our analysis does not cause a significant loss in accuracy as even state-of-the-art transition state calculations for proton-coupled electron transfer steps under constant potential contain error bars of at least 0.15 eV[33,59–61]. To this end, we conclude that it is beneficial to use the thermodynamic evaluation with a slightly larger uncertainty than the kinetic picture, as only this simplification allows us to conduct a comprehensive study of

the various pathways (cf. Figure 2) and surface configurations (cf. Figure 1b-e) for the OER over $IrO_2(110)$.

In section 4 of the SI, we discuss the OER mechanisms over the fully hydroxylated $IrO_2(110)$ surface – $2OH_{br} + 2^*_{cus}\text{-}OH_{ot}$ – by the construction of free-energy diagrams. Our procedure consists in investigating the five different mechanistic descriptions for the $2OH_{br} + 2^*_{cus}\text{-}OH_{ot}$ surface by extracting the descriptor $G_{max}(U)$ at $U = 1.53$ V vs. RHE. This is achieved by calculating the free-energy landscape under equilibrium conditions ($U = 1.23$ V vs. RHE), and subsequently the energetics is translated to $U = 1.53$ V vs. RHE using the CHE approach. Figure 3 shows the corresponding free-energy diagrams for the mononuclear, bifunctional I, and bifunctional II mechanisms at $U = 1.23$ V vs. RHE and 1.53 V vs. RHE; the two latter pathways are the preferred mechanistic descriptions for this surface termination. Please note that we apply the same methodology to all other surface configurations reported in Fig. 1b-e.

Table 1 summarizes the results for the modeling of the OER over the fully hydroxylated $IrO_2(110)$ surface, indicating that both bifunctional descriptions reveal identical electrocatalytic activity in the approximation of $G_{max}(U)$, whereas the other pathways can be fairly ruled out due to their larger values of the activity measure $G_{max}(U)$. Please note that the oxide mechanism cannot take place for the fully hydroxylated $IrO_2(110)$ surface due to the instability of a reaction intermediate containing the *OOH adsorbate (cf. SI, section 4.4). We note that the observation of the bifunctional description under OER conditions coincides with the previous works of Goddard and coworkers or Jones and coworkers[20,21].

For the partly hydroxylated $IrO_2(110)$ surface – $2O_{br} + 2^*_{cus}\text{-}OH_{ot}$ – we have constructed free-energy diagrams along the reaction coordinate in section 5 of the SI (cf. Figures S8–S12). It turns out that four mechanistic descriptions compete under OER conditions (cf. Table 1). While the oxide mechanism[61] is energetically preferred due to the smallest value of the descriptor $G_{max}(U)$, the mononuclear and bifunctional descriptions cannot be excluded due to their comparable electrocatalytic activity. Only the binuclear mechanism can be clearly ruled out because of the large energy penalty for the formation of gaseous oxygen by a chemical reaction step. It is relevant to point out that the oxide mechanism as the operating pathway is in line with the previous work by Binninger and Doublet[24]. For further information on the OER mechanisms over the partly fully hydroxylated $IrO_2(110)$ surface, we refer the reader to section 5 of the SI.

It is noteworthy that the limiting span for the four identified pathways of the partly hydroxylated $IrO_2(110)$ surface are different to a large extent compared to the case of the fully hydroxylated surface (cf. Table 1). Table 1 indicates that under OER conditions ($U = 1.53$ V vs. RHE), the decomposition of the *OOH or *OO adsorbates rather than their formation governs the rate in the approximation of $G_{max}(U)$. This finding agrees with the suggestion of Exner and Over on the limiting step in the OER over $IrO_2(110)$[22].

A similar situation is encountered for the fully oxygen-covered $IrO_2(110)$ surface – $2O_{br} + 2^*_{cus}\text{-}O_{ot}$ – where the mononuclear and bifunctional descriptions compete under OER conditions. While an extended discussion is provided in section 6 of the SI (cf. Figures S13–S17), Table 2 illustrates that the limiting span for these pathways comprises the decomposition of the *OOH intermediate.

The Pourbaix diagram in Fig. 1 illustrates that at applied electrode potentials exceeding 1.58 V vs. RHE, the *OOH adsorbate becomes thermodynamically stable on the electrode surface. Therefore, we study the different mechanistic pathways over the partly OOH-covered $IrO_2(110)$ surface – $2O_{br} + 1^*_{cus}\text{-}OOH_{ot}$ $1^*_{cus}\text{-}O_{ot}$ – and the energetics is shown in Figures S18–S21 (cf. SI, section 7). We observe that the bifunctional and oxide descriptions are preferred under OER conditions (cf. Table 2). For the OOH-covered surface, also the formation of the *OOH adsorbate can govern the activity descriptor $G_{max}(U)$ in case of the oxide pathway, which is in line with the proposal of Rossmeisl and coworkers on the limiting reaction step[16,17,53]. Please note that the

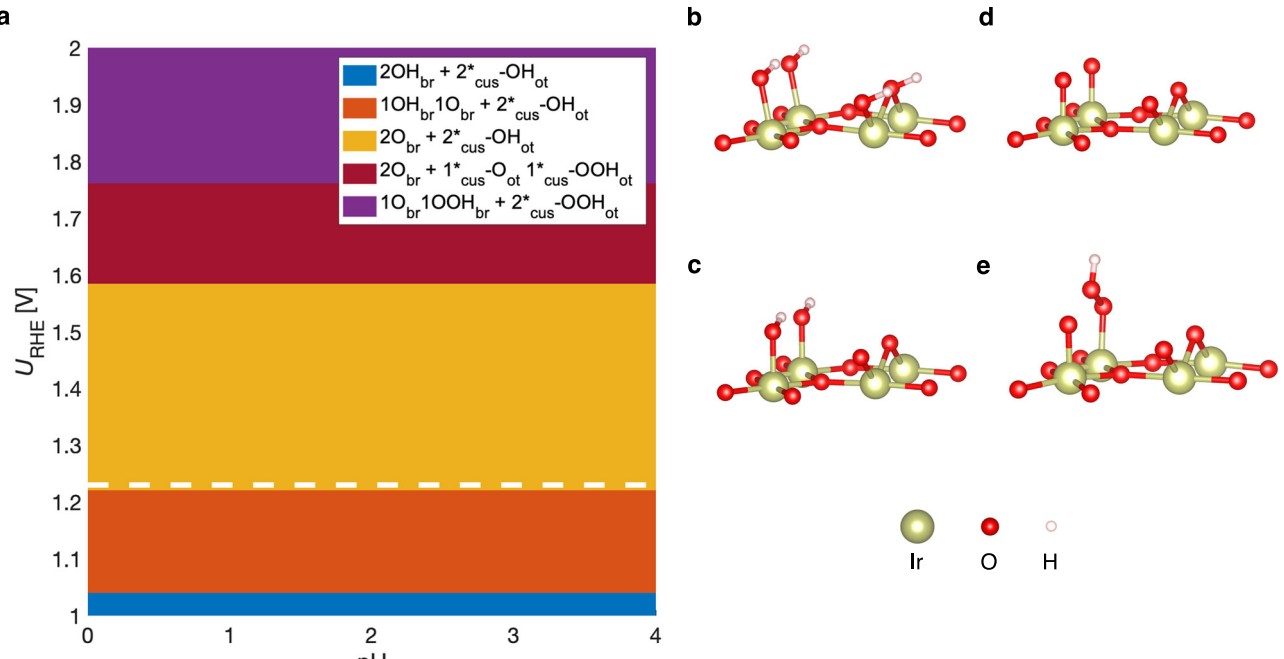

**Fig. 1 | Surface Configurations of $IrO_2(110)$ under Anodic Conditions. a** Pourbaix diagram for $IrO_2(110)$, indicating the thermodynamically most stable surface phase as a function of applied electrode potential, $U$, and pH. The white dotted line represents the OER equilibrium potential, $U^0_{OER} = 1.23$ V vs. reversible hydrogen electrode (RHE). While a partly hydroxylated $IrO_2(110)$ surface is observed at electrode potentials exceeding the OER equilibrium potential, we consider four different surface motifs for our mechanistic studies due to their comparable energetics under OER conditions: (**b**) fully hydroxylated surface ($2OH_{br} + 2^*_{cus}\text{-}OH_{ot}$), **c**) partly hydroxylated surface ($2O_{br} + 2^*_{cus}\text{-}OH_{ot}$), **d**) fully oxygen-covered surface ($2O_{br} + 2^*_{cus}\text{-}O_{ot}$), and **e**) partly OOH-terminated surface ($2O_{br} + 1^*_{cus}\text{-}OOH_{ot}$ $1^*_{cus}\text{-}O_{ot}$).

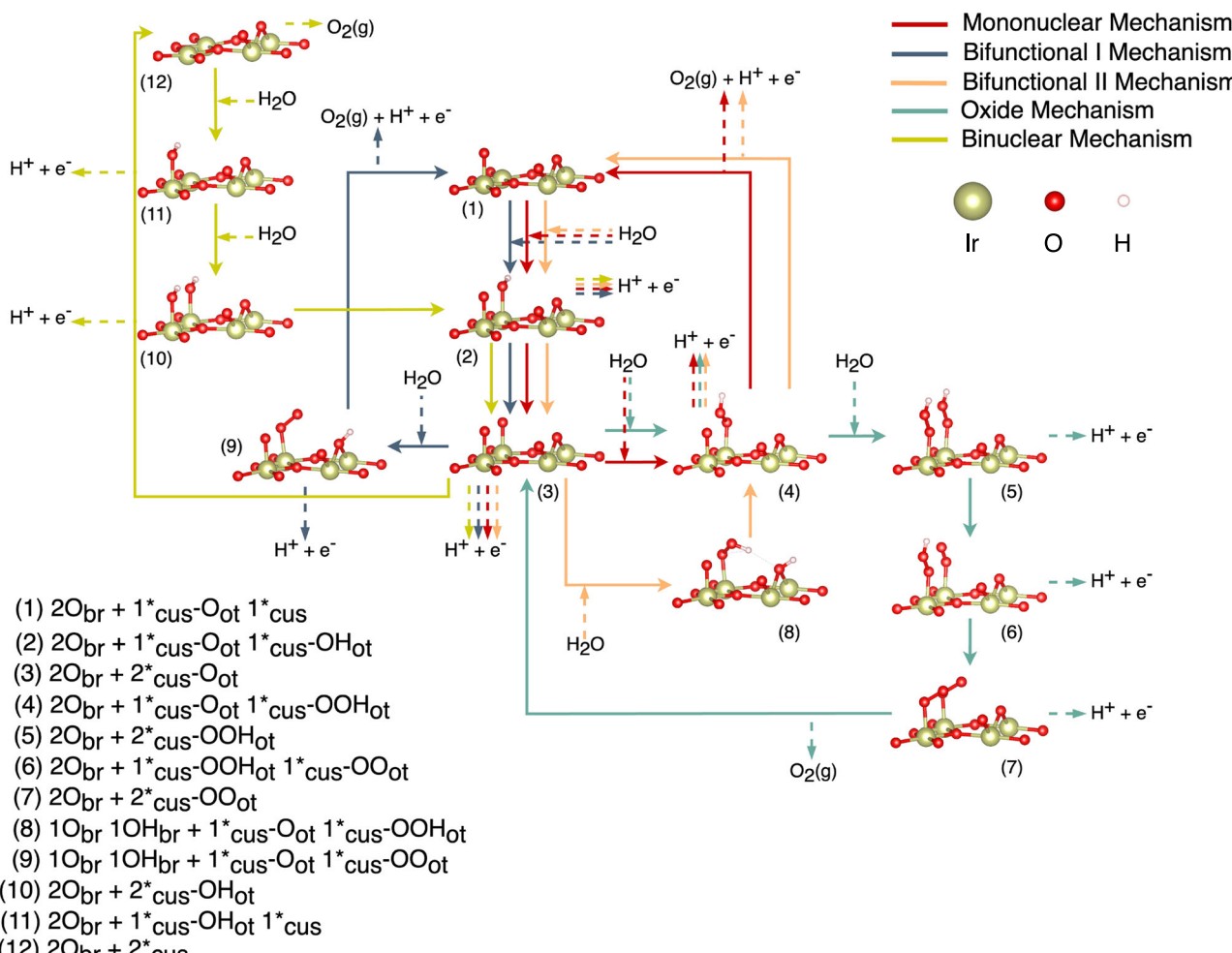

(1) $2O_{br} + 1*_{cus}\text{-}O_{ot}\ 1*_{cus}$
(2) $2O_{br} + 1*_{cus}\text{-}O_{ot}\ 1*_{cus}\text{-}OH_{ot}$
(3) $2O_{br} + 2*_{cus}\text{-}O_{ot}$
(4) $2O_{br} + 1*_{cus}\text{-}O_{ot}\ 1*_{cus}\text{-}OOH_{ot}$
(5) $2O_{br} + 2*_{cus}\text{-}OOH_{ot}$
(6) $2O_{br} + 1*_{cus}\text{-}OOH_{ot}\ 1*_{cus}\text{-}OO_{ot}$
(7) $2O_{br} + 2*_{cus}\text{-}OO_{ot}$
(8) $1O_{br}\ 1OH_{br} + 1*_{cus}\text{-}O_{ot}\ 1*_{cus}\text{-}OOH_{ot}$
(9) $1O_{br}\ 1OH_{br} + 1*_{cus}\text{-}O_{ot}\ 1*_{cus}\text{-}OO_{ot}$
(10) $2O_{br} + 2*_{cus}\text{-}OH_{ot}$
(11) $2O_{br} + 1*_{cus}\text{-}OH_{ot}\ 1*_{cus}$
(12) $2O_{br} + 2*_{cus}$

**Fig. 2 | Overview of the Reaction Mechanisms of Oxygen Evolution.** Investigated OER mechanisms on IrO$_2$(110) taking the fully oxygen-covered surface (cf. Figure 1d) as a representative example. Colors of the various mechanistic pathways are indicated in the top right corner, and surface structure details are given in the bottom left corner. Note that the same steps have also been studied over the other surface motifs depicted in Fig. 1b–e.

binuclear mechanism cannot be operative for the partly OOH-covered IrO$_2$(110) surface as further explained in section 7 of the SI, where free-energy diagrams along the reaction coordinate are provided for all the mechanistic descriptions.

In summary, it can be concluded that the different reports on the limiting reaction step and mechanism in the OER over IrO$_2$(110) in the literature are essentially reproduced when considering that different surface configurations are available on the electrode surface under reaction conditions. When comparing the intrinsic activity of these surface configurations in the approximation of $G_{max}(U)$, we observe that the fully oxygen-covered surface (cf. Table 2) is the most active phase at $U = 1.53$ V vs. RHE; however, the fully hydroxylated (cf. Table 1) and partly OOH-covered (cf. Table 2) surfaces cannot be excluded as $G_{max}(U)$ deviates less than 0.20 eV compared to the fully oxygen-covered configuration. This finding suggests that not a single mechanism or a single reaction step is governing the OER over IrO$_2$(110), but rather a variety of different steps and mechanisms control the rate of this reaction[50]. In the following section though, we demonstrate that none of the above mechanistic descriptions is operative for IrO$_2$(110) under OER conditions due to the necessity of considering Walden-type pathways in the analysis[39,40].

## Walden-type mechanisms

All mechanistic pathways summarized in the network of Fig. 2 rely on the notion that product formation is accompanied with the restoration of the catalytically active Ir cus site, $*_{cus}$, on the IrO$_2$(110) surface (cf. structure (1) in Fig. 2). Yet, removal of the product O$_2$ and adsorption of the reactant H$_2$O can also proceed simultaneously so that correspondingly, the vacant $*_{cus}$ site is no longer observed in the catalytic cycle. Mechanisms that follow these lines are called Walden pathways[39], and the elementary steps for the mononuclear- and bifunctional-Walden OER mechanisms are compiled in section 8 of the SI. A brief comment on the terminology of Walden pathways is needed. We note that the 'traditional Walden inversion', which is particularly observed in homogeneous catalysis[62–64], takes place at an angle of 180°: to minimize steric hindrance, the reactant enters the active site while the product leaves the active site exactly on the opposite side. In heterogeneous catalysis, it is definitely not possible for the reactant to enter the active center and the product to leave the active center at an angle of 180°; rather, the angle between reactant and product is compressed. Although this is a difference between 'Walden steps' in homogeneous and heterogeneous catalysis, the chemical processes in terms of concerted desorption-adsorption still remain the same. Therefore, in this work, we adopt the terminology of "Walden-like" mechanisms because we believe it will help bridge the knowledge gap between homogeneous and heterogeneous catalysis.

Figure 4 shows the free-energy landscape for the mononuclear- and bifunctional-Walden OER mechanisms using the example of the partly hydroxylated IrO$_2$(110) surface – $2O_{br} + 2*_{cus}\text{-}OH_{ot}$ – at $U = 1.23$ V vs. RHE and 1.53 V vs. RHE. For the other IrO$_2$(110) surface terminations

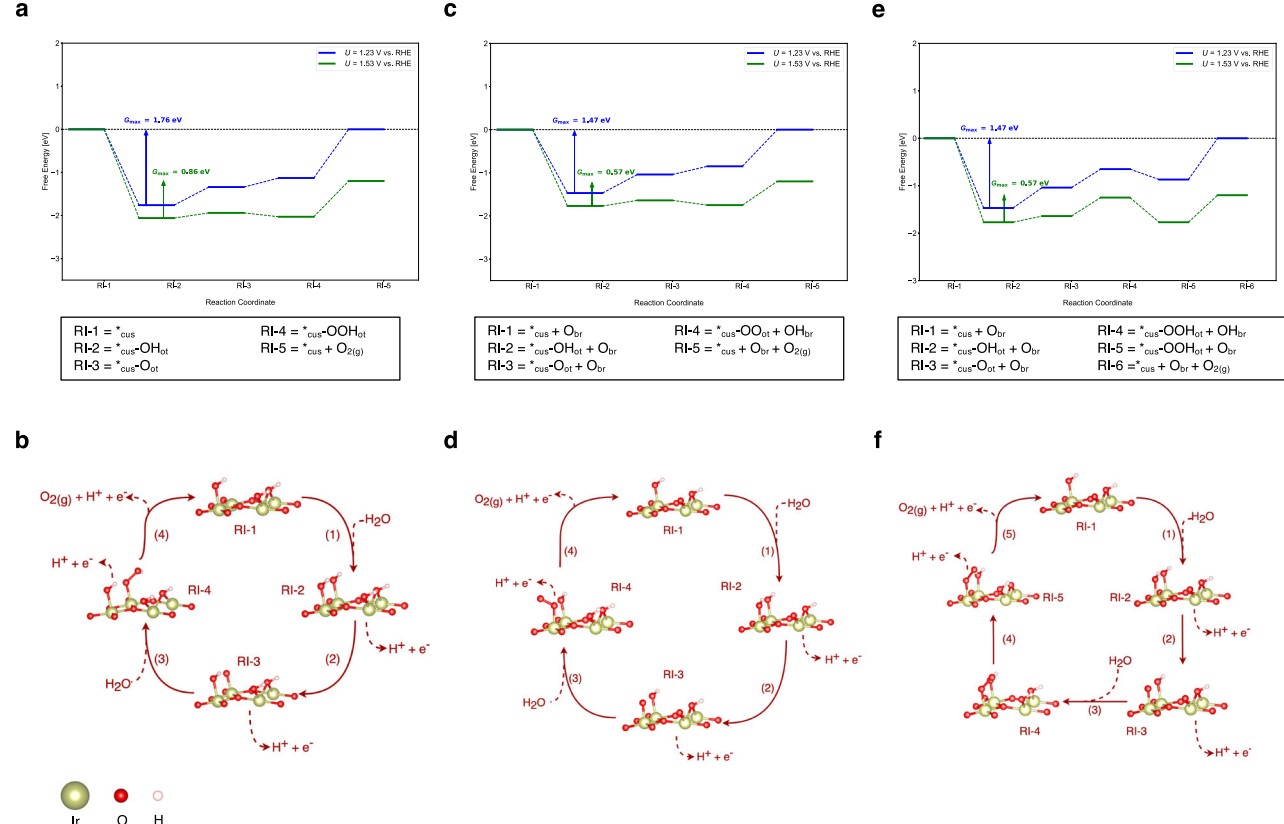

**Fig. 3 | Free-Energy Diagrams and Mechanistic Schemes for Conventional OER Mechanisms.** Free-energy diagram for the **a** mononuclear mechanism, **c** bifunctional I mechanism, and **e** bifunctional II mechanism on the fully hydroxylated $IrO_2(110)$ surface at 1.23 V and 1.53 V vs. RHE. The reaction intermediates of the mechanistic cycle are labeled on the *x*-axis. Blue and green solid lines indicate intermediates' free energies at 1.23 V and 1.53 V, respectively. Colored arrows indicate the free-energy span governing $G_{max}(U)$, with the respective value displayed. **b**, **d**, **f** Schematic illustration of the mononuclear, bifunctional I, and bifunctional II mechanisms, as described in Section 3, on the fully hydroxylated $IrO_2(110)$ surface. Numbers next to the arrows indicate the step sequence, and each structure represents the corresponding reaction intermediate.

considered in this work, the corresponding free-energy diagrams for the Walden pathways are provided in sections 9–12 of the SI. A summary of the mechanistic analysis is given in Table 3.

Table 3 indicates that the electrocatalytic activity in the approximation of $G_{max}(U)$ is on the order of 0.12 – 0.22 eV for the Walden pathways at $U = 1.53$ V vs. RHE, except for the partly *OOH-covered surface that reveals $G_{max}(U) = 0.35$ eV. This corresponds to a reduction of about 0.3 – 0.6 eV compared to the traditional OER mechanisms (cf. Tables 1–2). Therefore, we arrive at the intriguing finding that the OER on $IrO_2(110)$ is governed by Walden-type pathways rather than by any of the previously assumed reaction mechanisms, and this conclusion can be rendered in an unbiased manner by the descriptor $G_{max}(U)$ as the free-energy difference between the different pathways clearly exceeds the threshold of 0.20 eV[33].

Our extended mechanistic analysis also reveals that there is a change in the limiting free-energy span in the Walden-type pathways (cf. Table 3) compared to the traditional mechanisms (cf. Tables 1–2). While the latter are mainly governed by the decomposition of the *OOH adsorbate, the Walden pathways circumvent the vacant *cus site so that the limiting free-energy span shifts from *cus-OOHot → *cus + O2 to *cus-OHot → *cus-Oot. This alteration in the limiting span may explain the enhanced electrocatalytic activity of the Walden pathways on the different $IrO_2(110)$ surface configurations.

### Kinetics of water-assisted O₂ removal

In the previous section of this manuscript, we have discussed that Walden pathways outperform traditional pathways when referring to the picture of thermodynamic considerations by using descriptor-based analysis in the realm of $G_{max}(U)$. It is still unclear whether the desorption of O₂ with the help of a water molecule is also kinetically favored compared to conventional pathways in which O₂ desorption takes place without direct replacement by a solvent molecule. Therefore, we have investigated the kinetics of O₂ desorption by determining the transition state for the water-mediated Walden-type and the conventional O₂ desorption. Details of the transition state calculations and an in-depth discussion can be found in Section 15 of the SI. Figure 5 depicts the free-energy diagram for the conventional and Walden-type OER mechanisms using the example of the partly hydroxylated $IrO_2(110)$ surface – $2O_{br} + 2$*cus-OHot – at $U = 1.53$ V vs. RHE.

Our analysis reveals that the desorption barrier of O₂ is on the order of about 1 eV, and the activation barrier for the water-assisted route in the framework of a Walden step (cf. Figure 5b) is 0.07 eV lower in free energy than the desorption of O₂ without the involvement of a water molecule (cf. Figure 5a). The comparative analysis of the free-energy landscape in Fig. 5 reveals that the Walden-type pathway is preferred over the traditional pathway due to the consideration of thermodynamic and kinetic factors. We note that conventional electrolyzers operate at 80–90 °C, and at these temperatures, free-energy barriers of chemical steps above 1 eV are unproblematic. Furthermore, we do not aim at a quantitative discussion of the free energy barriers for O₂ desorption, but rather in the qualitative trends of these barriers. Indeed, Ping et al.[20] reported an O₂ desorption barrier of 0.56 eV for $IrO_2(110)$, indicating that water-mediated O₂ desorption is not kinetically hindered even at

**Table 1 | Energetic assessment of the various mechanistic pathways (cf. Figure 2) for the OER over a fully and partly hydroxylated IrO$_2$(110) surface (cf. Figure 1b–c) by the descriptor $G_{max}(U)$ at $U$ = 1.23 V and 1.53 V vs. RHE**

| Mechanisms | $U$ = 1.23 V | | $U$ = 1.53 V | |
|---|---|---|---|---|
| | $G_{max}(U)$ (eV) | Free-energy span | $G_{max}(U)$ (eV) | Free-energy span |
| **Fully hydroxylated IrO$_2$(110)** | | | | |
| Mononuclear mechanism | 1.76 | $^*_{cus}$-OH$_{ot}$ → $^*_{cus}$ + O$_2$ | 0.86 | $^*_{cus}$-OH$_{ot}$ → $^*_{cus}$ + O$_2$ |
| Bifunctional I mechanism | 1.47 | $^*_{cus}$-OH$_{ot}$ + O$_{br}$ → $^*_{cus}$ + O$_{br}$ + O$_2$ | 0.57 | $^*_{cus}$-OH$_{ot}$ + O$_{br}$ → $^*_{cus}$ + O$_{br}$ + O$_2$ |
| Bifunctional II mechanism | 1.47 | $^*_{cus}$-OH$_{ot}$ + O$_{br}$ → $^*_{cus}$ + O$_{br}$ + O$_2$ | 0.57 | $^*_{cus}$-OH$_{ot}$ + O$_{br}$ → $^*_{cus}$ + O$_{br}$ + O$_2$ |
| Oxide mechanism | - | - | - | - |
| Binuclear mechanism | 3.46 | $^*_{cus}$-OH$_{ot}$ + $^*_{cus}$-OH$_{ot}$ → $^*_{cus}$ + $^*_{cus}$ + O$_2$ | 2.86 | $^*_{cus}$-OH$_{ot}$ + $^*_{cus}$-OH$_{ot}$ → $^*_{cus}$ + $^*_{cus}$ + O$_2$ |
| **Partly hydroxylated IrO$_2$(110)** | | | | |
| Mononuclear mechanism | 1.45 | $^*_{cus}$-OH$_{ot}$ → $^*_{cus}$ + O$_2$ | 0.75 | $^*_{cus}$-OOH$_{ot}$ → $^*_{cus}$ + O$_2$ |
| Bifunctional – I mechanism | 1.45 | $^*_{cus}$-OH$_{ot}$ + O$_{br}$ → $^*_{cus}$ + O$_{br}$ + O$_2$ | 0.68 | $^*_{cus}$-OO$_{ot}$ + OH$_{br}$ → $^*_{cus}$ + O$_{br}$ + O$_2$ |
| Bifunctional – II mechanism | 1.45 | $^*_{cus}$-OH$_{ot}$ + O$_{br}$ → $^*_{cus}$ + O$_{br}$ + O$_2$ | 0.75 | $^*_{cus}$-OOH$_{ot}$ + O$_{br}$ → $^*_{cus}$ + O$_{br}$ + O$_2$ |
| Oxide mechanism | 1.07 | $^*_{cus}$-O$_{ot}$ + $^*_{cus}$-O$_{ot}$ → $^*_{cus}$-OOH$_{ot}$ + $^*_{cus}$-OOH$_{ot}$ | 0.66 | $^*_{cus}$-OOH$_{ot}$ + $^*_{cus}$-O$_{ot}$ → $^*_{cus}$-OOH$_{ot}$ + $^*_{cus}$-OOH$_{ot}$ |
| Binuclear mechanism | 2.78 | $^*_{cus}$-OH$_{ot}$ + $^*_{cus}$-OH$_{ot}$ → $^*_{cus}$ + $^*_{cus}$ + O$_2$ | 2.18 | $^*_{cus}$-OH$_{ot}$ + $^*_{cus}$-OH$_{ot}$ → $^*_{cus}$ + $^*_{cus}$ + O$_2$ |

For both potential conditions, the limiting free-energy span in the approximation of $G_{max}(U)$ is indicated. Further details are provided in section 4 and 5 of the SI.

**Table 2 | Energetic assessment of the various mechanistic pathways (cf. Figure 2) for the OER over a fully oxygen-covered and partly *OOH-covered IrO$_2$(110) surface (cf. Figure 1d-e) by the descriptor $G_{max}(U)$ at $U$ = 1.23 V and 1.53 V vs. RHE**

| Mechanisms | $U$ = 1.23 V | | $U$ = 1.53 V | |
|---|---|---|---|---|
| | $G_{max}(U)$ (eV) | Free-energy span | $G_{max}(U)$ (eV) | Free-energy span |
| **Fully oxygen-covered IrO$_2$(110)** | | | | |
| Mononuclear mechanism | 1.20 | $^*_{cus}$-OH$_{ot}$ → $^*_{cus}$ + O$_2$ | 0.41 | $^*_{cus}$-OOH$_{ot}$ → $^*_{cus}$ + O$_2$ |
| Bifunctional – I mechanism | 1.21 | $^*_{cus}$-OH$_{ot}$ + O$_{br}$ → $^*_{cus}$ + O$_{br}$ + O$_2$ | 0.38 | $^*_{cus}$-OO$_{ot}$ + OH$_{br}$ → $^*_{cus}$ + O$_{br}$ + O$_2$ |
| Bifunctional – II mechanism | 1.21 | $^*_{cus}$-OH$_{ot}$ + O$_{br}$ → $^*_{cus}$ + O$_{br}$ + O$_2$ | 0.39 | $^*_{cus}$-OOH$_{ot}$ + O$_{br}$ → $^*_{cus}$ + O$_{br}$ + O$_2$ |
| Oxide mechanism | 1.07 | $^*_{cus}$-O$_{ot}$ + $^*_{cus}$-O$_{ot}$ → $^*_{cus}$-OOH$_{ot}$ + $^*_{cus}$-OOH$_{ot}$ | 0.66 | $^*_{cus}$-OOH$_{ot}$ + $^*_{cus}$-O$_{ot}$ → $^*_{cus}$-OOH$_{ot}$ + $^*_{cus}$-OOH$_{ot}$ |
| Binuclear mechanism | 2.78 | $^*_{cus}$-OH$_{ot}$ + $^*_{cus}$-OH$_{ot}$ → $^*_{cus}$ + $^*_{cus}$ + O$_2$ | 2.18 | $^*_{cus}$-OH$_{ot}$ + $^*_{cus}$-OH$_{ot}$ → $^*_{cus}$ + $^*_{cus}$ + O$_2$ |
| **Partly *OOH-covered IrO$_2$(110)** | | | | |
| Mononuclear mechanism | 1.61 | $^*_{cus}$-OH$_{ot}$ → $^*_{cus}$-OOH$_{ot}$ | 1.01 | $^*_{cus}$-OH$_{ot}$ → $^*_{cus}$-OOH$_{ot}$ |
| Bifunctional – I mechanism | 1.46 | $^*_{cus}$-OH$_{ot}$ + O$_{br}$ → $^*_{cus}$ + O$_{br}$ + O$_2$ | 0.56 | $^*_{cus}$-OH$_{ot}$ + O$_{br}$ → $^*_{cus}$ + O$_{br}$ + O$_2$ |
| Bifunctional – II mechanism | 1.61 | $^*_{cus}$-OH$_{ot}$ + O$_{br}$ → $^*_{cus}$-OOH$_{ot}$ + O$_{br}$ | 1.01 | $^*_{cus}$-OH$_{ot}$ + O$_{br}$ → $^*_{cus}$-OOH$_{ot}$ + O$_{br}$ |
| Oxide mechanism | 1.07 | $^*_{cus}$-O$_{ot}$ + $^*_{cus}$-O$_{ot}$ → $^*_{cus}$-OOH$_{ot}$ + $^*_{cus}$-OOH$_{ot}$ | 0.66 | $^*_{cus}$-OOH$_{ot}$ + $^*_{cus}$-O$_{ot}$ → $^*_{cus}$-OOH$_{ot}$ + $^*_{cus}$-OOH$_{ot}$ |
| Binuclear mechanism | - | - | - | - |

For both potential conditions, the limiting free-energy span in the approximation of $G_{max}(U)$ is indicated. Further details are provided in section 6 and 7 of the SI.

room temperature. This also illustrates the sensitivity of activation free energies on the precise computational details, with no obvious way of pinpointing the physically relevant value.

The descriptor $G_{max}(U)$ suggests that the kinetic bottleneck in the Walden-type pathway is related to elementary steps involving charge transfer. This is further supported by experiments at the single-crystal level: Suntivich and coworkers reported Tafel slopes of 49 mV/dec. and 78 mV/dec. for the OER on IrO$_2$(110) in acid, highlighting that the reaction rate is determined by an electrochemical rather than a chemical step[22,65]. Considering that the water-assisted desorption-adsorption step is a chemical step, we do not propose the Walden-type water-mediated O$_2$ desorption as the rate-determining step (RDS) in the OER over IrO$_2$(110). An unbiased and clear-cut determination of the RDS would require the assessment of all possible transition states for all elementary steps in the catalytic cycle of the Walden pathway[65]. In addition, several elementary steps could contribute to the reaction rate, which can only be described by advanced analytical techniques, such as exploiting the degree of rate control[66]. However, such an analysis goes far beyond the scope of the present manuscript.

### Charge span as a descriptor for electrocatalytic processes

Identifying Walden-type mechanisms as the dominating pathway for the OER over IrO$_2$(110) can be seen as a paradigm change since, hitherto, this category of mechanisms has been largely overlooked for the modeling of proton-coupled electron transfer steps in electrocatalysis[39,40]. While we have provided reasoning for the importance of Walden-type pathways in the OER over IrO$_2$(110) based on thermodynamic and kinetic considerations in the realm of free-energy diagrams (cf. Figure 5), further evidence for the occurrence of Walden steps is given by Bader charge analysis using the example of the fully oxygen-covered IrO$_2$(110) surface – 2O$_{br}$ + 2$^*_{cus}$-O$_{ot}$ at $U$ = 1.53 V vs. RHE[67]. We determine the charge state of the active Ir site ($^*_{cus}$) in both the mononuclear and mononuclear-Walden mechanisms, depicted in Fig. 6a, b, respectively. In the traditional mononuclear mechanism (cf. Figure 6a), the charge state of the Ir atom at the $^*_{cus}$ site undergoes multiple fluctuations during the catalytic cycle, ranging from +1.47$e$ for the vacant active site, $^*_{cus}$, up to +1.85$e$ for the $^*_{cus}$-O$_{ot}$ adsorbate. On the other hand, changes in the charge state of the active $^*_{cus}$ site are much less pronounced for the Walden pathway (cf. Figure 6b). By

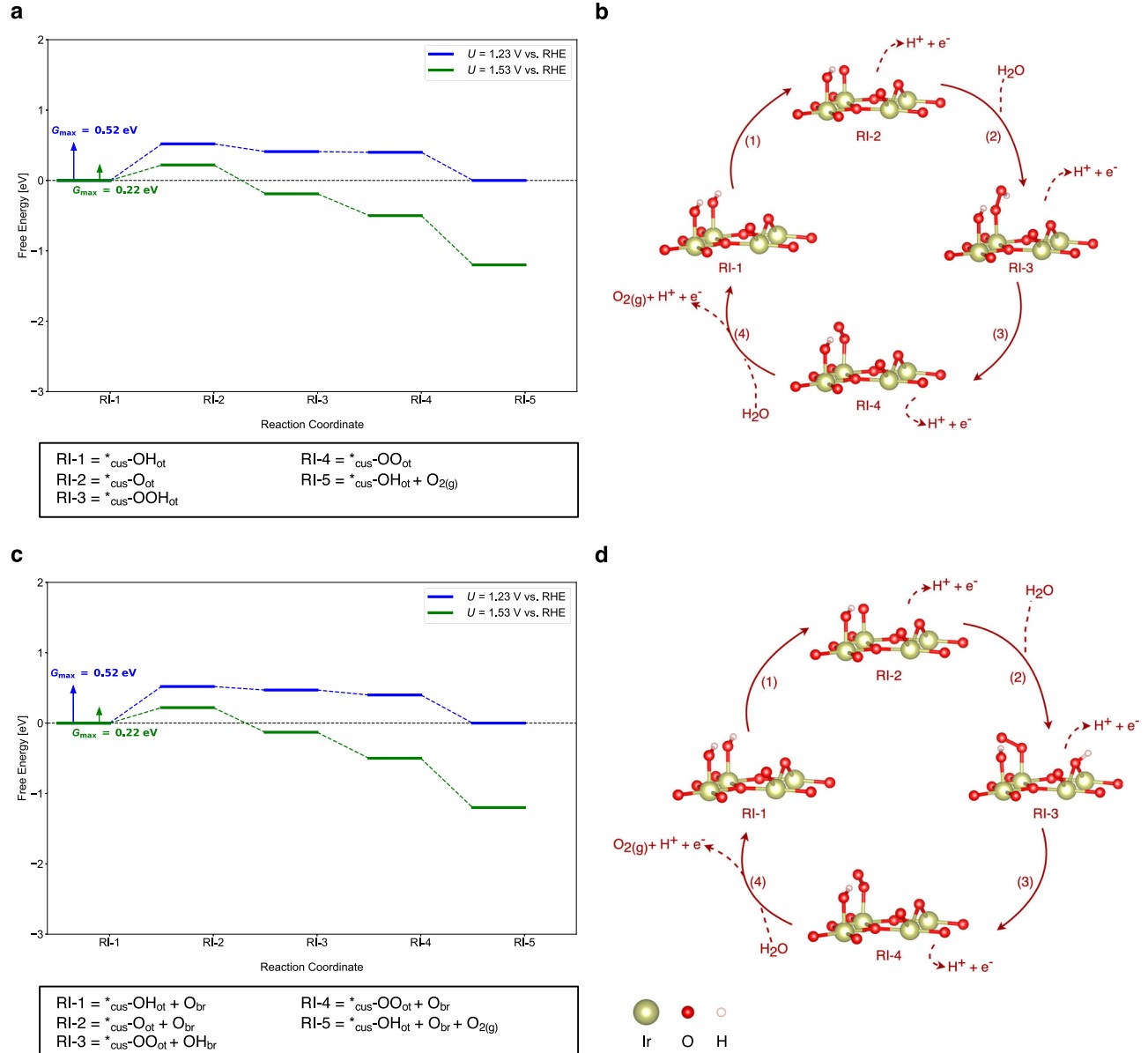

**Fig. 4 | Free-Energy Diagrams and Mechanistic Schemes for Walden-type OER Mechanisms.** **a** Free-energy diagram for the mononuclear-Walden mechanism and **c** bifunctional-Walden mechanism on the partly hydroxylated $IrO_2(110)$ surface at 1.23 V and 1.53 V vs. RHE. The reaction intermediates of the mechanistic cycle are labeled on the *x*-axis. Blue and green solid lines indicate intermediates' free energies at 1.23 V and 1.53 V, respectively. Colored arrows indicate the free-energy span governing $G_{max}(U)$, with the respective value displayed. **b**, **d** Schematic illustration of the mononuclear-Walden and bifunctional-Walden mechanisms, as described in Section 8, on the partly hydroxylated $IrO_2(110)$ surface. Numbers next to the arrows indicate the step sequence, and each structure represents the corresponding reaction intermediate.

defining $Q_{max}$ as the largest charge span in the catalytic cycle, we obtain $Q_{max} = +0.38e$ and $+0.17e$ for the mononuclear and mononuclear-Walden mechanism mechanisms, respectively. Intriguingly, these spans scale with the values of the activity descriptor $G_{max}(U = 1.53\,V)$ for the mononuclear and mononuclear-Walden mechanisms, which are 0.41 eV and 0.13 eV, respectively (cf. Figure 6).

Based on Fig. 6, we conclude that the stabilized charge state of the Ir atom at the $*_{cus}$ site in the mononuclear-Walden mechanism is linked to its enhanced activity compared to the traditional mechanism. Therefore, we propose that, besides the ubiquitous assessment of adsorption free energies to approximate the electrocatalytic activity, a span model based on the charge states of the active site in the catalytic cycle can be used to gain further insight into proton-coupled electron transfer steps in energy conversion and storage, and this statement may also hold for catalytic processes beyond the OER. A discussion of

the charge span approach in the context of the oxide mechanism[24] can be found in section 17 of the SI.

**Comparison with previous theoretical works and experiments**
While we emphasize the importance of Walden-type steps and mechanisms for the theoretical description of electrocatalytic processes, we note that a few previous works have already investigated concerted desorption-adsorption steps for the OER over $IrO_2(110)$. In the works by González et al. and Ping et al.[20,68], the authors only included a single option for the desorption of $O_2$ into their theoretical model – $*_{cus}\text{-}OO_{ot} + H_2O \rightarrow *_{cus}\text{-}(OH_2)_{ot} + O_{2(g)}$ (cf. Figure 5b)–but overlooked the conventional $O_2$ desorption step – $*_{cus}\text{-}OO_{ot} \rightarrow *_{cus} + O_{2(g)}$ (cf. Figure 5a) – in the analysis. In the present work, we close this gap by investigating the thermodynamics and kinetics of both elementary steps of $O_2$ desorption on the $IrO_2(110)$ surface. This allows us to obtain

**Table 3 | Energetic assessment of the Walden pathways (cf. Figure 4) for the OER over different surface motifs of a single-crystalline IrO$_2$(110) electrode (cf. Figure 1b–e) by the descriptor $G_{max}(U)$ at $U$ = 1.23 V and 1.53 V vs. RHE**

| Mechanisms | $U$ = 1.23 V | | $U$ = 1.53 V | |
|---|---|---|---|---|
| | $G_{max}(U)$ (eV) | Free-energy span | $G_{max}(U)$ (eV) | Free-energy span |
| **Fully hydroxylated IrO$_2$(110) surface** | | | | |
| Mononuclear-Walden | 0.63 | *$_{cus}$-OH$_{ot}$ → *$_{cus}$-OOH$_{ot}$ | 0.12 | *$_{cus}$-OH$_{ot}$ → *$_{cus}$-O$_{ot}$ |
| Bifunctional-Walden | 0.62 | *$_{cus}$-OH$_{ot}$ + O$_{br}$ → *$_{cus}$-OO$_{ot}$ + OH$_{br}$ | 0.13 | *$_{cus}$-OH$_{ot}$ + O$_{br}$ → *$_{cus}$-O$_{ot}$ + O$_{br}$ |
| **Partly hydroxylated IrO$_2$(110) surface** | | | | |
| Mononuclear-Walden | 0.52 | *$_{cus}$-OH$_{ot}$ → *$_{cus}$-O$_{ot}$ | 0.22 | *$_{cus}$-OH$_{ot}$ → *$_{cus}$-O$_{ot}$ |
| Bifunctional-Walden | 0.52 | *$_{cus}$-OH$_{ot}$ + O$_{br}$ → *$_{cus}$-O$_{ot}$ + O$_{br}$ | 0.22 | *$_{cus}$-OH$_{ot}$ + O$_{br}$ → *$_{cus}$-O$_{ot}$ + O$_{br}$ |
| **Fully oxygen-covered IrO$_2$(110) surface** | | | | |
| Mononuclear-Walden | 0.53 | *$_{cus}$-OH$_{ot}$ → *$_{cus}$-OOH$_{ot}$ | 0.13 | *$_{cus}$-OH$_{ot}$ → *$_{cus}$-O$_{ot}$ |
| Bifunctional-Walden | 0.52 | *$_{cus}$-OH$_{ot}$ + O$_{br}$ → *$_{cus}$-OO$_{ot}$ + OH$_{br}$ | 0.13 | *$_{cus}$-OH$_{ot}$ + O$_{br}$ → *$_{cus}$-O$_{ot}$ + O$_{br}$ |
| **Partly OOH-covered IrO$_2$(110) surface** | | | | |
| Mononuclear-Walden | 1.61 | *$_{cus}$-OH$_{ot}$ → *$_{cus}$-OOH$_{ot}$ | 1.01 | *$_{cus}$-OH$_{ot}$ → *$_{cus}$-OOH$_{ot}$ |
| Bifunctional-Walden | 1.09 | *$_{cus}$-OH$_{ot}$ + O$_{br}$ → *$_{cus}$-OO$_{ot}$ + O$_{br}$ | 0.35 | *$_{cus}$-OH$_{ot}$ + O$_{br}$ → *$_{cus}$-O$_{ot}$ + O$_{br}$ |

For both potential conditions, the limiting free-energy span in the approximation of $G_{max}(U)$ is indicated. Further details are provided in sections 9–12 of the supplemental.

reasonable statistics for the claim that, regardless of surface configuration, Walden-type mechanisms are energetically preferable to traditional mechanistic descriptions in the OER over IrO$_2$(110).

We note that the focus of the present work is on elucidating the elementary steps of the OER over IrO$_2$(110) and the mechanistic diversity considered requires the application of a community-standardized approach with respect to the application of the CHE approach, with some corrections to account for the applied electrode potential and solvation. Despite this, we would like to emphasize that there are approaches in the literature that go beyond conventional schemes to describe proton-coupled electron transfer steps and often rely on computationally intensive ab initio molecular dynamics simulations[69–73]. Although the consideration of improved schemes for describing the elementary steps of the OER could change the obtained estimates of free energies and electrocatalytic activity to some extent, the provided analysis of the OER over IrO$_2$(110) is robust due to the large difference of the $G_{max}(U)$ descriptor between traditional and Walden pathways. Going beyond the conventional scheme of the CHE approach is nevertheless desirable as it could clarify the effects of hydrogen bonding networks[74] on the relative stability of intermediates adsorbed on transition metal oxides.

There are already computational studies in the literature that dealt with the OER over IrO$_2$(110) and went beyond the thermodynamic picture[20,21,23,24]. In this context, it is important to mention the previous work of Binninger and Doublet, who, based on the evaluation of free-energy barriers for chemical reaction steps, proposed the oxide mechanism containing an Ir-OOOO-Ir association step as the preferred pathway for OER on IrO$_2$(110). In section 17 of the SI, we provide a detailed discussion of Binninger's approach and our analysis to the OER on IrO$_2$(110). There, we indicate that the discussion of free-energy barriers for chemical reaction steps as an indicator for electrocatalytic activity is subject to bias because it assumes a priori which elementary step limits the reaction rate. We believe that discussing the OER over IrO$_2$(110) using the $G_{max}(U)$ approach is the best compromise for a consistent and unbiased evaluation of the electrocatalytic activity of a model system. Future studies should aim to integrate machine learning and artificial intelligence approaches for the determination of transition states in an electrochemical environment because only the knowledge of all transition states, coupled with a degree of rate control analysis, would allow us to draw definite conclusions about the electrocatalytic activity, limiting

steps, and reaction mechanisms of proton-coupled electron transfer steps at electrified solid/ liquid interfaces.

To further validate the proposed Walden-type pathways in the OER over IrO$_2$(110), we compare the computed free-energy diagrams with experiments at the single crystal level. Although we have determined transition states related to the proposed Walden step (cf. Figure 5), we emphasize that these transition states cannot be directly compared with those determined in experimental single crystal studies. The reason for this is that the concerted desorption-adsorption process of the Walden step does not involve charge transfer and therefore this elementary step is unlikely to be one of the rate-determining steps (RDS) in the OER over IrO$_2$(110). Therefore, we perform microkinetic simulations[33,34] based on the evaluation of the descriptor $G_{max}(U)$ to estimate the current density ($j$) as a function of the applied electrode potential ($U$) for different surface configurations and reaction mechanisms. We refer to section 16 of the SI for a detailed overview of this procedure. A comparison with the experimental benchmark in the OER over IrO$_2$(110) based on the work of Kuo et al.[75,76] reveals that our theoretical model predicts current densities following the Walden-type pathway for the fully hydroxylated, partially hydroxylated and fully oxygen-covered IrO$_2$(110) surface that are in the same order of magnitude as the experiments. In contrast, there is a strong difference in current density for the Walden mechanism over the partly OOH-covered IrO$_2$(110) surface as well as for all traditional mechanisms over the different IrO$_2$(110) surface configurations with respect to the experimental benchmark. Therefore, we conclude that the IrO$_2$(110) surface is likely hydroxylated or covered with oxygen adsorbates under typical OER conditions, and our comparison with the experimental data further suggests the prevalence of Walden-type pathways over conventional OER mechanisms. A further comparison of the proposed Walden-type pathways with the oxide mechanism[24], containing an Ir-OOOO-Ir association step, is provided in section 17 of the SI.

## Discussion

In this work, we unravel the importance of Walden-type mechanisms in the OER over a single-crystalline IrO$_2$(110) model electrode. Previous theoretical considerations in the DFT approximation relied on the modeling of traditional mechanisms (cf. Figure 2), in which the catalytically active vacant Ir surface site is restored upon product formation. On the contrary, the simultaneous release of the product O$_2$ and adsorption of the reactant H$_2$O, which is denoted as a Walden step (cf.

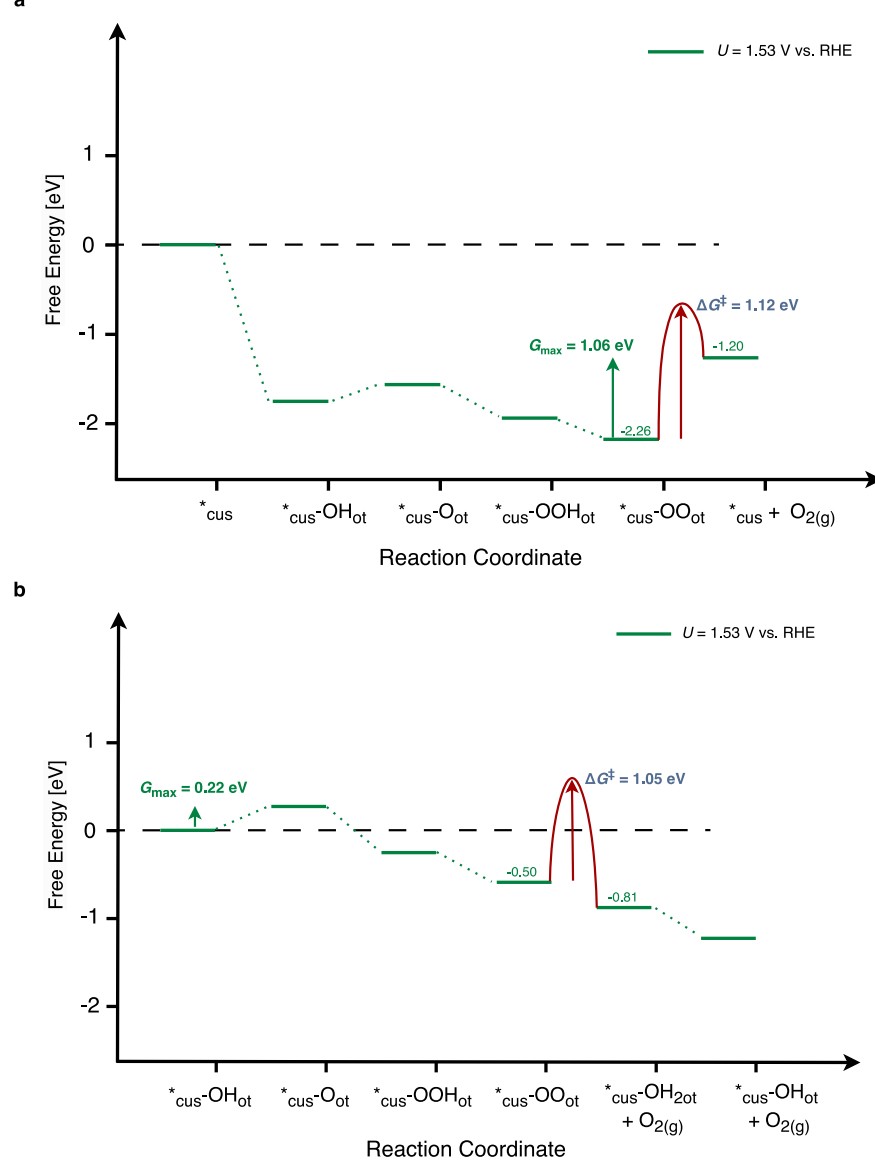

**Fig. 5 | Kinetics of Conventional and Water-Mediated O₂ Desorption. a** Free-energy diagram of a conventional OER mechanism over the partly hydroxylated IrO₂(110) surface at $U$ = 1.53 V vs. RHE. The calculated activation free energy for the desorption of the O₂ molecule (1.12 eV) is highlighted in red. **b** Free-energy diagram of the Walden-type OER mechanism over the partly hydroxylated IrO₂(110) surface at $U$ = 1.53 V vs. RHE. The calculated activation free energy for the water-assisted desorption of the O₂ molecule (1.05 eV) is highlighted in red.

Figure 4), has been largely omitted in earlier works aiming at the identification of limiting steps and reaction mechanisms.

We apply DFT calculations for a variety of different IrO₂(110) surface configurations, ranging from hydroxylated to oxygen- and OOH-covered phases, as these configurations are thermodynamically stable under typical OER conditions (cf. Figure 1). The elementary reaction steps and mechanisms in the OER are evaluated by the construction of free-energy diagrams connected with descriptor-based analyses in the realm of the activity measure $G_{max}(U)$. We pinpoint that skipping the catalytically active vacant Ir surface site by means of Walden-type mechanisms is beneficial for the catalysis, as it leads to a reduction of the activity descriptor $G_{max}(U)$ by about 0.3–0.6 eV compared to the traditional mechanisms. This picture does not change even if the kinetics in terms of the transition states are considered in the analysis of free-energy diagrams (cf. Figure 5). The energetic picture is connected to the analysis of the charge state for the active Ir (*cus) surface site, indicating that a span model based on the lowest and highest Bader charges during the catalytic cycle scales with the activity descriptor $G_{max}(U)$ (cf. Figure 6). We propose to apply the presented methodology of combining free-energy diagrams and charge state analyses to electrocatalytic processes beyond the OER to gain further insight into the factors controlling the complex proton-coupled electron transfer steps at electrified solid/liquid interfaces. These findings can be used in future research on materials discovery using electronic structure theory and artificial intelligence methods to identify improved OER catalysts.

## Methods
### DFT parameters
All DFT calculations are performed with the Vienna ab initio Simulation Package (VASP)[77–79] using the Perdew-Burke-Ernzerhof (PBE) and revised PBE functionals for correlation and exchange[80,81] as well as Grimme's D3 correction to account for dispersion effects[82]. The projector augmented wave (PAW) method is used to describe the interaction between core electrons and valence electrons[83], and the cutoff energy is set to 440 eV. We apply the Methfessel-Paxton smearing

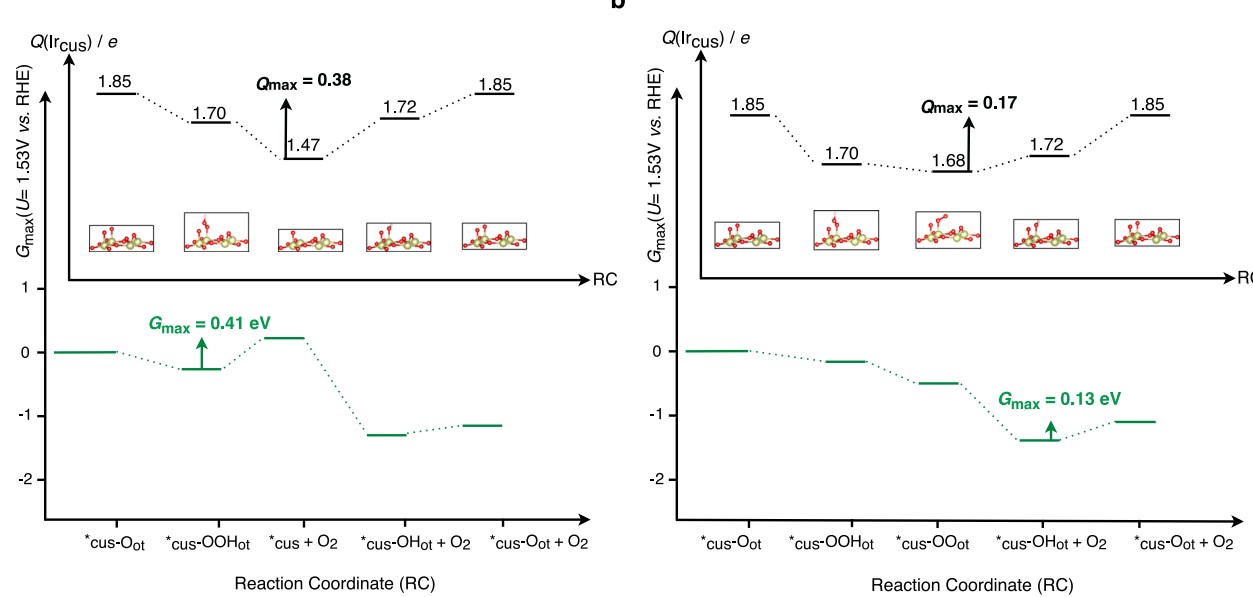

**Fig. 6 | Charge Span and Free-Energy Span as a Descriptor for OER Activity.** Charge states (upper panel) and free-energy diagram (lower panel) of the reaction intermediates in the OER over the fully oxygen-covered IrO$_2$(110) surface (cf. Figure 1d) for (**a**) the mononuclear mechanism and **b** the mononuclear-Walden mechanism at $U = 1.53$ V vs. RHE. The largest charge span, $Q_{max}$, scales with the activity descriptor $G_{max}(U)$ in both pathways.

method with a smearing width of 0.20 eV, and the electronic energy is considered self-consistent when the energy change was smaller than $10^{-6}$ eV. Geometry optimization is considered converged when the change in forces is smaller than 0.01 eV/Å.

### Surface calculations

Rutile IrO$_2$ is described by a (2 × 1) surface slab model along the (110) direction, and a 7 × 7 × 1 Γ-centered k-point mesh is applied to sample the Brillouin zone for the numerical integration in the reciprocal space.

### Computational hydrogen electrode approach

Free energies of reaction intermediates in the OER are obtained by applying the computational hydrogen electrode (CHE) approach of Nørskov and coworkers[84], thereby making use of gas-phase error corrections such as reported by Calle-Vallejo and coworkers to meet the experimental equilibrium potential of OER[85,86]. We refer to section 2.2 of the SI for further details on the CHE approach.

### Pourbaix diagrams

Analysis of the obtained adsorption free energies facilitates constructing Pourbaix diagrams, aiming at the determination of IrO$_2$(110) surface configurations under OER conditions. We refer to section 2.3 of the SI for further details.

### Reaction mechanisms of oxygen evolution reaction

OER over IrO$_2$(110) is described by five different reaction mechanisms (cf. section 3 of the SI) and two different Walden-type pathways (cf. section 8 of the SI). Note that we do not consider lattice oxygen evolution[87–92] in our model as it has been demonstrated both experimentally and theoretically that there is no lattice exchange for rutile IrO$_2$(110). This justifies refraining from pathways that contain reconstruction of the surface or lattice oxygen in the evaluation.

### Descriptor-based analysis

Electrocatalytic activity of the different reaction mechanisms over the active IrO$_2$(110) surface configurations under OER conditions is described by the descriptor $G_{max}(U)$[33,34]. We provide adsorption free energies, free-energy diagrams, limiting spans, and activity analyses

for all reaction mechanisms over the active IrO$_2$(110) surface configurations in sections 4–7 and 9–12 of the SI. We benchmark our electronic structure calculations by comparing the obtained results relating to adsorption free energies and activity predictions based on the CHE approach with implicit solvation using the VASPsol package (cf. sections 4.6, 5.6, 6.6, and 7.6 in the SI). In addition, we apply grand canonical DFT calculations (cf. section 6.7 of the SI) to evaluate the energetics under constant potential[93,94] rather than under constant charge as encountered with the CHE approach.

### Bader charge analysis

For the assessment of charge states during the catalytic OER cycle, we apply Bader charge analysis using the script of the Henkelman group for VASP[67].

## Data availability

The data generated in this study have been deposited in the Zenodo repository database without accession code [https://doi.org/10.5281/zenodo.15650140][95].

## Code availability

The DFT codes generated in this study have been deposited in the Zenodo repository database without accession code [https://doi.org/10.5281/zenodo.15650696][96].

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

## Acknowledgements

M.U., C.H., and K.S.E. thank the RESOLV Cluster of Excellence, funded by the Deutsche Forschungsgemeinschaft under Germany's Excellence Strategy – EXC 2033 – 390677874 – RESOLV, for financial support to carry out this study. CH and KSE further acknowledge funding by the CRC/ TRR247: "Heterogeneous Oxidation Catalysis in the Liquid Phase" (Project number 388390466-TRR 247). SNS gratefully acknowledges support from the CBPsmn (PSMN, Pôle Scientifique de Modélisation Numérique) of the ENS de Lyon for computing resources. The platform operates the SIDUS solution (Quemener, E. & Corvellec, M. SIDUS—the solution for extreme deduplication of an operating system. Linux J.2013, 3:3 (2013). https://dl.acm.org/doi/abs/10.5555/2555789.2555792).

## Author contributions

K.S.E. conceived the idea and directed the project. M.U. performed DFT calculations for IrO$_2$ related to thermodynamic aspects (adsorption free energies) in the OER. S.R. assisted M.U. in the DFT calculations for IrO$_2$. M.U. and S.N.S. performed DFT calculations for IrO$_2$ related to kinetic aspects (transition states) in the OER. C.H. provided the idea for the Bader charge analysis, and M.U. performed the calculations. M.U. ana-lyzed all data under the supervision of K.S.E. S.R., C.H., and S.N.S. contributed to the discussion of the results. M.U. prepared all figures. K.S.E. wrote the manuscript with contribution from all authors.

## Funding

## Competing interests

The authors declare no competing interests.
