## [Transparent Peer Review file · Nature Communications]

Oxygen evolution reaction on IrO₂(110) is governed by Walden-type mechanisms

Corresponding Author: Professor Kai S. Exner

Version 0:

Reviewer comments:

Reviewer #1

(Remarks to the Author)

The authors present a well structured and well written manuscript about density functional theory (DFT) calculations of the elementary steps of the oxygen evolution reaction on the (110) surface of IrO₂. They apply standard DFT calculations, including implicit solvation and the "computational hydrogen electrode scheme" with the common corrections for inclusion of the electrochemical potential. The authors assess aspects like surface termination and potential reaction steps. They discuss previously introduced relative free energy descriptors and propose that in previous simulations, an OER intermediate has been considered as rate determining step that can be replaced by a different step lower in energy which changes the common mechanistic hypotheses derived from DFT calculations. For this purpose, the authors introduce the concept of "Walden-type mechanisms" in which the replacement of the formed O₂ on the active site via a water molecule, leading back to the initial state, is proposed not to go through a high energy vacant site but via a concerted replacement reaction.

The paper is well very written, the methodology is mostly state-of-the-art and robust, I would consider it to be well suited for an interdisciplinary journal like Nature Communication, as it has potential impact in electrocatalysis and many aspects of water electrolysis, catalyst design and eventually sustainable energy production.

However, there are some points which might be criticized. While the proposition of a "Walden-type mechanism" for the regeneration of the active site is an interesting proposition, some chemists in catalysis might wonder why the authors consider this as such a remarkable discovery, as the OER in an electrolyzer takes place in aqueous medium and the formation of high energy vacant sites is rather something that could be expected in the gas phase at elevated temperature, but not in water electrolysis, where water is not only reactant, but also solvent. Note that in several existing studies (experimental and a few computational) on IrO_x materials the authors don't assume that this step is rate determining.

I would also state that the manuscript rather reads like an index for the supporting information - several really interesting aspects are not shown and in the manuscript the authors very frequently refer to the SI. The SI contains data on the relative energies of a remarkable 26 (!!) possible mechanisms and reactions paths. I would consider this the bulk of the results presented in this paper and it would be great if the authors would consider giving these results and their details more space in the discussion.

Hence, while I can recommend the work for publication in Nature Communications in principle, I would rather recommend the paper to be published as full paper after major revision to do the data and ideas justice.

Upon preparation for resubmission of the manuscript, I would suggest considering the following aspects :

1) A descriptor based approach might provide a help in interpreting the computational results but a solid link to experimental observables or at least a thorough comparison to experimental observations would help to underline the relevance of computed relative energies for non-theoreticians who

work on the actual materials. This would also help to judge the plausibility of the mechanistic hypothesis derived from simulations.

2) While the authors use the computational Pourbaix diagram scheme to determine the surface termination depending on pH and potential, the results seem a bit counterintuitive and contradict other published work:

Lee et al. [<https://doi.org/10.1038/s41467-022-30838-y>] presented a DFT simulation including AIMD studies in which they report a computational Pourbaix diagram for IrO₂, Wang et al. also reported computational Pourbaix diagrams for IrO₂ using DFT, the "SCAN" functional and an approach within the Materials Project [<https://doi.org/10.1038/s41524-020-00430-3>]. Flores et al. even used an active learning approach for the discovery of stable IrO_x polymorphs and present a revised Pourbaix diagram for Ir-H₂O system [<https://dx.doi.org/10.1021/acs.chemmater.0c01894>]. Bhattacharyya used a cluster approach to study the protonation state of the interface and discuss computational Pourbaix diagrams for their model system [<https://dx.doi.org/10.1021/acs.jpcc.0c10092>]. Papers by Gonzalez, Solans-Monfort et al. nicely demonstrate how DFT simulations can be used to identify the correct surface termination under acidic conditions and high potentials (<https://doi.org/10.1016/j.jcat.2021.02.026> and <https://doi.org/10.1039/D1NR03592D>). Dang et al. (Nat. Comm. 2012 doi.org/10.1038/s41467-021-26336-2) use schemes to assess the surface structure and termination (see also Klyukin et al. 2018 JPCC)

Comparing the work presented here, from Fig. 1 a) it appears, that all states of the model the authors use occur potential dependent, but each one is pH independent in the range of pH 0-4. Compared to most other computational (and experimental!) Pourbaix diagrams published (see above) this is very rarely the case. Note that the references 44-47 given by the authors are either not IrO_x or their own previous studies on different materials. Maybe the authors could compare their results to other similar studies and explain whether the strange behavior they find is physical or an artifact of their model which only allows for simultaneous de-protonation and oxidation of centers in the interface. This would be crucial, as all results are based on the corresponding surface termination the reaction is studied on.

3) The authors apply periodic boundary DFT calculations for the simulation of electrochemical reactions and mostly use a computational hydrogen electrode approach with some corrections for the inclusion of the electrochemical potential. This only allows to describe PCET steps and any chemical or electrochemical step in the mechanism is determined prior to the calculation. Solvent effects are included implicitly, so there is no direct interaction of the reaction center with solvent molecules. While this is a fairly common way of describing electrochemistry, there are several strategies that go beyond the implicit inclusion of effects like pH and solvation of the rigid CHE. Here it would be nice if the authors could at least acknowledge/cite some of the recent computational developments that go beyond their approximations, as especially the "conflicting results of previous theoretical studies" can be understood in the light of the shortcomings of common approximations (implicit solvation in this case).

There are plenty of examples in the literature - Yoo and Wippermann et al. present an AIMD scheme with explicit solvation and inclusion of the electrochemical potential for electrochemical reactions (<https://doi.org/10.1038/s41524-021-00529-1>). Bhattacharyya et al. describe the explicit inclusion of potential and pH in DFT models of the OER on IrO_x nanoparticles (<https://doi.org/10.1021/acs.jpcc.0c10092>). Other studies include explicit water molecules (Gonzalez et al. J. Catal. 2022 doi.org/10.1016/j.jcat.2022.05.023, Nanoscale 2021 doi.org/10.1039/D1NR03592D) For an insightful discussion of the issue for RuO₂ and IrO₂, see also Liberto et al. JPCC 2023 (doi.org/10.1021/acs.jpcc.3c02733). This also covers the topic of how hydrogen bonding networks on transition metal oxides influence the relative stability of intermediates. In all of these cases, new phenomena can be discussed and more detailed information about the physics and chemistry in the interface can be obtained. It would be nice if the authors could put their approximations into perspective with other approaches which are currently developed. Note that this should not only serve the purpose to give a thorough overview over the state-of-the-art in theory, but also allow the reader to assess possible sources or errors in the description.

4) As mentioned above, the authors have moved a considerable amount of results into the SI and opted to keep five tables in the manuscript. I would strongly recommend to move details like reaction paths / structures and plots of the relative free energies into the main manuscript and move numerical data to the SI. Note that there are several recent works (for example from the Goddard group on OER in alkaline conditions on materials like CoO_x and NiO_x), in which microkinetic models have been used to assess complex reaction networks on different surface sites. With the wealth of data the authors present, it would be a pity not to be able to directly compare the energetics of the different mechanisms and discuss bottlenecks and common intermediates in more detail. Note that an unfavorable mechanism might still exhibit a low energy intermediate that indicates possible catalyst poisoning or degradation.

In summary - the work presented here is quite extensive and a lot of interesting and novel results are outlined. However, I don't think a communication would do the work justice. I think the work would rather be suited for a journal like Nature Catalysis or Nature Chemistry. Then on the other hand, I'm not sure whether it has so much impact to scientists outside the computational catalysis community, as it is a purely computational work mostly addressed to theoreticians. The authors have chosen to go for Nature Communications and I respect this, so my overall recommendation is accept with major revision with several aspects that need thorough revision (especially with respect to the literature discussed).

Reviewer #2

(Remarks to the Author)

The authors report a systematic exploration of electro-catalytic pathways for the conversion of water into oxygen at the IrO₂ surface for four alternative surface terminations that are consistent with operating conditions. The results clarify some of the inconsistencies on the OER mechanism reported in the literature by different computational studies. Moreover the authors seem to identify a previously-unexplored catalytic step as the most relevant reaction mechanism for OER on IrO₂. The pathways identified in the first screening of the different IrO₂ surfaces all seem to involve the freeing of a 'cus' site as their limiting steps. Thus, the so-called Walden mechanism, a concerted adsorption/release step, provides a more thermodynamically favorable mechanism. As the concerted mechanism involves a solvent molecule, I assume there are no statistical drawback or kinetic limitations with respect to the separate release and adsorption, although the authors may want to expand on this aspect. The presentation of the manuscript is very clear and the results are produced following state-of-the-art approaches, with enough details to make them reproducible.

Reviewer #3

(Remarks to the Author)

In this manuscript Exner and Coworkers made DFT calculations of four different "traditional" OER mechanisms and two with Walden type steps for the OER on IrO₂(110). The elementary reaction steps and mechanisms in the OER are evaluated by constructing free-energy diagrams in conjunction with descriptor-based analyses in the range of the activity measure $G_{\text{max}}(\text{U})$. Their analysis shows that Walden type mechanism is beneficial for OER catalysis since it leads to a reduction in free-energy.

The analysis is thorough and the relationships are clearly described – overall a very good manuscript. I believe that only this type of approach can provide access to reliable values for comparison of energy parameters and provide valuable insights for understanding the processes at the IrO₂ interface in water electrolysis. Therefore, I think this article is very timely and important.

However, I have a few comments that I would like to have addressed by the authors:

Comment 1: This paper is a specific and thorough analysis mechanistic pathways of IrO₂(110) for OER. And indeed, IrO₂ is important to understand for OER, and their DFT calculations is also interesting.

Reading the authors prior publication of potential dependent switching of OER-mechanisms and of their investigation Walden type inversion (<https://doi.org/10.1039/D3MH00047H> and DOI: 10.1002/adv.202305505), I see this publication more as continued work to the older one – What is the key insight here that warrants publication in Nature Comm.? One could consider it in a more specific journal.

Comment 2: Maybe some suggestions could help to improve the manuscript to a broader readership. I believe the introduction could help the reader with a bit more discussing on other OER catalyst. Why was IrO₂(110) chosen? And how do energies change for other IrO₂ sites?

Moreover, from a practical point of view, what can I do with the knowledge gained in this work? How can I use it to improve OER catalysts?

Comment 3: Figure 2 gives a great overview of the different OER mechanisms. Only one thing doesn't seem right to me. Between step (1) and step (2), IrO₂(110) adsorb water and not release it, or am I missing something?

Reviewer #4

(Remarks to the Author)

This manuscript from the groups of Exner and Hättig reports on a computational-theoretical study of the oxygen evolution reaction (OER) on IrO₂. The authors present a comparison of different OER mechanisms based on the activity descriptor $G_{\text{max}}(\text{U})$, previously introduced by Exner and coworkers. In addition to previously reported OER mechanisms, the authors include a set of "Walden-type" mechanisms, where desorption of oxygen and adsorption of water at the Ir surface site are considered to occur simultaneously in one step. The authors conclude that the Walden-type mechanisms outperform the non-Walden equivalents in providing lowest $G_{\text{max}}(\text{U})$ and thus govern the OER activity on IrO₂(110).

Indeed, there exists some controversy with respect to the OER mechanism on the "archetype" IrO₂ catalyst and deciphering the respective mechanism is of great topical relevance. This manuscript deals with the central question whether the unsaturated state of the surface Ir cation after oxygen desorption can be avoided by simultaneous bond breaking and formation. The critical relevance of this question has recently been emphasised by Binniger et al. (Current Opinion in Electrochemistry 2023, 42:101382). Binniger et al. criticised the practise to consider the elementary steps of bond breaking during O₂ desorption and bond formation by H₂O (re)adsorption as only one "lumped" step, because this can hide the possibly unfavourable unsaturated state of the cation (Ir^{*}). The present study adopts an opposing view and suggests that both processes can indeed be treated as only one elementary step, similar to the Walden inversion, circumventing the unsaturated state of the Ir cation. This is certainly a very interesting perspective and a clear demonstration of such

simultaneous bond breaking and formation would significantly contribute to the understanding of the OER mechanism on IrO₂(110), and possibly beyond. However, the present manuscript presents no such evidence, since only initial and final states of each step are computed. Therefore, the suggested Walden step essentially consists in omitting the intermediate state (Ir*) from the sequence Ir-OOH → Ir* → Ir-OH. Neglecting the energetically unfavourable intermediate point on the free-energy profile, it is not surprising that "Walden-type" mechanisms seem to provide a lower overall barrier (characterised by G_{max}(U)). As such, the "Walden mechanisms" appear to correspond to a lumped version of conventional mechanisms. To show that they indeed represent a new pathway, the free-energy profile and intermediate/transition states along the Walden steps need to be computed. Ping et al. (Ref. 20, see Figure 7 therein) performed NEB calculations for such direct substitution of O₂ by H₂O on IrO₂(110). However, they did not observe simultaneous bond breaking and formation and concluded that O₂ desorption and H₂O adsorption rather proceed one after the other. The unsaturated Ir* state still appeared as the "transition state" and resulted in a comparably large energy barrier. I therefore think that the central claims of the present manuscript are not sufficiently supported. If, however, in a revised work, the authors will show unambiguous evidence that the proposed Walden steps indeed avoid the unsaturated Ir* (even at the transition state) and provide smaller barriers in comparison to other mechanisms, such demonstration would surely be of significant relevance.

A few additional comments and questions:

- The comparison of different mechanisms is not fully consistent in my opinion, because certain identical intermediates are included in some mechanisms, while neglected in others. As an example, I refer to the comparison between the mononuclear standard (Fig. S13) and Walden (Fig. S26) pathways. Steps RI-1 → RI-2 → RI-3 of the Walden mechanism (Fig. S26) are identical to steps RI-2 → RI-3 → RI-4 of the standard mechanism (Fig. S13), resulting in the formation of *OOH. The subsequent step in the Walden pathway consists in the deprotonation of *OOH, producing an *OO adsorbate (RI-4 in Fig. S26). In contrast, for the standard mechanism (Fig. S13), this deprotonation step is lumped together with the subsequent desorption of O₂ and the *OO adsorbate is entirely neglected. For consistency, the *OO adsorbate should be explicitly included in the standard mechanism.

- What is the difference between the proposed Walden-type steps in this study compared to other studies that considered O₂ desorption and H₂O adsorption as a single step, see, e.g., Scheme 1 a),b) in González et al., Journal of Catalysis 396 (2021) 192–201?

- The terminology of "Walden-type" mechanisms appears to be borrowed from the Walden inversion reaction, where one bond at a center gets broken due to formation of a new bond from the back side. I wonder whether such an inversion can happen at a surface Ir cation of IrO₂(110). Is this terminology thus appropriate?

- The authors state that they performed grand-canonical DFT (GC-DFT) calculations under constant potential using VASPsol. As far as I know, VASPsol does not offer the GC-DFT method. I would be interested in better understanding how the authors performed the GC-DFT method.

Version 1:

Reviewer comments:

Reviewer #1

(Remarks to the Author)

The authors present a resubmission of a previous manuscript. Already the original manuscript on their simulation work on the OER or IrO_x materials was insightful, novel and contained an extensive study on the topic. Following the suggestions of four referees, the authors have considerably extended their data, added further simulations and results, improved their citations and references and thoroughly worked on the clarity and presentation of their arguments. Especially the inclusion of further illustrations and data and the microkinetic modelling that was performed further improve the quality of the results. I would now consider this a very clear and conclusive work that contains many interesting aspects that theory can provide for novel catalysts for water electrolysis. Given the relevance of the subject and the refined methods applied by the authors, I would also consider the manuscript of broader interest and perfectly suited for Nature Communications. I recommend publication without further changes.

Reviewer #3

(Remarks to the Author)

The manuscript has been thoroughly revised, and all my comments have been adequately addressed. I therefore recommend it for publication without further comments.

Reviewer #4

(Remarks to the Author)

The new results added during revision disprove the asserted role of "Walden-type" mechanisms in the OER on IrO₂ (110). Hence, I do not consider this work to be suitable for publication.

As suggested, the authors have performed calculations of activation barriers of both the conventional O₂ desorption step and the concerted O₂ desorption-H₂O adsorption step (termed "Walden-type" by the authors). A barrier of 1.12 eV was

obtained for the former, while the barrier of the "Walden-type" step was 1.05 eV. The difference of 0.07 eV is insignificant and shows that both steps are governed by the energy cost for O₂ desorption alone, while the "assistance" by water is not significant. Given total barrier heights beyond 1 eV, none of these steps can be expected to proceed at room temperature at any relevant rate. Being "chemical" steps, the barriers are largely insensitive to electrode potential, which thus does not help. Finally, the structure of the transition state of the "Walden" step (Fig. S35) shows that O₂ desorption and H₂O adsorption, in fact, proceed one after the other, and there is no indication of a concerted bond breaking and formation. Seeing these results, the authors' conclusions appear incomprehensible to me.

From my point of view, these results show the following: First, O₂ desorption and H₂O adsorption proceed sequentially and, hence, there is no "Walden" step on IrO₂ (110), which is in agreement to previous findings by Ping et al. Second, neither the conventional O₂ desorption nor the water-assisted ("Walden") steps can proceed at relevant rates at ambient temperature on IrO₂ (110). Instead, the OER must proceed via the alternative pathway with a Ir-O-O-O-Ir association step, for which we found a feasible barrier of about 0.34 eV in our previous work (Energy Environ. Sci., 2022, 15, 2519). In the present work, the authors did not consider the barrier of this step.

-Tobias Binninger

Version 2:

Reviewer comments:

Reviewer #1

(Remarks to the Author)

I am very happy to see a critical, clear and constructive scientific discussion being carried in a factual and respectful way.

My impression is that on one hand, the discussion revolves around the definition of what a "Walden type" mechanism really is, for example how "concerted" it really is etc.

Exner outlines what he means exactly in Fig. R3 irrespective what one would call a mechanism like this, so I would say even though there is disagreement about the details of what a Walden mechanism includes and what the exact atomic arrangement might be, Exner's discussion and results are well documented and reproducible.

On the other hand, it seems that while Exner mainly states that a high energy intermediate after oxygen desorption is unlikely to be favorable no matter what the exact mechanism or the details are, Binninger's criticism mostly focuses on the finer details of the mechanism and its thermodynamics/kinetics.

While I respect Binninger's opinion, I would really agree with Exner's statement "Ultimately, however, we must acknowledge that all current works in the field of theoretical electrocatalysis are based on approximations ..."

To my mind it is unlikely that this debate will be resolved by theory alone, the best way is to clearly describe and discuss discrepancies and hope that eventually, it will be possible to find the relevant influences via experiment.

I think Exner's work is solid, clear, innovative and will be impactful and I strongly recommend publication of the manuscript in the current form, but I also respect the doubts of Binninger and would suggest the following solution:

Binninger's work is acknowledged in detail in the manuscript, so every reader can in principle look up the corresponding publications. However, I would suggest that Exner's detailed discussion given in the latest reply letter (page 8 and following) will be included in the SI of the paper. This way, the readers will be able to make up their mind about the interpretation of the results and how valid the one or the other claim is themselves.

Response:

We would like to express our sincere thanks to the four reviewers for their time and efforts in reviewing our manuscript. Their insightful comments and constructive criticisms have been carefully considered in this revised version of the article. The responses to the reviewers are organized as follows:

- | | |
|------------------------|---------|
| a) Reply to referee #1 | page 2 |
| b) Reply to referee #2 | page 14 |
| c) Reply to referee #3 | page 19 |
| d) Reply to referee #4 | page 27 |

We sincerely hope that with these revisions, our manuscript meets the high standards for publication in *Nature Communications*.

Sincerely,

Kai S. Exner (on behalf of all authors)

Reviewer #1

The authors present a well structured and well written manuscript about density functional theory (DFT) calculations of the elementary steps of the oxygen evolution reaction on the (110) surface of IrO₂. They apply standard DFT calculations, including implicit solvation and the "computational hydrogen electrode scheme" with the common corrections for inclusion of the electrochemical potential. The authors assess aspects like surface termination and potential reaction steps. They discuss previously introduced relative free energy descriptors and propose that in previous simulations, an OER intermediate has been considered as rate determining step that can be replaced by a different step lower in energy which changes the common mechanistic hypotheses derived from DFT calculations. For this purpose, the authors introduce the concept of "Walden-type mechanisms" in which the replacement of the formed O₂ on the active site via a water molecule, leading back to the initial state, is proposed not to go through a high energy vacant site but via a concerted replacement reaction.

The paper is well very written, the methodology is mostly state-of-the-art and robust, I would consider it to be well suited for an interdisciplinary journal like Nature Communication, as it has potential impact in electrocatalysis and many aspects of water electrolysis, catalyst design and eventually sustainable energy production.

Response: We thank the referee for their support of our manuscript for publication in Nature Communications.

However, there are some points which might be criticized. While the proposition of a "Walden-type mechanism" for the regeneration of the active site is an interesting proposition, some chemists in catalysis might wonder why the authors consider this as such a remarkable discovery, as the OER in an electrolyzer takes place in aqueous medium and the formation of high energy vacant sites is rather something that could be expected in the gas phase at elevated temperature, but not in water electrolysis, where water is not only reactant, but also solvent. Note that in several existing studies (experimental and a few computational) on IrO_x materials the authors don't assume that this step is rate determining.

I would also state that the manuscript rather reads like an index for the supporting information - several really interesting aspects are not shown and in the manuscript the authors very frequently refer to the SI. The SI contains data on the relative energies of a remarkable 26 (!) possible mechanisms and reactions paths. I would consider this the bulk of the results presented in this paper and it would be great if the authors would consider giving these results and their details more space in the discussion.

Hence, while I can recommend the work for publication in Nature Communications in principle, I would rather recommend the paper to be published as full paper after major revision to do the data and ideas justice.

Response: We thank the reviewer for pointing out that the discussion of mechanisms and reaction pathways in the SI deserves more recognition in the main text of our paper. We also agree with the reviewer that it makes sense to expand the current communication format to the format of a full research article. To this end, we resubmitted our manuscript as a research article and incorporated several results from the SI into the main text of our paper.

Change: In the revised version of our manuscript, we have made several extensions on pages 7-11 and pages 13-14. For the sake of brevity, we do not list them here, but we wanted to clarify our procedure by moving data for a selected example from the SI to the main text for both the traditional and Walden mechanisms.

Please note that our article has also been extended based on the comments of the other referees, and we have added further analyses relating to the kinetics of the Walden step (pages 15-17) and other scientific questions (pages 19-20). Nature Communications' word limit for the main text is 5,000 words, and we have almost fully utilized this limit in the revised version of our manuscript.

Upon preparation for resubmission of the manuscript, I would suggest considering the following aspects :

1) A descriptor based approach might provide a help in interpreting the computational results but a solid link to experimental observables or at least a thorough comparison to experimental observations would help to underline the relevance of computed relative energies for non-theoreticians who work on the actual materials. This would also help to judge the plausibility of the mechanistic hypothesis derived from simulations.

Response: We thank the reviewer for the idea to link our calculations with experimental work on this topic. In the following, we discuss this point in detail and provide context to connect theoretical and experimental perspectives.

i) The direct experimental detection of reaction intermediates at electrified interfaces remains a challenge, since surface science techniques developed for the solid-gas interface are difficult to transfer to the solid-liquid interface relevant for electrocatalysis. However, Shao-Horn and coworkers have provided qualitative evidence for the existence of *OO adsorbates on RuO₂ surfaces during the OER using in situ surface X-ray scattering (**Ref. R1-R2**). These studies demonstrate that *OO intermediates are formed on transition-metal oxide surfaces under sufficiently large anodic bias, thus indirectly supporting the proposed Walden mechanism, which largely depends on the formation of the *OO adsorbate as a key reaction intermediate on IrO₂(110).

ii) A quantitative comparison between experimental and theoretical studies in electrocatalysis refers to the construction of free-energy diagrams, as discussed in recent contributions by one of the authors (**Ref. R3-R4**). A comparison of experimentally and theoretically derived free-energy diagrams enables a benchmarking of reaction mechanisms, since the computationally calculated activation free energies can be directly compared with experimentally determined transition-state free energies. Due to fundamental limitations of theoretical approaches to describe proton-coupled electron transfer steps, achieving such a level of agreement between experiment and theory often remains a challenge. For instance, a major difficulty in theoretical investigations of electrocatalytic processes concerns the determination of transition states while maintaining a constant electrode potential during charge transfer.

Although we have determined transition states related to the proposed Walden step in our revised manuscript (we refer to the discussion in our rebuttal letter of referees #2 and #4), we emphasize that these transition states cannot be directly compared with those determined in experimental single crystal studies. The reason for this is that the concerted desorption-adsorption process of the Walden step does not involve charge transfer and therefore this elementary step is unlikely to be one of the rate-determining steps (RDS) in the OER over

IrO₂(110). To unambiguously determine the RDS at a given applied overpotential, all transition states in the reaction mechanism must be identified, and only the transition state corresponding to the actual RDS can be meaningfully compared with experimental activation free energies determined from experimental studies of single-crystalline model electrodes. However, the determination of all transition states in the OER mechanism over IrO₂(110) is far beyond the scope of the present contribution.

To close this gap, there is a semi-quantitative way to map our theoretical framework to experiments. We performed microkinetic simulations (**Ref. R5-R6**) based on the evaluation of the descriptor $G_{\max}(U)$ to estimate the current density (j) as a function of the applied electrode potential for different surface configurations and reaction mechanisms (cf. **Table R1**), which can be compared to the experimentally measured current density by Suntivich and coworkers for an IrO₂(110) model electrode (**Ref. R7-R8**). In our analysis, we use an applied electrode potential of $U = 1.60$ V vs. RHE (reversible hydrogen electrode) to link experimental and theoretical investigations.

Following previous work on the descriptor $G_{\max}(U)$ (**Ref. R5-R6**), this activity measure allows the determination of the current density using equation (1):

$$j(U) = \frac{4k_B T}{h} e \Gamma_{\text{act}} e^{\left[\frac{-(G_{\max}(U) + \beta)}{k_B T} \right]} \quad (1)$$

In equation (1), e , k_B , T , and h denote the elementary charge, Boltzmann constant, absolute temperature in Kelvin, and Planck constant, respectively, while the density of active surface site (cus sites) amounts to $\Gamma_{\text{act}} = 7 \times 10^{14}$ cm⁻² for IrO₂(110). β refers to the Brønsted-Evans-Polanyi (BEP) intercept constant, which links the thermodynamic analysis in terms of $G_{\max}(U)$ with the kinetics related to the transition-state free energy. Based on previous work on the descriptor $G_{\max}(U)$ (**Ref. R5-R6**), $\beta = 0.6$ is chosen in the analysis. **Table R1** summarizes the estimated current densities at $U = 1.60$ V vs. RHE for the traditional and Walden mechanisms for four different surface configurations of IrO₂(110) (cf. **Figure 1b-e** of the main text).

Table R1. Comparison of the theoretically calculated current densities (j) for the traditional and Walden-type OER mechanisms over four different IrO₂(110) surface configurations at $U = 1.60$ V vs. RHE.

Traditional Mechanism	$U = 1.60$ V vs. RHE	Walden-type Mechanism	$U = 1.60$ V vs. RHE
	Current Density (j) [mA/cm ²]		Current Density (j) [mA/cm ²]
Fully hydroxylated IrO₂(110) surface			
Mononuclear	8.9×10^{-12}	Mononuclear-Walden	28.70
Partly hydroxylated IrO₂(110) surface			
Mononuclear	6.4×10^{-10}	Mononuclear-Walden	0.59
Fully oxygen-covered IrO₂(110) surface			
Mononuclear	3.6×10^{-4}	Mononuclear-Walden	19.44
Partly OOH-covered IrO₂(110) surface			
Mononuclear	2.6×10^{-14}	Mononuclear-Walden	2.6×10^{-14}

Considering that the experimental benchmark in the OER over IrO₂(110) based on the work of Kuo et al. (**Ref. R7-R8**) amounts to $j \approx 0.01$ mA/cm² at $U = 1.60$ V vs. RHE, it is obvious that our theoretical model predicts current densities following the Walden-type pathway for the fully hydroxylated, partially hydroxylated and fully oxygen-covered IrO₂(110) surface that are

in the same order of magnitude as the experiments. In contrast, there is a strong difference in current density for the Walden mechanism over the partly OOH-covered IrO₂(110) surface as well as for all traditional mechanisms over the different IrO₂(110) surface configurations with respect to the experimental benchmark. Therefore, we conclude that the IrO₂(110) surface is likely hydroxylated or covered with oxygen adsorbates under typical OER conditions, and our comparison with the experimental data further suggests the prevalence of Walden-type pathways over conventional OER mechanisms.

Changes:

i) We have added the following discussion to pages 19-20 of the main text:

*To further validate the proposed Walden-type pathways in the OER over IrO₂(110), we compare the computed free-energy diagrams with experiments at the single crystal level. Although we have determined transition states related to the proposed Walden step (cf. **Figure 5**), we emphasize that these transition states cannot be directly compared with those determined in experimental single crystal studies. The reason for this is that the concerted desorption-adsorption process of the Walden step does not involve charge transfer and therefore this elementary step is unlikely to be one of the rate-determining steps (RDS) in the OER over IrO₂(110). Therefore, we perform microkinetic simulations based on the evaluation of the descriptor $G_{\max}(U)$ to estimate the current density (j) as a function of the applied electrode potential (U) for different surface configurations and reaction mechanisms. We refer to section 16 of the SI for a detailed overview of this procedure. A comparison with the experimental benchmark in the OER over IrO₂(110) based on the work of Kuo et al. reveals that our theoretical model predicts current densities following the Walden-type pathway for the fully hydroxylated, partially hydroxylated and fully oxygen-covered IrO₂(110) surface that are in the same order of magnitude as the experiments. In contrast, there is a strong difference in current density for the Walden mechanism over the partly OOH-covered IrO₂(110) surface as well as for all traditional mechanisms over the different IrO₂(110) surface configurations with respect to the experimental benchmark. Therefore, we conclude that the IrO₂(110) surface is likely hydroxylated or covered with oxygen adsorbates under typical OER conditions, and our comparison with the experimental data further suggests the prevalence of Walden-type pathways over conventional OER mechanisms.*

ii) We have added the above comparison with experimental studies in a new section of the SI: section 16 "Comparison with experimental data ". Changes are made on pages S48-S49 of the SI.

2) While the authors use the computational Pourbaix diagram scheme to determine the surface termination depending on pH and potential, the results seem a bit counterintuitive and contradict other published work:

Lee et al. [<https://doi.org/10.1038/s41467-022-30838-y>] presented a DFT simulation including AIMD studies in which they report a computational Pourbaix diagram for IrO₂, Wang et al. also reported computational Pourbaix diagrams for IrO₂ using DFT, the "SCAN" functional and an approach within the Materials Project [<https://doi.org/10.1038/s41524-020-00430-3>]. Flores et al. even used an active learning approach for the discovery of stable IrO_x polymorphs and present a revised Pourbaix diagram for Ir-H₂O system [<https://dx.doi.org/10.1021/acs.chemmater.0c01894>].

Bhattacharyya used a cluster approach to study the protonation state of the interface and discuss computational Pourbaix diagrams for their model system

[<https://dx.doi.org/10.1021/acs.jpcc.0c10092>]. Papers by Gonzalez, Solans-Monfort et al. nicely demonstrate how DFT simulations can be used to identify the correct surface termination under acidic conditions and high potentials (<https://doi.org/10.1016/j.jcat.2021.02.026> and <https://doi.org/10.1039/D1NR03592D>). Dang et al. (Nat. Comm. 2012 doi.org/10.1038/s41467-021-26336-2) use schemes to assess the surface structure and termination (see also Klyukin et al. 2018 JPCC)

Comparing the work presented here, from Fig. 1 a) it appears, that all states of the model the authors use occur potential dependent, but each one is pH independent in the range of pH 0-4. Compared to most other computational (and experimental !) Pourbaix diagrams published (see above) this is very rarely the case. Note that the references 44-47 given by the authors are either not IrOx or their own previous studies on different materials. Maybe the authors could compare their results to other similar studies and explain whether the strange behavior they find is physical or an artifact of their model which only allows for simultaneous de-protonation and oxidation of centers in the interface. This would be crucial, as all results are based on the corresponding surface termination the reaction is studied on.

Response: We thank the referee for pointing out that the computational Pourbaix diagram requires further discussion. We are aware that the active phase of IrO₂(110) under OER conditions is controversially discussed in the community (**Ref. R9-R24**). The choice of a different surface configuration can lead to different mechanistic claims and limiting steps, as shown by the example of traditional mechanisms in our manuscript (cf. **Table 1-4** of the main text).

In order to decouple our analysis from the energetically most favorable surface configuration of IrO₂(110) under OER conditions, which can vary based on the chosen computational approach (exchange correlation functional, description of solvation, usage of a canonical or a grand canonical formalism, size of unit cell, among others), we have considered all relevant structure configurations in the potential range of the OER, ranging from a fully hydroxylated to a partly OOH-covered IrO₂ surface (cf. **Figure 1b-e** of the main text). We have further clarified this fact in the revised version of our manuscript.

Although we have already provided reasoning for our approach, we gladly make use of the opportunity to compare our computational Pourbaix diagram to previous studies on the topic. **Table R2** summarizes our extensive literature survey, in which we compiled the thermodynamically most stable surface configuration at $U = 1.53$ V vs. RHE depending on the exchange-correlation functional and the description of solvation used in the DFT framework. As shown in the overview, different surface phases ranging from a partially hydroxylated to a partially OOH-terminated surface are energetically favored at $U = 1.53$ V vs. RHE, and the full range of possible surface configurations is considered in the mechanistic analysis presented in this work.

Table R2. Summary of the thermodynamically stable surfaces of IrO₂(110) at $U = 1.53$ V vs. RHE as reported in previous studies (**Ref. R9-R20**), together with the computational methods and solvation models used. The inclusion of “vdW” indicates that dispersion interactions were considered in the computational setup.

	Thermodynamically stable surface @ $U = 1.53$ V vs. RHE	Exchange correlation functional	Description of solvation
Ref. R9	Partly OOH-terminated	RPBE	No
Ref. R10	Partly OOH-terminated	BEEF-vdW	No
Ref. R11	Partly OOH-terminated	RPBE	Explicit
Ref. R12	Partly OOH-terminated	RPBE + vdW	Explicit

Ref. R13	Fully Oxygen-Covered	PBE + vdW	Implicit
Ref. R14	Fully Oxygen-Covered	PBE	No
Ref. R15	Fully Oxygen-Covered	PBE + vdW	Implicit
Ref. R16	Partly Hydroxylated	PBE + vdW	Implicit
Ref. R17	Fully Oxygen-Covered	PBE + vdW	Implicit
Ref. R18	Partly Hydroxylated	PBE + vdW	Implicit & Explicit
Ref. R19	Fully Oxygen-Covered	PBE + vdW	Implicit
Ref. R20 (*)	Fully Oxygen-Covered	PBE + vdW	No
This Work	Partly Hydroxylated	PBE + vdW	Implicit

*The applied electrode potential in this work is 1.20 V vs. RHE instead of 1.53 V vs. RHE.

In the following, we comment on the various references highlighted by the referee in their report:

Lee et al. (**Ref. R25**) present a DFT-derived Pourbaix diagram, illustrating the stability regions of various iridium oxides and their potassium-intercalated phases at different electrode potential and pH values. This diagram provides insights into the thermodynamic stability of rutile IrO₂ with respect to other Ir-based phases under different electrochemical conditions. While this information is valuable for understanding the broader electrochemical behavior of iridium oxides, our current study focuses explicitly on rutile IrO₂(110) under the anodic conditions of the OER. Therefore, the phase diagram by Lee et al. cannot be used for comparison in **Table R2** with previous studies in the literature.

The study by Wang et al. (**Ref. R26**) has similar scope compared to that of Lee et al. (**Ref. R25**): the stability of various iridium oxide phases is compared at different electrode potential and pH values. Therefore, the phase diagram by Wang et al. cannot be used for comparison in **Table R2** with previous studies in the literature.

Flores et al. (**Ref. R27**) employed an active-learning algorithm to identify stable iridium oxide polymorphs, focusing on bulk phase stability. Their results do not directly relate to our study, which focuses on the surface configuration of rutile IrO₂(110), and therefore we do not list this work in **Table R2** for comparison.

Bhattacharyya et al. (**Ref. R16**) investigated the surface structure of an iridium cluster model under OER conditions. The authors demonstrated that the bridge sites (μ_2) are predominantly oxygen-covered, while the top sites (μ_1) are hydroxylated, resulting in a partly hydroxylated IrO₂ surface, which coincides with the Pourbaix diagram presented in this work (cf. **Table 2**).

González et al. (**Ref. R17**) performed DFT calculations to investigate various terminations of IrO₂ surfaces. Their study indicates that for the (110) facet at a potential of $U = 1.53$ V vs. RHE, the surface is completely covered with oxygen. This finding agrees with the study by Binniger et al. (**Ref. R15**), who also concluded that an oxygen-covered surface is stable under OER conditions (cf. **Table 2**).

The study by González et al. (**Ref. R28**) employs DFT calculations and ab initio molecular dynamics (AIMD) simulations to investigate how surface morphology influences water adsorption and dissociation on IrO₂ nanoparticles. Their work focuses on different nanoparticle sizes and investigates the role of metal coordination and hydrogen bonding in determining water adsorption energies and preferred adsorption structures. Although this study provides

valuable insights into the interaction between water molecules and IrO₂ surfaces, it is beyond the scope of the present work, so **Ref. 28** is not listed in **Table R2**.

Dang et al. (**Ref. R29**) uses a combination of experimental techniques and DFT calculations to explore the structure and reactivity of a 1T-IrO₂(0001) surface. However, this study does not provide a Pourbaix diagram for the IrO₂(110) facet, and its primary focus is on the 1T-IrO₂(0001) surface, which is oxygen terminated at $U = 1.53$ V vs. RHE. Therefore, this reference is not listed in **Table R2**.

Klyukin et al. (**Ref. R20**) performed a systematic thermodynamic analysis using DFT calculations to investigate the surface configuration of low-index iridium surfaces or IrO₂(110) in an electrochemical environment. Their results indicate that at potentials above 0.62 V vs. RHE, the IrO₂(110) surface is covered with hydroxyl (OH*) groups. At potentials above 1.20 V vs. RHE, the adsorption of oxygen (O*) on -top of Ir atoms is energetically favored. Therefore, we could extrapolate that at higher potentials, such as 1.53 V vs. RHE, the IrO₂(110) surface is likely covered with oxygen adsorbates, which is consistent with the results of other studies (**Ref. R13-R15, R17**).

In the present contribution, Pourbaix diagrams are constructed using the computational hydrogen electrode (CHE) approach (**Ref. R30**). This method defines the chemical potential of a proton-electron pair as equal to half the chemical potential of hydrogen gas under standard conditions. Consequently, the free-energy change associated with proton-coupled electron transfer steps becomes independent of pH at the RHE scale, leading to horizontal lines in the Pourbaix diagram. Therefore, the pH independence of the surface configurations in the Pourbaix diagram is a direct result of the CHE method and therefore differs from experimental Pourbaix diagrams, where boundary lines typically exhibit a pH dependence. We note that the pH dependence of the boundary lines can be resolved using DFT calculations in a grand canonical framework, as discussed in the literature (**Ref. R31-R32**). Although we have used a grand canonical approach in the present work to benchmark the application of the CHE method for the modeling of mechanistic pathways (see also our response to reviewer #4 on the application of grand canonical DFT in VASP), we emphasize that the calculation of all surface configurations and intermediate states using the more computationally intensive grand canonical scheme is beyond the scope of the present work. The actual pH dependence of the surface configurations in the Pourbaix diagram under acidic conditions does not affect the presented mechanistic picture of the OER over IrO₂(110), and the derived conclusions regarding the prevalence of Walden-like pathways are robust due to the benchmark of the CHE method.

Changes:

i) We have added the above comparison with other theoretical studies in a new section of the SI: section 13 “Thermodynamically stable surface of IrO₂(110) at $U = 1.53$ V vs. RHE”. Changes are made on **pages S41-S42** of the SI.

ii) We have added the following statement on **page 3** to the main text of our work:

To underpin this point, we have compiled the data from previous DFT studies in Table S33 (cf. SI, section 13): the thermodynamically stable surface structure of IrO₂(110) depends on the exchange correlation function and the solvation description used in the analysis.

iii) We have clarified the Pourbaix approach and the horizontal lines in the main text on page 4 of the main text:

Please note that the pH independence of the surface configurations in the Pourbaix diagram is a direct result of the CHE method and therefore differs from experimental Pourbaix diagrams, where boundary lines typically exhibit a pH dependence. We note that the pH dependence of the boundary lines can be resolved using DFT calculations in a grand canonical framework, which we have used herein to benchmark the application of the CHE method for the modeling of mechanistic pathways (cf. SI, section 6.7).

iii) We have added the following sentence to page 5 of the main text:

The observed surface configurations are consistent with previous DFT-based studies on IrO₂(110) (cf. SI, section 13).

3) The authors apply periodic boundary DFT calculations for the simulation of electrochemical reactions and mostly use a computational hydrogen electrode approach with some corrections for the inclusion of the electrochemical potential. This only allows to describe PCET steps and any chemical or electrochemical step in the mechanism is determined prior to the calculation. Solvent effects are included implicitly, so there is no direct interaction of the reaction center with solvent molecules. While this is a fairly common way of describing electrochemistry, there are several strategies that go beyond the implicit inclusion of effects like pH and solvation of the rigid CHE. Here it would be nice if the authors could at least acknowledge/cite some of the recent computational developments that go beyond their approximations, as especially the "conflicting results of previous theoretical studies" can be understood in the light of the shortcomings of common approximations (implicit solvation in this case).

There are plenty of examples in the literature - Yoo and Wippermann et al. present an AIMD scheme with explicit solvation and inclusion of the electrochemical potential for electrochemical reactions (<https://doi.org/10.1038/s41524-021-00529-1>). Bhattacharyya et al. describe the explicit inclusion of potential and pH in DFT models of the OER on IrO_x nanoparticles (<https://doi.org/10.1021/acs.jpcc.0c10092>). Other studies include explicit water molecules (Gonzalez et al. J.Catal. 2022 doi.org/10.1016/j.jcat.2022.05.023, Nanoscale 2021 doi.org/10.1039/D1NR03592D) For an insightful discussion of the issue for RuO₂ and IrO₂, see also Liberto et al. JPCC 2023 (doi.org/10.1021/acs.jpcc.3c02733). This also covers the topic of how hydrogen bonding networks on transition metal oxides influences the relative stability of intermediates.

In all of these cases, new phenomena can be discussed and more detailed information about the physics and chemistry in the interface can be obtained. It would be nice if the authors could put their approximations into perspective with other approaches which are currently developed. Note that this should not only serve the purpose to give a thorough overview over the state-of-the-art in theory, but also allow the reader to assess possible sources or errors in the description.

Response: We thank the referee for pointing out the importance of going beyond common approaches when describing electrochemical processes by electronic structure theory calculations. While we must admit that the application of the outlined more sophisticated approaches (Ref. R16, R28, R33-35) is beyond the scope of the present contribution, since they would not enable us to obtain such a wealth of data on the elementary steps and reaction mechanisms of the OER over IrO₂(110), we are aware that the consideration of improved schemes for describing the energetics could change the obtained estimates of free energies and electrocatalytic activity to some extent. This fact is acknowledged in the revised version of our manuscript.

Change: We have added a brief discussion to the main text of our revised manuscript, outlining recent advances in the theoretical description of proton-coupled electron transfer steps beyond conventional models. Changes have been made on page 19:

We note that the focus of the present work is on elucidating the elementary steps of the OER over IrO₂(110) and the mechanistic diversity considered requires the application of a community-standardized approach with respect to the application of the CHE approach, with some corrections to account for the applied electrode potential and solvation. Despite this, we would like to emphasize that there are approaches in the literature that go beyond conventional schemes to describe proton-coupled electron transfer steps and often rely on computationally intensive ab initio molecular dynamics simulations. Although the consideration of improved schemes for describing the elementary steps of the OER could change the obtained estimates of free energies and electrocatalytic activity to some extent, the provided analysis of the OER over IrO₂(110) is robust due to the large difference of the $G_{max}(U)$ descriptor between traditional and Walden pathways. Going beyond the conventional scheme of the CHE approach is nevertheless desirable as it could clarify the effects of hydrogen bonding networks on the relative stability of intermediates adsorbed on transition metal oxides.

4) As mentioned above, the authors have moved a considerable amount of results into the SI and opted to keep five tables in the manuscript. I would strongly recommend to move details like reaction paths / structures and plots of the relative free energies into the main manuscript and move numerical data to the SI. Note that there are several recent works (for example from the Goddard group on OER in alkaline conditions on materials like CoOx and NiOx), in which microkinetic models have been used to assess complex reaction networks on different surface sites. With the wealth of data the authors present, it would be a pity not to be able to directly compare the energetics of the different mechanisms and discuss bottlenecks and common intermediates in more detail. Note that an unfavorable mechanism might still exhibit a low energy intermediate that indicates possible catalyst poisoning or degradation.

In summary - the work presented here is quite extensive and a lot of interesting and novel results are outlined. However, I don't think a communication would do the work justice. I think the work would rather be suited for a journal like Nature Catalysis or Nature Chemistry. Then on the other hand, I'm not sure whether it has so much impact to scientists outside the computational catalysis community, as it is a purely computational work mostly addressed to theoreticians.

The authors have chosen to go for Nature Communications and I respect this, so my overall recommendation is accept with major revision with several aspects that need thorough revision (especially with respect to the literature discussed).

Response: We thank the referee for their enthusiastic referee report and valuable comments to improve our manuscript for publication in Nature Communications. We agree with the referee that our work – in the context of the computational community – may not receive the same attention in journals such as Nature Catalysis or Nature Chemistry, where manuscripts are often based on experimental studies supported by theory. It is precisely for this reason that we have submitted our article to Nature Communications and hope that the revised version of our manuscript, which we have expanded into a full article, will meet this reviewer's expectations for publication in this journal.

Ref. R1. Rao, R. R. et al. Towards identifying the active sites on RuO₂(110) in catalyzing oxygen evolution. *Energy Environ. Sci.* **10**, 2626–2637 (2017).

Ref. R2. Rao, R. R. et al. Operando identification of site-dependent water oxidation activity on ruthenium dioxide single-crystal surfaces. *Nat. Catal.* **3**, 516–525 (2020).

Ref. R3. Exner, K. S., Sohrabnejad-Eskan, I. & Over, H. A universal approach to determine the free energy diagram of an electrocatalytic reaction. *ACS Catal.* **8**, 1864–1879 (2018).

Ref. R4. Exner, K. S. & Over, H. Beyond the rate-determining step in the oxygen evolution reaction over a single-crystalline IrO₂(110) model electrode: kinetic scaling relations. *ACS Catal.* **9**, 6755–6765 (2019).

Ref. R5. Exner, K. S. A universal descriptor for the screening of electrode materials for multiple-electron processes: Beyond the thermodynamic overpotential. *ACS Catal.* **10**, 12607–12617 (2020).

Ref. R6. Razzaq, S. & Exner, K. S. Materials screening by the descriptor $G_{\max}(\eta)$: The free-energy span model in electrocatalysis. *ACS Catal.* **13**, 1740–1758 (2023).

Ref. R7. Kuo, D.-Y. et al. Influence of surface adsorption on the oxygen evolution reaction on IrO₂(110). *J. Am. Chem. Soc.* **139**, 3473–3479 (2017).

Ref. R8. Kuo, D.-Y. et al. Measurements of oxygen electroadsorption energies and oxygen evolution reaction on RuO₂(110): A discussion of the Sabatier principle and its role in electrocatalysis. *J. Am. Chem. Soc.* **140**, 17597–17605 (2018).

Ref. R9. Hansen, H. A. et al. Electrochemical chlorine evolution at rutile oxide (110) surfaces. *Phys. Chem. Chem. Phys.* **12**, 283–290 (2010).

Ref. R10. Sumaria, V., Krishnamurthy, D. & Viswanathan, V. Quantifying confidence in DFT predicted surface Pourbaix diagrams and associated reaction pathways for chlorine evolution. *ACS Catal.* **8**, 9034–9042 (2018).

Ref. R11. Gauthier, J. A., Dickens, C. F., Chen, L. D., Doyle, A. D. & Nørskov, J. K. Solvation effects for oxygen evolution reaction catalysis on IrO₂(110). *J. Phys. Chem. C* **121**, 11455–11463 (2017).

Ref. R12. Briquet, L. G., Sarwar, M., Mugo, J., Jones, G. & Calle-Vallejo, F. A new type of scaling relations to assess the accuracy of computational predictions of catalytic activities applied to the oxygen evolution reaction. *ChemCatChem* **9**, 1261–1268 (2017).

Ref. R13. Ping, Y., Nielsen, R. J. & Goddard, W. A. The reaction mechanism with free energy barriers at constant potentials for the oxygen evolution reaction at the IrO₂ (110) surface. *J. Am. Chem. Soc.* **139**, 149–155 (2016).

Ref. R14. Ha, M.-A. & Larsen, R. E. Multiple reaction pathways for the oxygen evolution reaction may contribute to IrO₂ (110)'s high activity. *J. Electrochem. Soc.* **168**, 024506 (2021).

Ref. R15. Binniger, T. & Doublet, M.-L. The Ir–OOOO–Ir transition state and the mechanism of the oxygen evolution reaction on IrO₂ (110). *Energy Environ. Sci.* **15**, 2519–2528 (2022).

Ref. R16. Bhattacharyya, K., Poidevin, C. & Auer, A. A. Structure and reactivity of iron nanoparticles for the oxygen evolution reaction in electrocatalysis: An electronic structure theory study. *J Phys. Chem. C* **125**, 4379–4390 (2021).

Ref. R17. González, D., Heras-Domingo, J., Sodupe, M., Rodríguez-Santiago, L. & Solans-Monfort, X. Importance of the oxyl character on the IrO₂ surface dependent catalytic activity for the oxygen evolution reaction. *J. Catal.* **396**, 192–201 (2021).

Ref. R18. Mou, T., Bushiri, D. A., Esposito, D. V., Chen, J. G. & Liu, P. Rationalizing acidic oxygen evolution reaction over IrO₂: Essential role of hydronium cation. *Angew. Chem.* **63**, e202409526 (2024).

Ref. R19. Kwon, S. et al. Facet-dependent oxygen evolution reaction activity of IrO₂ from quantum mechanics and experiments. *J. Am. Chem. Soc.* **146**, 11719–11725 (2024).

Ref. R20. Klyukin, K., Zagalskaya, A. & Alexandrov, V. Ab initio thermodynamics of Iridium surface oxidation and oxygen evolution reaction. *J. Phys. Chem. C* **122**, 29350–29358 (2018).

Ref. R21. Hansen, H. A., Rossmeisl, J. & Nørskov, J. K. Surface Pourbaix diagrams and oxygen reduction activity of Pt, Ag and Ni(111) surfaces studied by DFT. *Phys. Chem. Chem. Phys.* **10**, 3722 (2008).

Ref. R22. Eslamibidgoli, M. J., Huang, J., Kowalski, P. M., Eikerling, M. H. & Groß, A. Deprotonation and cation adsorption on the NiOOH/water interface: A grand-canonical first-principles investigation. *Electrochim. Acta* **398**, 139253 (2021).

Ref. R23. Exner, K. S., Anton, J., Jacob, T. & Over, H. Chlorine evolution reaction on RuO₂(110): Ab initio atomistic thermodynamics study - Pourbaix diagrams. *Electrochim. Acta* **120**, 460–466 (2014).

Ref. R24. Zagalskaya, A., Chaudhary, P. & Alexandrov, V. Corrosion of electrochemical energy materials: stability analyses beyond Pourbaix diagrams. *J. Phys. Chem. C* **127**, 14587–14598 (2023).

Ref. R25. Lee, S., Lee, Y.-J., Lee, G. & Soon, A. Activated chemical bonds in nanoporous and amorphous iridium oxides favor low overpotential for oxygen evolution reaction. *Nat. Commun.* **13**, 3171 (2022).

Ref. R26. Wang, Z., Guo, X., Montoya, J. & Nørskov, J. K. Predicting aqueous stability of solid with computed Pourbaix diagram using scan functional. *npj Comput. Mater.* **6**, 160 (2020).

Ref. R27. Flores, R. A. et al. Active learning accelerated discovery of stable iridium oxide polymorphs for the oxygen evolution reaction. *Chem. Mater.* **32**, 5854–5863 (2020).

Ref. R28. González, D., Sodupe, M., Rodríguez-Santiago, L. & Solans-Monfort, X. Surface morphology controls water dissociation on hydrated IrO₂ nanoparticles. *Nanoscale* **13**, 14480–14489 (2021).

Ref. R29. Dang, Q. et al. Iridium metallene oxide for acidic oxygen evolution catalysis. *Nat. Commun.* **12**, 6007 (2021).

Ref. R30. Nørskov, J. K. et al. Origin of the overpotential for oxygen reduction at a fuel-cell cathode. *J. Phys. Chem. B* **108**, 17886–17892 (2004).

Ref. R31. Groß, A. Reversible vs standard hydrogen electrode scale in interfacial electrochemistry from a theoretician’s atomistic point of view. *J Phys. Chem. C* **126**, 11439–11446 (2022).

Ref. R32. Groß, A. Grand-canonical approaches to understand structures and processes at electrochemical interfaces from an atomistic perspective. *Curr. Opin. Electrochem.* **27**, 100684 (2021).

Ref. R33. Yoo, S.-H. *et al.* Finite-size correction for slab supercell calculations of materials with spontaneous polarization. *npj Comput. Mater.* **7**, 58 (2021).

Ref. R34. Di Liberto, G., Pacchioni, G., Shao-Horn, Y. & Giordano, L. Role of water solvation on the key intermediates catalyzing oxygen evolution on RuO₂. *J. Phys. Chem. C* **127**, 10127–10133 (2023).

Ref. R35. González, D., Sodupe, M., Rodríguez-Santiago, L. & Solans-Monfort, X. Metal coordination determines the catalytic activity of IrO₂ nanoparticles for the oxygen evolution reaction. *J. Catal.* **412**, 78–86 (2022).

Reviewer #2

The authors report a systematic exploration of electro-catalytic pathways for the conversion of water into oxygen at the IrO₂ surface for four alternative surface terminations that are consistent with operating conditions. The results clarify some of the inconsistencies on the OER mechanism reported in the literature by different computational studies. Moreover the authors seem to identify a previously-unexplored catalytic step as the most relevant reaction mechanism for OER on IrO₂. The pathways identified in the first screening of the different IrO₂ surfaces all seem to involve the freeing of a 'cus' site as their limiting steps. Thus, the so-called Walden mechanism, a concerted adsorption/release step, provides a more thermodynamically favorable mechanism. As the concerted mechanism involves a solvent molecule, I assume there are no statistical drawback or kinetic limitations with respect to the separate release and adsorption, although the authors may want to expand on this aspect. The presentation of the manuscript is very clear and the results are produced following state-of-the-art approaches, with enough details to make them reproducible.

Response: We thank the referee for their support of our manuscript for publication in Nature Communications.

i) Statistical assessment of reported Walden-type mechanisms: we have evaluated the Walden mechanisms for four different surface configurations of IrO₂(110) (cf. **Figure 1b-e** of the main text). Given that the Walden pathways are energetically preferred over the traditional reaction mechanism in all four cases (cf. **Table 5** of the main text), we believe that our analysis provides a reasonable statistics for the occurrence of the Walden-type mechanisms.

ii) Kinetic limitations: we appreciate the reviewer's comments on the assessment of the kinetics of the Walden step, in which O₂ and H₂O are replaced in a concerted manner. Taking the advice of this referee in conjunction with the report of referee #4 into account, we evaluated the activation free energy for the Walden step using the partly hydroxylated IrO₂(110) surface (cf. **Figure 1c** in the main text) as a representative example.

Please note that this analysis is also motivated by the fact that in a recent contribution, a chemical Walden step was proposed to be the rate-determining step (RDS) in the oxygen reduction reaction (ORR) on Fe–N–C SAC (single-atom catalyst) (**Ref. R1**). Following the approach reported by Ping and Goddard (**Ref. R2**), we initially performed nudge elastic band (NEB) calculations to map the reaction pathway and locate an approximate transition state. Subsequently, the dimer method was employed to refine and identify the precise transition state structure. The identified transition state was validated by frequency analysis. All these calculations were performed including spin polarization and an implicit solvation model (VASPsol) to determine the transition state of a chemical Walden step on the IrO₂(110) surface, where water replaces O₂ (cf. equation (R1)):

Please note that we rely on a canonical framework to calculate the transition state of equation (R1) to ensure consistency with the thermodynamic analysis of the reaction intermediates' free energies. This is further backed up by the fact that Ping and Goddard reported that the difference between constant potential and constant charge calculations for IrO₂(110) in their study did not exceed 0.10 eV (**Ref. R2**), which is considered to be sufficiently accurate for the present qualitative assessment of the Walden transition state. While **Figure R1** illustrates the initial, transition, and final states for the process of equation (R1), **Table R1** summarizes the energetics of the states involved.

Figure R1. Initial (IS), transition (TS), and final (FS) states for the water-mediated O₂ desorption (cf. equation (R1)) from the partly hydroxylated IrO₂(110) surface. All bond lengths are given in Å.

Table R1. Energetics of the initial (IS), transition (TS), and final (FS) states of water-mediated O₂ desorption (cf. equation (R1)) from the partly hydroxylated IrO₂(110) surface to derive the activation free energy, ΔG^\ddagger , for this process.

	E_{DFT} [eV]	ZPE [eV]	TS [eV]
IS	-449.43	1.16	0.32
TS	-448.31	1.15	0.38
FS	-449.47	1.14	0.31
G_{IS} [eV]		-448.59	
G_{TS} [eV]		-447.54	
ΔG^\ddagger [eV]		1.05	

We determine an activation free energy of 1.05 eV for the Walden step of equation (R1). This activation free energy is compared with the kinetics of O₂ desorption without the involvement of a water molecule (cf. equation (R2)):

For the process in equation (R2), the initial, transition, and final states are shown in **Figure R2**, while **Table R2** gives the energetics of the states involved.

Figure R2. Initial (IS), transition (TS), and final (FS) states for O₂ desorption (cf. equation (R2)) from the partly hydroxylated IrO₂(110) surface. All bond lengths are given in Å.

Table R2. Energetics of the initial (IS), transition (TS), and final (FS) states of O₂ desorption (cf. equation (R2)) from the partly hydroxylated IrO₂(110) surface to derive the activation free energy, ΔG^\ddagger , for this process.

	E_{DFT} [eV]	ZPE [eV]	TS [eV]
IS	-434.66	0.52	0.20
TS	-433.38	0.47	0.31
FS	-433.34	0.48	0.24

G_{IS} [eV]	-434.34
G_{TS} [eV]	-433.22
$\Delta G^\#$ [eV]	1.12

Our analysis reveals that the desorption barrier of O_2 is on the order of about 1 eV, and the water-assisted route in the framework of a Walden step (cf. equation (R1)) has a slightly lower activation barrier than the desorption of O_2 without the involvement of a water molecule (cf. equation (R2)). To incorporate these activation free energies into the free-energy landscape of the OER on $IrO_2(110)$, we extend our thermodynamic analysis by additionally calculating the free energy of the $*_{cus}-(OH_2)_{ot}$ adsorbate, which was previously not part of the catalytic cycle.

For the Walden-type mechanism, we investigate the following elementary steps:

The corresponding free-energy diagram for equations (R3) – (R7) is shown in **Figure R3**.

Figure R3. Free-energy diagram of the OER over the partly hydroxylated $IrO_2(110)$ surface according to the mechanistic description in equations (R3) – (R7) at $U = 1.53$ V vs. RHE. The calculated activation free energy for the water-assisted desorption of the O_2 molecule (cf. **Table R1**) has been added to the figure.

With respect to the traditional pathways, we reformulate the mononuclear mechanism by including the chemical desorption of O_2 as an additional step into the analysis:

The corresponding free-energy diagram for equations (R8) – (R12) is shown in **Figure R4**.

Figure R4. Free-energy diagram of the OER over the partly hydroxylated IrO₂(110) surface according to the mechanistic description in equations (R8) – (R12) at $U = 1.53$ V vs. RHE. The calculated activation free energy for the desorption of the O₂ molecule (cf. **Table R2**) has been added to the figure.

When comparing the free-energy landscapes of **Figures R3-R4**, it turns out that the inclusion of the additional intermediate states $*_{\text{cus}}\text{-(OH}_2\text{)}_{\text{ot}}$ and $*_{\text{cus}}\text{-OO}_{\text{ot}}$ for the Walden or traditional mononuclear mechanism, respectively leads to an alteration of the activity descriptor $G_{\text{max}}(U = 1.53 \text{ V})$ in the latter case: for the modified mononuclear pathway, $G_{\text{max}}(U = 1.53 \text{ V})$ amounts to 1.06 eV, which is substantially larger than $G_{\text{max}}(U = 1.53 \text{ V}) = 0.22 \text{ eV}$ for the Walden-type mechanism. Given that the sensitivity of the descriptor $G_{\text{max}}(U = 1.53 \text{ V})$ was specified as 0.20 eV (**Ref. R3**), we observe clear evidence for the preference of the Walden mechanism compared to the traditional description. This result is also consistent with the calculated desorption barriers for O₂ of the competing pathways and underlines that the Walden-type description has lower kinetic constraints. Therefore, we conclude that the Walden-type mechanism represents the most favorable mechanistic description of the OER over IrO₂(110). This statement is also supported when the kinetics of product desorption is explicitly considered in the model.

Last but not least, we would like to leave a comment concerning the rate-determining step (RDS). We do not aim to make a statement about water-mediated O₂ desorption as a potential RDS in the OER over IrO₂(110). We emphasize that an unbiased and clear-cut determination of the RDS would require the assessment of all possible transition states for all elementary steps in the catalytic cycle (cf. equations (R3) – (R7)). In addition, it is possible that not just a single elementary step, but several elementary steps contribute to the reaction rate, which can only be described by advanced analytical techniques, such as exploiting the degree of rate control (**Ref. R4**). However, such an analysis goes far beyond the scope of the present

manuscript, which argues that Walden-type mechanisms are energetically preferable to traditional mechanistic descriptions in the OER over IrO₂(110).

Ref. R1. Yu, S., Levell, Z., Jiang, Z., Zhao, X. & Liu, Y. What is the rate-limiting step of oxygen reduction reaction on Fe–N–C catalysts? *J. Am. Chem. Soc.* **145**, 25352–25356 (2023).

Ref. R2. Ping, Y., Nielsen, R. J. & Goddard, W. A. The reaction mechanism with free energy barriers at constant potentials for the oxygen evolution reaction at the IrO₂(110) surface. *J. Am. Chem. Soc.* **139**, 149–155 (2017).

Ref. R3. Exner, K. S. A universal descriptor for the screening of electrode materials for multiple-electron processes: beyond the thermodynamic overpotential. *ACS Catal.* **10**, 12607–12617 (2020).

Ref. R4. Exner, K. S. Standard-state entropies and their impact on the potential-dependent apparent activation energy in electrocatalysis. *J. Energy Chem.* **83**, 247–254 (2023).

Changes:

i) We have added a discussion on the kinetics of O₂ desorption to the main text of our manuscript. For the sake of brevity, we do not copy these changes here, and we refer the reviewer to pages 15-17.

ii) We have added the above discussion on the kinetics to a new section of the SI: section 15 “Kinetics of O₂ desorption”. Changes are made on pages S45-S48 of the SI.

Reviewer #3

In this manuscript Exner and Coworkers made DFT calculations of four different “traditional” OER mechanisms and two with Walden type steps for the OER on IrO₂(110). The elementary reaction steps and mechanisms in the OER are evaluated by constructing free-energy diagrams in conjunction with descriptor-based analyses in the range of the activity measure $G_{\max}(U)$. Their analysis shows that Walden type mechanism is beneficial for OER catalysis since it leads to a reduction in free-energy.

The analysis is thorough and the relationships are clearly described – overall a very good manuscript. I believe that only this type of approach can provide access to reliable values for comparison of energy parameters and provide valuable insights for understanding the processes at the IrO₂ interface in water electrolysis. Therefore, I think this article is very timely and important.

Response: We thank the referee for their support of our manuscript for publication in Nature Communications.

However, I have a few comments that I would like to have addressed by the authors:

Comment 1: This paper is a specific and thorough analysis mechanistic pathways of IrO₂(110) for OER. And indeed, IrO₂ is important to understand for OER, and their DFT calculations is also interesting.

Reading the authors prior publication of potential dependent switching of OER-mechanisms and of their investigation Walden type inversion (<https://doi.org/10.1039/D3MH00047H> and DOI: 10.1002/adv.202305505), I see this publication more as continued work to the older one – What is the key insight here that warrants publication in Nature Comm.? One could consider it in a more specific journal.

Response: We thank the referee for asking about the key findings of our work. Indeed, the solo articles by Exner on the potential-dependent switching of OER mechanisms (**Ref. R1**) and Walden-type inversion (**Ref. R2**) definitely motivated this work; however, the present DFT-based analysis goes far beyond our previous articles:

i) The two previous articles mentioned by the referee do not contain any DFT calculations. Instead, these works rely on the derivation of volcano plots through a data-driven analysis of adsorption free energies and scaling relationships without a clear reference to a catalyst material (see **Ref. R3** for a short perspective in relation to this research area). IrO₂ was not explicitly investigated in these former works. Considering that IrO₂ is a highly active OER catalyst located at the volcano apex, it was only possible to suppose that a switch in the reaction mechanism or Walden-type pathways might be of relevance to IrO₂.

In the present contribution, we demonstrate that the OER over IrO₂(110) is governed by Walden-type mechanisms. Considering that IrO₂ is the industrial state-of-the-art catalyst for OER in acid, this is an important discovery with far-reaching implications for scientists working on OER (*vide infra*).

ii) Several density functional theory (DFT)-based studies have been published on the OER over IrO₂(110), yet a significant mechanistic controversy persists (**Ref. R4-R11**). We believe that our current work solves this controversy by carefully addressing two key aspects: a) the specific surface configurations employed in the computational models, and b) the reaction pathways considered in the mechanistic analysis. Therefore, our work contains an important message for

the theoretical electrocatalysis community and underlines the urgent need for a consistent and comprehensive treatment of surface configurations (adsorbate coverage) and reaction mechanisms in DFT-based studies.

iii) We employ DFT calculations to investigate seven distinct reaction mechanisms for four different surface configurations of IrO₂(110), which goes far beyond the scope of previous DFT-based works of the OER over IrO₂(110) (**Ref. R4-R11**). To the best of our knowledge, no prior research has explored such an extensive range of surface configurations and pathways for the OER on IrO₂. Therefore, we are convinced that our work represents a state-of-the-art contribution to the field, relying exclusively on DFT without the use of machine learning (ML) or artificial intelligence techniques. On the other hand, our first-principles data could provide a solid foundation for future ML-based studies to further expand the scope of the elementary steps and surface chemistry of IrO₂ under anodic polarization.

iv) In the revised version of our manuscript, we have added transition-state calculations to support our mechanistic analysis. Specifically, we verify the transition state for O₂ desorption in both the Walden-type and traditional mechanism. Our findings demonstrate that the Walden-type step exhibits lower kinetic constraints compared to the traditional mechanism. This result provides evidence that goes beyond the thermodynamic picture and supports the occurrence of Walden mechanisms in electrocatalysis, which previously have been largely overlooked in mechanistic studies.

v) In **Figure 5** of the main text, we present a detailed charge analysis of the elementary steps of the traditional and Walden-type mechanisms for OER over IrO₂(110). We introduce the charge span, Q_{\max} , as a new descriptor for the electrocatalytic activity and thus offer a different approach for mechanistic evaluation besides the usual picture of adsorption free energies. The use of Q_{\max} as a measure for the electrocatalytic activity represents a significant advantage by reducing the computational effort compared to conventional analyses based on adsorption free energies. Our results suggest that Q_{\max} could serve as an efficient and rapid screening criterion to identify promising material motifs, not only for OER but also for other electrocatalytic processes. Therefore, we are confident that our manuscript also provides new avenues for material discovery that can be exploited in future studies.

Based on the above explanations, we are confident that our manuscript provides sufficient new insights into the OER over IrO₂(110), which is an important reaction system due to the usage of IrO₂ as OER catalyst in acid, and also provides new directions for computational materials scientists. Therefore, we remain committed that Nature Communications is the right forum to disseminate our results and our carefully conducted study on the analysis of mechanistic pathways of an electrocatalytic process using calculations within the framework of electronic structure theory.

Changes:

i) We have added a discussion on the kinetics of O₂ desorption to the main text of our manuscript. For the sake of brevity, we do not copy these changes here, and we refer the reviewer to **pages 15-17**.

ii) We have added the above discussion on the kinetics to a new section of the SI: section 15 “Kinetics of O₂ desorption”. Changes are made on **pages S45-S48** of the SI.

iii) We have added a discussion of the relationship of our work to previous theoretical studies and experiments on pages 19-20 of the main text, and we have provided further evidence for our work by adding additional analyses to the SI (pages S41-S45 and S48-S49).

Comment 2: Maybe some suggestions could help to improve the manuscript to a broader readership. I believe the introduction could help the reader with a bit more discussing on other OER catalyst. Why was IrO₂(110) chosen? And how do energies change for other IrO₂ sites? Moreover, from a practical point of view, what can I do with the knowledge gained in this work? How can I use it to improve OER catalysts?

Response: We thank the referee for their helpful suggestions to improve the manuscript and make it accessible to a wider audience. A detailed response can be found below.

i) IrO₂(110) was chosen because IrO₂ is the industrial state-of-the-art catalyst for OER in acid.

ii) The (110) facet is the thermodynamically most stable surface facet of IrO₂ and therefore represents a good model system to investigate the OER mechanisms, since this facet is likely also found in IrO₂ nanoparticles or IrO_x catalysts.

iii) Over and coworkers have performed stability experiments for IrO₂(110) at the single crystal level (Ref. R12-R13). They show that the (110) surface does not tend to oxidation despite the anodic reaction conditions of the OER and that catalyst degradation is almost negligible. This is another reason why the (110) facet of IrO₂ is a suitable model system to study the elementary steps of the OER, since our model can be based on the intact rutile structure of IrO₂ without having to consider a reconstructed surface.

iv) According to Figure S1 of the SI, there are two different surface sites on the (110) facet of IrO₂, namely cus and bridge sites. It has been reported in previous works (Ref. R4-R11) that the cus sites are the catalytically active centers for the OER or other surface reactions whereas the bridge sites are mainly spectators.

To further convince the reviewer of the above, we performed additional DFT calculations to investigate the OER at the bridge sites of the IrO₂(110) surface. Please note that we have chosen the fully oxygen-covered IrO₂(110) surface (cf. Figure 1d of the main text) as a representative example. The mononuclear mechanism of the OER on an Ir bridge site (*_{br}) of IrO₂(110) reads:

While Table R3 compiles the free-energy changes for each elementary step at $U = 0$ V vs. RHE and the activity descriptor $G_{max}(U)$ at different applied electrode potentials under OER conditions, the corresponding free-energy diagram is depicted in Figure R4a, with a visual representation of the elementary steps in Figure R4b. While the activity descriptor $G_{max}(U)$ is governed by the span $*_{br}-OH_{br} \rightarrow *_{br}-O_{br} \rightarrow *_{br}-OOH_{br} \rightarrow *_{br} + O_2$ at $U = 1.23$ V vs. RHE, the limiting free-energy span switches to $*_{br}-O_{br} \rightarrow *_{br}-OOH_{br} \rightarrow *_{br} + O_2$ for larger applied overpotentials ($U = 1.53$ V vs. RHE).

Based on the determination of the activity descriptor $G_{\max}(U = 1.53 \text{ V}) = 1.86 \text{ eV}$ for the mononuclear mechanism, which exceeds the value for the Ir cus site ($G_{\max}(U = 1.53 \text{ V}) = 0.13 \text{ eV}$ for the energetically favored mononuclear-Walden mechanism) by more than 1.70 eV, the Ir bridge site can be fairly excluded as an active site in the OER over IrO₂(110).

Table R3. Energetic evaluation of the mononuclear mechanism on the bridge site of the fully oxygen-covered IrO₂(110) surface (cf. **Figure 1d** in the main text) by the framework of the descriptor $G_{\max}(U)$. The table indicates the free-energy changes of each step at $U = 0 \text{ V}$ vs. RHE and $G_{\max}(U)$ values at different applied electrode potentials (U).

ΔG_1 [eV]	ΔG_2 [eV]	ΔG_3 [eV]	ΔG_4 [eV]	$G_{\max}(U)$ [eV]				
				1.23 V	1.33 V	1.43 V	1.53 V	1.63 V
-1.29	1.29	2.00	2.92	2.52	2.26	2.06	1.86	1.66

Figure R4. a) Free-energy diagram for the mononuclear mechanism on the bridge site of the fully oxygen-covered IrO₂(110) surface at 1.23 V and 1.53 V vs. RHE. The reaction intermediates of the mechanistic cycle are labelled on the x-axis. Blue and green solid lines indicate intermediates' free energies at 1.23 V and 1.53 V, respectively. Colored arrows indicate the free-energy span governing $G_{\max}(U)$, with the respective value displayed.

b) Schematic illustration of the mononuclear mechanism, as described in Section 3.1, on the bridge site of the fully oxygen-covered IrO₂(110) surface. Numbers next to the arrows indicate the step sequence, and each structure represents the corresponding reaction intermediate.

Besides the mononuclear mechanism, we have also evaluated the energetics of the mononuclear-Walden pathway at the Ir bridge site. The mechanistic description is given by equations (R17) – (R20):

While **Table R4** compiles the free-energy changes for each elementary step at $U = 0$ V vs. RHE and the activity descriptor $G_{\max}(U)$ at different applied electrode potentials under OER conditions, the corresponding free-energy diagram is depicted in **Figure R5a**, with a visual representation of the elementary steps in **Figure R5b**. While the activity descriptor $G_{\max}(U)$ is governed by the span $*_{\text{br}}\text{-OH}_{\text{br}} \rightarrow *_{\text{br}}\text{-O}_{\text{br}} \rightarrow *_{\text{br}}\text{-OOH}_{\text{br}}$ at $U = 1.23$ V vs. RHE, the limiting free-energy span switches to $*_{\text{br}}\text{-O}_{\text{br}} \rightarrow *_{\text{br}}\text{-OOH}_{\text{br}}$ for larger applied overpotentials ($U = 1.53$ V vs. RHE).

Table R4. Energetic evaluation of the mononuclear-Walden mechanism on the bridge site of the fully oxygen-covered $\text{IrO}_2(110)$ surface (cf. **Figure 1d** in the main text) by the framework of the descriptor $G_{\max}(U)$. The table indicates the free-energy changes of each step at $U = 0$ V vs. RHE and $G_{\max}(U)$ values at different applied electrode potentials (U).

ΔG_1 [eV]	ΔG_2 [eV]	ΔG_3 [eV]	ΔG_4 [eV]	$G_{\max}(U)$ [eV]				
				1.23 V	1.33 V	1.43 V	1.53 V	1.63 V
1.29	2.00	1.21	0.42	0.83	0.67	0.57	0.47	0.37

Figure R5. a) Free-energy diagram for the mononuclear-Walden mechanism on the bridge site of fully oxygen-covered $\text{IrO}_2(110)$ surface at 1.23 V and 1.53 V vs. RHE. The reaction intermediates of the mechanistic cycle are labeled on the x-axis. Blue and green solid lines indicate intermediates' free energies at 1.23 V and 1.53 V, respectively. Colored arrows indicate the free-energy span governing $G_{\max}(U)$, with the respective value displayed. **b)** Schematic illustration of the mononuclear-Walden mechanism, as described in Section 8.1, on the bridge site of the fully oxygen-covered $\text{IrO}_2(110)$ surface. Numbers next to the arrows indicate the step sequence, and each structure represents the corresponding reaction intermediate.

We note that the mononuclear-Walden mechanism reveals a significantly lower $G_{\max}(U)$ value compared to the mononuclear mechanism at the Ir bridge site. This result is in qualitative agreement to the discussion of the Walden-type pathways at the Ir cus site in the main text of our work. This underlines the scope of Walden-like pathways for electrocatalytic processes, as concerted desorption-adsorption processes can efficiently reduce thermodynamic constraints in proton-coupled electron transfer steps.

Despite the reduced $G_{\max}(U)$ value, the electrocatalytic activity of the Ir bridge site is still sufficiently lower ($G_{\max}(U = 1.53 \text{ V}) = 0.47 \text{ eV}$) compared to the Ir cus site ($G_{\max}(U = 1.53 \text{ V})$

= 0.13 eV). Therefore, we conclude that the bridge sites are not the active sites in the OER over IrO₂(110), which is in line with previous works on the topic (**Ref. R4-R11**).

v) The referee has raised an interesting question whether the knowledge gained in this work can be used to improve OER catalysts. There is a strong belief that identifying the reaction mechanism and the kinetic bottleneck (often referred to as the rate-limiting step) can contribute to the development of improved OER catalysts. While our research is still at the fundamental level with the investigation of a state-of-the-art catalyst for OER, the insights gained in this study (particularly with regard to the reported Walden-type mechanisms) can be exploited in future research on materials discovery using methods from electronic structure theory and artificial intelligence. This may potentially lead to the identification of improved OER catalysts considering that all relevant elementary steps — such as Walden-type processes — or advanced descriptors — such as the charge span Q_{\max} — are considered in the analysis.

Changes:

i) We have added a short statement why IrO₂(110) was chosen in this work on **page 2** of the main text:

In the present work, we reinvestigate the OER on the (110) facet of IrO₂ as this is a suitable model system to better understand the factors limiting the OER at the atomic level.

ii) We have added the following sentence to **page 4** of the main text:

To clearly rule out the Ir bridge sites as active sites in the OER, we have compared their activity to that of the Ir cus sites in section 14 of the SI.

iii) We have added a discussion of the elementary steps of the OER over the Ir bridge site in the supplemental. Our detailed discussion can be found in the new section 14 of the SI: “OER on the bridge sites of IrO₂(110)”. Changes are made on **pages S42-S45** of the SI.

iv) We have added the following sentence to **page 21** of the main text:

These findings can be used in future research on materials discovery using electronic structure theory and artificial intelligence methods to identify improved OER catalysts.

Comment 3: Figure 2 gives a great overview of the different OER mechanisms. Only one thing doesn't seem right to me. Between step (1) and step (2), IrO₂(110) adsorb water and not release it, or am I missing something?

Response: The referee is right. Thank you for noticing this error.

Change: We have corrected Figure 2 in the revised version of our work. An updated version of the figure can be found on **page 6** in the main text or below (**Figure R6**).

Figure R6. Investigated OER mechanisms on $\text{IrO}_2(110)$ taking the fully oxygen-covered surface (cf. **Figure 1d**) as a representative example. Colors of the various mechanistic pathways are indicated in the top right corner, and surface structure details are given in the bottom left corner. Note that the same steps have also been studied over the other surface motifs depicted in **Figure 1b-e** of the main text.

Ref. R1. Exner, K. S. On the mechanistic complexity of oxygen evolution: potential-dependent switching of the mechanism at the volcano apex. *Mater. Horiz.* **10**, 2086–2095 (2023).

Ref. R2. Exner, K. S. Importance of the walden inversion for the activity volcano plot of oxygen evolution. *Adv. Sci.* **10**, 2305505 (2023).

Ref. R3. Exner, K. S. How data-driven approaches advance the search for materials relevant to energy conversion and storage. *Mater. Today Energy.* **36**, 101364 (2023).

Ref. R4. Opalka, D., Scheurer, C. & Reuter, K. Ab initio thermodynamics Insight into the structural evolution of working IrO_2 catalysts in proton-exchange membrane electrolyzers. *ACS Catal.* **9**, 4944–4950 (2019).

Ref. R5. Rossmeisl, J., Qu, Z.-W., Zhu, H., Kroes, G.-J. & Nørskov, J. K. Electrolysis of water on oxide surfaces. *J. Electroanal. Chem.* **607**, 83–89 (2007).

Ref. R6. Ping, Y., Nielsen, R. J. & Goddard, W. A. The reaction mechanism with free energy barriers at constant potentials for the oxygen evolution reaction at the $\text{IrO}_2(110)$ surface. *J. Am. Chem. Soc.* **139**, 149–155 (2017).

Ref. R7. Nong, H. N. et al. Key role of chemistry versus bias in electrocatalytic oxygen evolution. *Nature* **587**, 408–413 (2020).

Ref. R8. Exner, K. S. & Over, H. Beyond the rate-determining step in the oxygen evolution reaction over a single-crystalline IrO₂(110) model electrode: kinetic scaling relations. *ACS Catal.* **9**, 6755–6765 (2019).

Ref. R9. Ha, M.-A. & Larsen, R. E. Multiple Reaction Pathways for the oxygen evolution reaction may contribute to IrO₂(110)'s high activity. *J. Electrochem. Soc.* **168**, 024506 (2021).

Ref. R10. Binniger, T. & Doublet, M.-L. The Ir–OOOO–Ir transition state and the mechanism of the oxygen evolution reaction on IrO₂(110). *Energy Environ. Sci.* **15**, 2519–2528 (2022).

Ref. R11. Geppert, J. et al. Microkinetic analysis of the oxygen evolution performance at different stages of iridium oxide degradation. *J. Am. Chem. Soc.* **144**, 13205–13217 (2022).

Ref. R12. Over, H. Fundamental studies of planar single-crystalline oxide model electrodes (RuO₂, IrO₂) for acidic water splitting. *ACS Catal.* **11**, 8848–8871 (2021).

Ref. R13. Weber, T. et al. Potential-induced pitting corrosion of an IrO₂(110)-RuO₂(110)/Ru(0001) model electrode under oxygen evolution reaction conditions. *ACS Catal.* **9**, 6530–6539 (2019).

Reviewer #4

This manuscript from the groups of Exner and Hättig reports on a computational-theoretical study of the oxygen evolution reaction (OER) on IrO₂. The authors present a comparison of different OER mechanisms based on the activity descriptor $G_{\text{max}}(\text{U})$, previously introduced by Exner and coworkers. In addition to previously reported OER mechanisms, the authors include a set of "Walden-type" mechanisms, where desorption of oxygen and adsorption of water at the Ir surface site are considered to occur simultaneously in one step. The authors conclude that the Walden-type mechanisms outperform the non-Walden equivalents in providing lowest $G_{\text{max}}(\text{U})$ and thus govern the OER activity on IrO₂(110).

Indeed, there exists some controversy with respect to the OER mechanism on the "archetype" IrO₂ catalyst and deciphering the respective mechanism is of great topical relevance. This manuscript deals with the central question whether the unsaturated state of the surface Ir cation after oxygen desorption can be avoided by simultaneous bond breaking and formation. The critical relevance of this question has recently been emphasised by Binniger et al. (Current Opinion in Electrochemistry 2023, 42:101382). Binniger et al. criticised the practise to consider the elementary steps of bond breaking during O₂ desorption and bond formation by H₂O (re)adsorption as only one "lumped" step, because this can hide the possibly unfavourable unsaturated state of the cation (Ir^{*}). The present study adopts an opposing view and suggests that both processes can indeed be treated as only one elementary step, similar to the Walden inversion, circumventing the unsaturated state of the Ir cation. This is certainly a very interesting perspective and a clear demonstration of such simultaneous bond breaking and formation would significantly contribute to the understanding of the OER mechanism on IrO₂(110), and possibly beyond. However, the present manuscript presents no such evidence, since only initial and final states of each step are computed. Therefore, the suggested Walden step essentially consists in omitting the intermediate state (Ir^{*}) from the sequence Ir-OOH → Ir^{*} → Ir-OH. Neglecting the energetically unfavourable intermediate point on the free-energy profile, it is not surprising that "Walden-type" mechanisms seem to provide a lower overall barrier (characterised by $G_{\text{max}}(\text{U})$). As such, the "Walden mechanisms" appear to correspond to a lumped version of conventional mechanisms. To show that they indeed represent a new pathway, the free-energy profile and intermediate/transition states along the Walden steps need to be computed. Ping et al. (Ref. 20, see Figure 7 therein) performed NEB calculations for such direct substitution of O₂ by H₂O on IrO₂(110). However, they did not observe simultaneous bond breaking and formation and concluded that O₂ desorption and H₂O adsorption rather proceed one after the other. The unsaturated Ir^{*} state still appeared as the "transition state" and resulted in a comparably large energy barrier. I therefore think that the central claims of the present manuscript are not sufficiently supported. If, however, in a revised work, the authors will show unambiguous evidence that the proposed Walden steps indeed avoid the unsaturated Ir^{*} (even at the transition state) and provide smaller barriers in comparison to other mechanisms, such demonstration would surely be of significant relevance.

Response: We thank the referee for their support of our manuscript for publication in Nature Communications and for the critical discussion. We agree that we hold a slightly opposite view compared to Binniger's article (Current Opinion in Electrochemistry 2023, 42:101382), but we believe that the present work closes a community gap between homogeneous and heterogeneous catalysis: while in homogeneous catalysis elementary steps associated with bond breaking and bond making, such as the concerted desorption of products (here O₂) and adsorption of reactants (here H₂O), are well accepted and considered in theoretical models, the opposite is often encountered for computational studies in heterogeneous catalysis, especially

in electrocatalysis. Due to the abundance of the reactant (H₂O) at the reactive interface, we are convinced that concerted desorption-adsorption steps may also be important for catalytic processes under applied bias. These processes have been largely overlooked in previous computational studies of electrocatalysis. To our knowledge, previous studies that have explicitly highlighted the relevance of concerted desorption-adsorption steps in electrocatalysis are the work of Exner, who used a generalized data-driven approach to capture Walden-like steps in the OER through volcano diagrams (**Ref. R1**), and the work of Liu and colleagues on the ORR over Fe-N-C catalysts (**Ref. R2**), which showed that the concerted desorption-adsorption of O₂ and H₂O represents the rate-limiting step. Therefore, we believe that our present article represents an emerging topic in the literature, since the general nature of Walden-like steps consisting of concerted desorption-adsorption processes has implications that extend far beyond the discussed model system of OER on IrO₂(110).

We understand the concern of the referee that in the initial submission of our manuscript, we did not discuss the kinetics of the Walden-like step, and the kinetic picture is needed to provide further evidence for the occurrence of concerted desorption-adsorption processes. Therefore, we provide such evidence in the revised version of our manuscript. Before we delve into the discussion of the kinetics (*vide infra*), we would like to comment on the reviewer’s concern that “Walden mechanisms appear to correspond to a lumped version of conventional mechanisms”. This statement was put forth by the reviewer based on the fact that “the suggested Walden step essentially consists in omitting the intermediate state (Ir*) from the sequence Ir-OOH → Ir* → Ir-OH”. We do not agree with this statement because the sequence Ir-OOH → Ir* → Ir-OH consists of a total of two electron transfers. If we omit the intermediate state (Ir*), this would imply that two electrons are transferred in a single reaction step, which is not the case in our model. We adhere to the formalism of the Marcus theory that each elementary step consists of a maximum of a single electron transfer or corresponds to a chemical reaction step without charge transfer. To this end, we do not think that Walden-like steps are a lumped version of conventional mechanisms; rather, we argue that the surface chemistry of the elementary steps of the OER is more diverse than initially assumed. Let us make a comparison to the formation of the *OOH adsorbate, which is often considered a bottleneck in the OER kinetics. The *OOH adsorbate is formed at the cus site by the splitting of a water molecule – *_{cus}-O_{ot} + H₂O → *_{cus}-OOH_{ot} + H⁺ + e⁻. On the other hand, a neighboring bridge site can also be involved in the process of *OOH formation – *_{cus}-O_{ot} + O_{br} + H₂O → *_{cus}-OOH_{ot} + OH_{br}. In short, several elementary steps can lead to the formation of *OOH and theoretical models should capture this diversity to identify the energetically preferred pathway. The same finding is observed with the process of O₂ desorption. This process can take place by leaving a vacant Ir* cation at the cus site – *_{cus}-OO_{ot} → *_{cus} + O_{2(g)}. On the other hand, a water-mediated route in the realm of a Walden-type step is also possible – *_{cus}-OO_{ot} + H₂O → *_{cus}-(OH₂)_{ot} + O_{2(g)}. The latter is often ignored in theoretical studies, but our work reports that OER pathways on IrO₂(110) following a concerted desorption-adsorption process are thermodynamically and kinetically favored over the conventionally considered O₂ desorption step, leaving behind a vacant Ir* cation.

Next, we comment on the statement of the referee that “Ping et al. (Ref. 20, see Figure 7 therein) performed NEB calculations for such direct substitution of O₂ by H₂O on IrO₂(110). However, they did not observe simultaneous bond breaking and formation and concluded that O₂ desorption and H₂O adsorption rather proceed one after the other.” We do not agree with this statement. Indeed, Ping et al. have calculated the transition state for the concerted replacement of O₂ by H₂O. We would like to emphasize that the scope of Ping’s study is a different one than reported in the present manuscript: Ping investigated two chemical steps by transition-

state calculations: a) $*_{\text{cus}}\text{-O}_{\text{ot}} + \text{O}_{\text{br}} + \text{H}_2\text{O} \rightarrow *_{\text{cus}}\text{-OOH}_{\text{ot}} + \text{OH}_{\text{br}}$; b) $*_{\text{cus}}\text{-OO}_{\text{ot}} + \text{H}_2\text{O} \rightarrow *_{\text{cus}}\text{-(OH}_2\text{)}_{\text{ot}} + \text{O}_{2(\text{g})}$. Based on the calculated free-energy barrier, Ping concluded that the formation of the $*\text{OOH}$ adsorbate and not of the Walden-like O_2 by H_2O substitution corresponds to the rate-determining step in the OER over $\text{IrO}_2(110)$. Importantly, Ping did not comment on the possibility of direct O_2 desorption without water mediation via $*_{\text{cus}}\text{-OO}_{\text{ot}} \rightarrow *_{\text{cus}} + \text{O}_{2(\text{g})}$. In other words, Ping only considered the Walden-type step – $*_{\text{cus}}\text{-OO}_{\text{ot}} + \text{H}_2\text{O} \rightarrow *_{\text{cus}}\text{-(OH}_2\text{)}_{\text{ot}} + \text{O}_{2(\text{g})}$ – but overlooked the ‘traditional’ step – $*_{\text{cus}}\text{-OO}_{\text{ot}} \rightarrow *_{\text{cus}} + \text{O}_{2(\text{g})}$ – in his model. In the present work, we close this gap by assessing the thermodynamics and kinetics for both elementary steps for O_2 desorption using $\text{IrO}_2(110)$.

Last but not least, we also comment on the sentence “The unsaturated Ir^* state still appeared as the “transition state” and resulted in a comparably large energy barrier”. To avoid misunderstandings, we do not claim that the unsaturated Ir^* state is no longer observed in the entire free-energy landscape of the OER over $\text{IrO}_2(110)$ when kinetics are included into the picture. This statement only applies to the oxide path proposed by Binniger (**Ref. R3**), since the electrode surface is completely covered with oxygen adsorbates during the elementary steps of the OER in the oxide pathway. During the concerted desorption-adsorption step, a transition state is formed in which both O_2 and H_2O loosely interact with the Ir_{cus} site. One can debate whether this transition state corresponds to the state of a metal cation or a coordinated Ir_{cus} site. However, this is not the focus of this manuscript, where we put emphasis on the reaction intermediates: when inspecting the reaction intermediates for the energetically preferred pathway of the OER over $\text{IrO}_2(110)$, we observe that, independent of surface configuration and adsorbate coverage, Walden-type pathways are energetically preferred over traditional pathways. This observation agrees with the finding of Binniger that in the oxide pathway – also denoted as tetraoxidane mechanism (**Ref. R4**) – the $\text{IrO}_2(110)$ surface is always capped by an adsorbate. This could also provide a further explanation for the high stability of $\text{IrO}_2(110)$ under OER conditions, in agreement with experimental studies (**Ref. R5-R6**).

A few additional comments and questions:

- The comparison of different mechanisms is not fully consistent in my opinion, because certain identical intermediates are included in some mechanisms, while neglected in others. As an example, I refer to the comparison between the mononuclear standard (Fig. S13) and Walden (Fig. S26) pathways. Steps RI-1 \rightarrow RI-2 \rightarrow RI-3 of the Walden mechanism (Fig. S26) are identical to steps RI-2 \rightarrow RI-3 \rightarrow RI-4 of the standard mechanism (Fig. S13), resulting in the formation of $*\text{OOH}$. The subsequent step in the Walden pathway consists in the deprotonation of $*\text{OOH}$, producing an $*\text{OO}$ adsorbate (RI-4 in Fig. S26). In contrast, for the standard mechanism (Fig. S13), this deprotonation step is lumped together with the subsequent desorption of O_2 and the $*\text{OO}$ adsorbate is entirely neglected. For consistency, the $*\text{OO}$ adsorbate should be explicitly included in the standard mechanism.

Response: We thank the reviewer for pointing out that the $*\text{OO}$ adsorbate was not considered in the traditional mononuclear mechanism. To address this point, we reformulate the mononuclear mechanism by including the $*\text{OO}$ adsorbate as well as the chemical desorption of O_2 as an additional step into the analysis:

The corresponding free-energy diagram at $U = 1.53$ V vs. RHE for equations (R1) – (R5) is shown in **Figure R1a**, using the partly hydroxylated $\text{IrO}_2(110)$ surface (cf. **Figure 1c** in the main text) as a representative example. In addition, we plot the free-energy diagram for the traditional mononuclear mechanism (where steps R4 and R5 are combined) in **Figure R1b**.

Figure R1. a) Free-energy diagram of the OER over the partly hydroxylated $\text{IrO}_2(110)$ surface according to the mechanistic description in equations (R1) – (R5) at $U = 1.53$ V vs. RHE. **b)** Free-energy diagram of the OER over the partly hydroxylated $\text{IrO}_2(110)$ surface for the traditional mononuclear mechanism at $U = 1.53$ V vs. RHE

To compare the energetics of the refined mononuclear mechanism containing a chemical step (cf. equation (R5)) with the energetics of the Walden-type pathway, we refine the mechanistic description of the mononuclear-Walden mechanism by considering a chemical step for the desorption of O_2 (cf. equation (R9)). Therefore, we consider the $*_{\text{cus}}\text{-(OH}_2\text{)}_{\text{ot}}$ intermediate as an additional adsorbate in our analysis:

The corresponding free-energy diagram at $U = 1.53$ V vs. RHE for equations (R6) – (R10) is shown in **Figure R2a**, using the partly hydroxylated $\text{IrO}_2(110)$ surface (cf. **Figure 1c** in the main text) as a representative example. In addition, we plot the free-energy diagram for the original description of the mononuclear-Walden pathway (where steps R9 and R10 are combined) in **Figure R2b**.

Figure R2. **a)** Free-energy diagram of the OER over the partly hydroxylated $\text{IrO}_2(110)$ surface according to the mechanistic description in equations (R6) – (R10) at $U = 1.53$ V vs. RHE. **b)** Free-energy diagram of the OER over the partly hydroxylated $\text{IrO}_2(110)$ surface for the mononuclear-Walden mechanism at $U = 1.53$ V vs. RHE.

When comparing the thermodynamic free-energy landscapes of **Figure R1-R2**, we observe that the electrocatalytic activity for the Walden-type pathway is not affected by the consideration of the $*_{\text{cus}}\text{-(OH}_2\text{)}_{\text{ot}}$ intermediate in the analysis: the descriptor $G_{\text{max}}(U)$ amounts to 0.22 eV, and the limiting free-energy span is $*_{\text{cus}}\text{-OH}_{\text{ot}} \rightarrow *_{\text{cus}}\text{-O}_{\text{ot}}$ at $U = 1.53$ V vs. RHE. In contrast, the consideration of an additional chemical step by means of the $*\text{OO}$ adsorbate

for the mononuclear mechanism leads to a significant increase in the descriptor $G_{\max}(U)$ from 0.75 eV to 1.06 eV, indicating that the Walden pathway is clearly favored over the traditional mechanism when using the span model as a measure for the electrocatalytic activity.

In the following, we provide evidence by the calculation of transition states that the water-mediated desorption of O_2 is kinetically more facile than the desorption of O_2 without the direct adsorption of water. To this end, we performed nudge elastic band (NEB) calculations to map the reaction pathway and locate an approximate transition state for equations (R5) and (R9). Subsequently, the dimer method was employed to refine and identify the precise transition state structure. The identified transition state was validated by frequency analysis. All these calculations were performed including spin polarization and an implicit solvation model (VASPsol).

To calculate the transition state of equations (R5) and (R9), we rely on a canonical framework to ensure consistency with the thermodynamic analysis of the reaction intermediates' free energies. This is further backed up by the fact that Ping and Goddard reported that the difference between constant potential and constant charge calculations for $IrO_2(110)$ in their study did not exceed 0.10 eV (Ref. R7), which is considered to be sufficiently accurate for the present qualitative assessment of the Walden transition state. While **Figure R3** illustrates the initial, transition, and final states for the process of equation (R5), **Table R1** summarizes the energetics of the states involved.

Figure R3. Initial (IS), transition (TS), and final (FS) states for O_2 desorption (cf. equation (R5)) from the partly hydroxylated $IrO_2(110)$ surface. All bond lengths are given in Å.

Table R1. Energetics of the initial (IS), transition (TS), and final (FS) states of O_2 desorption (cf. equation (R5)) from the partly hydroxylated $IrO_2(110)$ surface to derive the activation free energy, ΔG^\ddagger , for this process.

	E_{DFT} [eV]	ZPE [eV]	TS [eV]
IS	-434.66	0.52	0.20
TS	-433.38	0.47	0.31
FS	-433.34	0.48	0.24
G_{IS} [eV]		-434.34	
G_{TS} [eV]		-433.22	
ΔG^\ddagger [eV]		1.12	

In the same fashion, we investigate the transition state for the water-mediated Walden-type step. While **Figure R4** illustrates the initial, transition, and final states for the process of equation (R9), **Table R2** summarizes the energetics of the states involved.

Figure R4. Initial (IS), transition (TS), and final (FS) states for the water-mediated O₂ desorption (cf. equation (R9)) from the partly hydroxylated IrO₂(110) surface. All bond lengths are given in Å.

Table R2. Energetics of the initial (IS), transition (TS), and final (FS) states of water-mediated O₂ desorption (cf. equation (R9)) from the partly hydroxylated IrO₂(110) surface to derive the activation free energy, ΔG^\ddagger , for this process.

	E_{DFT} [eV]	ZPE [eV]	TS [eV]
IS	-449.43	1.16	0.32
TS	-448.31	1.15	0.38
FS	-449.47	1.14	0.31
G_{IS} [eV]	-448.59		
G_{TS} [eV]	-447.54		
ΔG^\ddagger [eV]	1.05		

Our analysis reveals that the desorption barrier of O₂ is on the order of about 1 eV, and the activation barrier for the water-assisted route in the framework of a Walden step (cf. equation (R9)) is 0.07 eV lower in free energy than the desorption of O₂ without the involvement of a water molecule (cf. equation (R5)). These activation free energies are incorporated into the free-energy landscapes of the OER on IrO₂(110), which are shown in **Figure R5**.

The comparative analysis of the free-energy landscape in **Figure R5** reveals that the Walden-type pathway is preferred over the traditional pathway due to the consideration of thermodynamic and kinetic factors. The descriptor $G_{\text{max}}(U)$ suggests that the kinetic bottleneck in the Walden-type pathway is related to elementary steps involving charge transfer, whereas the water-assisted desorption-adsorption step is not thermodynamically activated; this could be an indication that the Walden-type step of equation (R9) is not related to the rate-determining step (RDS). This finding is further supported by experiments at the single-crystal level: Suntivich and coworkers (**Ref. R8**) reported Tafel slopes of 49 mV/dec. and 78 mV/dec. in acid, highlighting that the reaction rate is determined by an electrochemical rather than a chemical step (**Ref. R9**). Therefore, we do not propose the water-mediated O₂ desorption as the RDS in the OER over IrO₂(110). We emphasize that an unbiased and clear-cut determination of the RDS would require the assessment of all possible transition states for all elementary steps in the catalytic cycle of the Walden pathway (cf. equations (R5) – (R10)). In addition, it is possible that not just a single elementary step, but several elementary steps contribute to the reaction rate, which can only be described by advanced analytical techniques, such as exploiting the degree of rate control (**Ref. R10**). However, such an analysis goes far beyond the scope of the present manuscript, which argues – based on thermodynamic and kinetic analyses – that Walden-type mechanisms are energetically preferable to traditional mechanistic descriptions in the OER over IrO₂(110).

Figure R5. **a)** Free-energy diagram of the OER over the partly hydroxylated IrO₂(110) surface according to the mechanistic description in equations (R1) – (R5) at $U = 1.53$ V vs. RHE. The calculated activation free energy for the desorption of the O₂ molecule (cf. **Table R1**) has been added to the figure. **b)** Free-energy diagram of the OER over the partly hydroxylated IrO₂(110) surface according to the mechanistic description in equations (R6) – (R10) at $U = 1.53$ V vs. RHE. The calculated activation free energy for the water-assisted desorption of the O₂ molecule (cf. **Table R2**) has been added to the figure.

Changes:

i) We have added a discussion on the kinetics of O₂ desorption to the main text of our manuscript. For the sake of brevity, we do not copy these changes here, and we refer the reviewer to pages 15-17.

ii) We have added the above discussion on the kinetics to a new section of the SI: section 15 “Kinetics of O₂ desorption”. Changes are made on pages S45-S48 of the SI.

- What is the difference between the proposed Walden-type steps in this study compared to other studies that considered O₂ desorption and H₂O adsorption as a single step, see, e.g., Scheme 1 a),b) in González et al., Journal of Catalysis 396 (2021) 192–201?

Response: We thank the reviewer for pointing out the article by González et al. in Journal of Catalysis, which was previously unknown to us. We agree with the referee that the proposed Walden-type step in our work is essentially the same as that studied by González in the previous article after the refined discussion based on equations (R6) – (R10). The main difference of our work to the article by González is that González – similar to Ping (**Ref. R7**) – only included a single option for the desorption of O₂ into their model – $*_{\text{cus}}\text{-OO}_{\text{ot}} + \text{H}_2\text{O} \rightarrow *_{\text{cus}}\text{-(OH}_2\text{)}_{\text{ot}} + \text{O}_{2(\text{g})}$ – but overlooked the ‘traditional’ O₂ desorption step – $*_{\text{cus}}\text{-OO}_{\text{ot}} \rightarrow *_{\text{cus}} + \text{O}_{2(\text{g})}$ – in their analysis. In the present work, we close this gap by assessing the thermodynamics and kinetics for both elementary steps for O₂ desorption using IrO₂(110). It is also relevant to mention that González investigated the thermodynamics of the OER over IrO₂(110) for two selected mechanisms and a single surface configuration of the (110) facet, whereas we take seven reaction mechanisms for four different surface configurations into account. This allows us to obtain reasonable statistics for the claim that, regardless of surface configuration, Walden-type mechanisms are energetically preferable to traditional mechanistic descriptions in the OER over IrO₂(110).

Change: We have cited and discussed the article by González et al. on page 19 in the revised version of our manuscript. There, we have added the following discussion:

*While we emphasize the importance of Walden-type steps and mechanisms for the theoretical description of electrocatalytic processes, we note that a few previous works have already investigated concerted desorption-adsorption steps for the OER over IrO₂(110). In the works by González et al. and Ping et al., the authors only included a single option for the desorption of O₂ into their theoretical model – $*_{\text{cus}}\text{-OO}_{\text{ot}} + \text{H}_2\text{O} \rightarrow *_{\text{cus}}\text{-(OH}_2\text{)}_{\text{ot}} + \text{O}_{2(\text{g})}$ (cf. **Figure 5b**) – but overlooked the conventional O₂ desorption step – $*_{\text{cus}}\text{-OO}_{\text{ot}} \rightarrow *_{\text{cus}} + \text{O}_{2(\text{g})}$ (cf. **Figure 5a**) – in the analysis. In the present work, we close this gap by investigating the thermodynamics and kinetics of both elementary steps of O₂ desorption on the IrO₂(110) surface. This allows us to obtain reasonable statistics for the claim that, regardless of surface configuration, Walden-type mechanisms are energetically preferable to traditional mechanistic descriptions in the OER over IrO₂(110).*

- The terminology of "Walden-type" mechanisms appears to be borrowed from the Walden inversion reaction, where one bond at a center gets broken due to formation of a new bond from the back side. I wonder whether such an inversion can happen at a surface Ir cation of IrO₂(110). Is this terminology thus appropriate?

Response: We thank the reviewer for critically reviewing the terminology of “Walden-type” mechanisms. We note that the ‘traditional Walden inversion’, which is particularly observed in homogeneous catalysis (**Ref. R11-R13**), takes place at an angle of 180°: to minimize steric hindrance, the reactant enters the active site while the product leaves the active site exactly on the opposite side. In heterogeneous catalysis, it is definitely not possible for the reactant to enter the active center and the product to leave the active center at an angle of 180°; rather, the angle between reactant and product is compressed, although our transition-state calculations (cf. **Figure R4**) clearly indicate that desorption and adsorption are concerted. This is actually a difference between Walden steps in homogeneous and heterogeneous catalysis, although we find that the chemical processes in terms of concerted desorption-adsorption still remain the same. Therefore, in this work, we adopt the terminology of “Walden-like” mechanisms because

we believe it will help bridge the knowledge gap between homogeneous and heterogeneous catalysis.

Change: We have provided an explanation of the terminology of “Walden-type” mechanisms on page 13 in the revised version of our manuscript:

A brief comment on the terminology of Walden pathways is needed. We note that the ‘traditional Walden inversion’, which is particularly observed in homogeneous catalysis, takes place at an angle of 180°: to minimize steric hindrance, the reactant enters the active site while the product leaves the active site exactly on the opposite side. In heterogeneous catalysis, it is definitely not possible for the reactant to enter the active center and the product to leave the active center at an angle of 180°; rather, the angle between reactant and product is compressed. Although this is a difference between ‘Walden steps’ in homogeneous and heterogeneous catalysis, the chemical processes in terms of concerted desorption-adsorption still remain the same. Therefore, in this work, we adopt the terminology of “Walden-like” mechanisms because we believe it will help bridge the knowledge gap between homogeneous and heterogeneous catalysis.

- The authors state that they performed grand-canonical DFT (GC-DFT) calculations under constant potential using VASPsol. As far as I know, VASPsol does not offer the GC-DFT method. I would be interested in better understanding how the authors performed the GC-DFT method.

Response:

The grand-canonical DFT (GC-DFT) calculations at constant potential were conducted using VASP version 6.4.1 in combination with the VASPsol implicit solvent model. While the version of VASPsol used in our work does not explicitly offer a built-in GC-DFT feature, this functionality is now available in the newer VASPsol++ implementation (**Ref. R14**). In our calculations, constant-potential conditions were achieved by calibrating the electron chemical potential relative to the standard hydrogen electrode (SHE) (**Ref. R15**).

In this approach, the electrochemical potential of the electron in the system is kept constant, while the charge of the system can vary. The absolute potential is determined as the difference between the Fermi level and the vacuum level, where this value is related to the calibrated electron chemical potential of -4.66 eV, which corresponds to 0 V vs. SHE. The calculation begins with the definition of a target potential. The number of electrons in the system is iteratively adjusted to ensure that the Fermi energy matches the specified potential. The applied electrode potential is then expressed as:

$$U = -(\text{shift} + \text{Fermi energy} + 4.66 \text{ eV}) \quad (\text{R11})$$

where the shift (representing the electrostatic potential) and the Fermi energy are extracted directly from the VASPsol outputs.

Furthermore, in the grand-canonical formalism, the thermodynamic potential of interest is the grand free energy (Ω), as opposed to the standard (Gibbs) free energy. The grand free energy is calculated using the expression:

$$\Omega = E_{\text{tot}} - (N_q - N_0) \times \text{Fermi energy} \quad (\text{R12})$$

where N_q is the number of electrons at the target potential, and N_0 is the number of electrons in the neutral system.

It is important to note that the vacuum level is not explicitly included in VASPsol. To account for this, the chemical potential of the electron at 0 V vs. SHE is calibrated based on the experimental potential of zero charge (PZC) for specific transition-metal electrodes. For further information, we relate the reader to the corresponding VASPsol publication (**Ref. R15**).

Change: We have added further explanation to **section 1** of the supporting information. Changes have been made on **page S2** of the SI.

Ref. R1. Exner, K. S. Importance of the walden inversion for the activity volcano plot of oxygen evolution. *Adv. Sci.* **10**, 2305505 (2023).

Ref. R2. Yu, S., Levell, Z., Jiang, Z., Zhao, X. & Liu, Y. What is the rate-limiting step of oxygen reduction reaction on Fe–N–C catalysts? *J. Am. Chem. Soc.* **145**, 25352–25356 (2023).

Ref. R3. Binninger, T. & Doublet, M.-L. The Ir–OOOO–Ir transition state and the mechanism of the oxygen evolution reaction on IrO₂(110). *Energy Environ. Sci.* **15**, 2519–2528 (2022).

Ref. R4. Binninger, T., Moss, G. C., Rajan, Z. S. H. S., Mohamed, R. & Eikerling, M. H. How to break the activity-stability conundrum in oxygen evolution electrocatalysis: mechanistic insights. *ChemCatChem* **16**, e202400567 (2024).

Ref. R5. Weber, T. et al. Potential-induced pitting corrosion of an IrO₂(110)-RuO₂(110)/Ru(0001) model electrode under oxygen evolution reaction conditions. *ACS Catal.* **9**, 6530–6539 (2019).

Ref. R6. Over, H. Fundamental studies of planar single-crystalline oxide model electrodes (RuO₂, IrO₂) for acidic water splitting. *ACS Catal.* **11**, 8848–8871 (2021).

Ref. R7. Ping, Y., Nielsen, R. J. & Goddard, W. A. The reaction mechanism with free energy barriers at constant potentials for the oxygen evolution reaction at the IrO₂ (110) surface. *J. Am. Chem. Soc.* **139**, 149–155 (2017).

Ref. R8. Kuo, D.-Y. et al. Measurements of oxygen electroadsorption energies and oxygen evolution reaction on RuO₂(110): A discussion of the sabatier principle and its role in electrocatalysis. *J. Am. Chem. Soc.* **140**, 17597–17605 (2018).

Ref. R9. Exner, K. S. & Over, H. Beyond the rate-determining step in the oxygen evolution reaction over a single-crystalline IrO₂ (110) model electrode: kinetic scaling relations. *ACS Catal.* **9**, 6755–6765 (2019).

Ref. R10. Exner, K. S. Standard-state entropies and their impact on the potential-dependent apparent activation energy in electrocatalysis. *J. Energy Chem.* **83**, 247–254 (2023).

Ref. R11. Tong, L. & Thummel, R. P. Mononuclear ruthenium polypyridine complexes that catalyze water oxidation. *Chem. Sci.* **7**, 6591–6603 (2016).

Ref. R12. Wang, L.-P., Wu, Q. & Van Voorhis, T. Acid–base mechanism for ruthenium water oxidation catalysts. *Inorg. Chem.* **49**, 4543–4553 (2010).

Ref. R13. Amthor, S. et al. Strong ligand stabilization based on π -extension in a series of ruthenium terpyridine water oxidation catalysts. *Chem. Eur. J.* **27**, 16871–16878 (2021).

Ref. R14. Islam, S. M., Khezeli, F., Ringe, S. & Plaisance, C. An implicit electrolyte model for plane wave density functional theory exhibiting nonlinear response and a nonlocal cavity definition. *J. Chem. Phys.* **159**, 234117 (2023).

Ref. R15. Mathew, K. et al. Implicit self-consistent electrolyte model in plane-wave density-functional theory. *J. Chem. Phy.* **151**, 234101 (2019).

Response:

We would like to express our sincere thanks to the four reviewers for their time and efforts in reviewing our manuscript. Their insightful comments and constructive criticisms have been carefully considered in this revised version of the article. The responses to the reviewers are organized as follows:

- | | |
|------------------------|--------|
| a) Reply to referee #1 | page 2 |
| b) Reply to referee #2 | page 3 |
| c) Reply to referee #3 | page 4 |
| d) Reply to referee #4 | page 5 |

We sincerely hope that with these revisions, our manuscript meets the high standards for publication in *Nature Communications*.

Sincerely,
Kai S. Exner (on behalf of all authors)

Reviewer #1

The authors present a resubmission of a previous manuscript. Already the original manuscript on their simulation work on the OER or IrOx materials was insightful, novel and contained an extensive study on the topic. Following the suggestions of four referees, the authors have considerably extended their data, added further simulations and results, improved on their citations and references and thoroughly worked on the clarity and presentation of their arguments. Especially the inclusion of further illustrations and data and the microkinetic modelling that was performed further improve the quality of the results.

I would now consider this a very clear and conclusive work that contains many interesting aspects that theory can provide for novel catalysts for water electrolysis. Given the relevance of the subject and the refined methods applied by the authors, I would also consider the manuscript of broader interest and perfectly suited for Nature Communications.

I recommend publication without further changes.

Response: We thank the referee for their strong support of our manuscript for publication in Nature Communications. No further changes were made on the recommendation of this referee.

Reviewer #2

-

Response: We thank the referee for their strong support of our manuscript for publication in Nature Communications. No further changes were made on the recommendation of this referee.

Reviewer #3

The manuscript has been thoroughly revised, and all my comments have been adequately addressed. I therefore recommend it for publication without further comments.

Response: We thank the referee for their strong support of our manuscript for publication in Nature Communications. No further changes were made on the recommendation of this referee.

Reviewer #4

The new results added during revision disprove the asserted role of "Walden-type" mechanisms in the OER on IrO₂ (110). Hence, I do not consider this work to be suitable for publication.

Response: We accept the reviewer's critical discussion of our manuscript for publication in Nature Communications. Although the reviewer states that the new findings added during the revision refute the role of "Walden-type" mechanisms, we take a different view on this topic, which we explain in detail below.

As suggested, the authors have performed calculations of activation barriers of both the conventional O₂ desorption step and the concerted O₂ desorption-H₂O adsorption step (termed "Walden-type" by the authors). A barrier of 1.12 eV was obtained for the former, while the barrier of the "Walden-type" step was 1.05 eV. The difference of 0.07 eV is insignificant and shows that both steps are governed by the energy cost for O₂ desorption alone, while the "assistance" by water is not significant.

Response: We agree with the reviewer that the assistance of O₂ desorption by water does not cause a significantly smaller free-energy barrier, as the difference of 0.07 eV is within the error bars of the DFT approach. However, the referee overlooked the fact that the water-mediated pathway significantly alters the thermodynamic free-energy landscape of the OER on IrO₂(110). This is the main point we discuss in our manuscript and which, in our opinion, represents a highly relevant result for the scientific discipline of theoretical electrocatalysis.

Figure R1. a) Thermodynamic free-energy diagram of the OER over the partly hydroxylated IrO₂(110) surface for the traditional mononuclear mechanism with O₂ desorption at $U = 1.53$ V vs. RHE. b) Thermodynamic free-energy diagram of the OER over the partly hydroxylated IrO₂(110) surface for the Walden-type mechanism with water-assisted O₂ desorption at $U = 1.53$ V vs. RHE.

As shown in **Figure R1**, the thermodynamics of the OER mechanism changes by more than 0.80 eV when the Walden mechanism with water-assisted O₂ desorption is considered in the modeling. This is a significant effect that cannot be overlooked and clearly shows that the Walden-type description is thermodynamically preferred over conventional reaction mechanisms. These findings and their implications are the core message of our contribution to Nature Communications.

Please note that it is not our aim to discuss the kinetics in terms of transition states of the OER on IrO₂(110). If one wanted to go down this route, one would have to calculate all transition states of all mechanistic pathways, rather than just a few selected one. It is evident that the calculation of all transition states of all reaction mechanisms is far beyond the scope of this manuscript. However, we use the transition state for the kinetics of O₂ desorption to pinpoint that the Walden-type description cannot be kinetically excluded, since the transition

state for water-mediated O₂ desorption is kinetically slightly favored over the conventional O₂ desorption step involving a solvent molecule.

Given total barrier heights beyond 1 eV, none of these steps can be expected to proceed at room temperature at any relevant rate.

Response: There are two points to note about this statement.

First, conventional electrolyzers operate at 80 – 90 °C, and at these temperatures, free-energy barriers of chemical steps above 1 eV are unproblematic.

Second, we are not interested in a quantitative discussion of the free-energy barriers. The error bars of the DFT approach to quantify transition states in an electrochemical environment are significant and therefore do not allow a quantitative discussion of the transition states. A different situation arises, however, with the thermodynamics of the elementary steps along the reaction coordinate, since this information can be resolved with a much higher accuracy.

To prove our point, we have extracted the free-energy barrier for the water-mediated O₂ desorption from the study of Ping et al. (**Ref. R1**). The corresponding graphs are shown in **Figure R2**.

Figure R2. Free-energy diagram of the OER over the partly hydroxylated IrO₂(110) surface containing water-assisted O₂ desorption at $U = 1.53$ V vs. RHE from the study of Ping et al. (**Ref. R1**). The Walden-type step is given by $4 \rightarrow \text{TS2} \rightarrow 5$. Ping et al. states that “a transition state barrier of 0.39 eV (assuming that the translational and rotational enthalpy and entropy of O₂ is from the gas phase) or 0.56 eV (if we consider the translational and rotational enthalpy and entropy of O₂ is from 1M O₂ in liquid H₂O)” is obtained at $U = 1.53$ V vs. RHE.

As shown in **Figure R2**, the transition state for the water-mediated O₂ desorption step is determined to be 0.39 eV or 0.56 eV, depending on the respective approximation. Such a barrier can be easily overcome even at room temperature. In addition, such a barrier is

comparable to the free-energy barrier determined by Binniger and Doublet for the oxide pathway (**Ref. R2**), which is mentioned by the referee in the below.

Since the transition states for the water-mediated O₂ desorption step in Ping's work (**Ref. R1**) and our current contribution differ by about 0.50 eV, we do not consider a quantitative discussion of the kinetics for this particular step to be justified for the reasons mentioned above.

Change: We have added a statement on this matter on **page 17** of the revised version of our manuscript.

Being "chemical" steps, the barriers are largely insensitive to electrode potential, which thus does not help.

Response: We agree with the referee that the free-energy barriers of chemical steps are largely insensitive to the applied electrode potential. For this reason, we have also claimed that the water-mediated O₂ desorption step is not the rate-determining step in the OER over IrO₂(110). This result is in agreement with the work by Ping et al. (**Ref. R1**), who, based on the comparison of two transition states at $U = 1.53$ V vs. RHE (cf. **Figure R2**), arrived at the conclusion that the formation of *OOH is kinetically limiting.

A side remark on all previous theoretical works is needed. The works by Ping et al. (**Ref. R1**) as well as Binniger and Doublet (**Ref. R2**) resolve the transition states of selected chemical reaction steps in the complex four proton-coupled electron transfer pathway of the OER.

None of these chemical steps can actually be rate determining when examining the experimental data on this topic. Previous work by Suntivich and coworkers on IrO₂(110) at the single-crystal level (**Ref. R3-R4**), which was further analyzed by Exner and Over (**Ref. R5**), provides evidence that electrochemical reaction steps are governing the reaction rate of the OER over IrO₂(110). Therefore, the calculation of transition states for chemical reaction steps is not a suitable method to identify the kinetic bottleneck in the OER, which again underlines that the thermodynamic picture in terms of the energetics of the intermediate states is necessary to draw conclusions about the OER activity for IrO₂(110).

Change: We note that this aspect was already discussed in our previous revision. Associated changes were made on **page 17** of the revised version of our manuscript. We have highlighted these changes in teal.

Finally, the structure of the transition state of the "Walden" step (Fig. S35) shows that O₂ desorption and H₂O adsorption, in fact, proceed one after the other, and there is no indication of a concerted bond breaking and formation. Seeing these results, the authors' conclusions appear incomprehensible to me.

Response: We disagree with the statement of the referee that "there is no indication of a concerted bond breaking and formation". We discussed the terminology of the Walden inversion in the first revision of our manuscript:

We note that the 'traditional Walden inversion', which is particularly observed in homogeneous catalysis, takes place at an angle of 180°: to minimize steric hindrance, the reactant enters the active site while the product leaves the active site exactly on the opposite side. In heterogeneous catalysis, it is definitely not possible for the reactant to enter the active center and the product to leave the active center at an angle of 180°; rather, the angle between reactant and product is compressed. Although this is a difference between 'Walden steps' in homogeneous and heterogeneous catalysis, the chemical processes in terms of concerted desorption-adsorption still remain the same. Therefore, in this work, we adopt the

terminology of “Walden-like” mechanisms because we believe it will help bridge the knowledge gap between homogeneous and heterogeneous catalysis.

As can be seen from the above, in the traditional Walden inversion in homogeneous catalysis, the reactant enters the active site from one side, while the product leaves the active site from the other side at an angle of 180° . Our transition state structure, which we repeat in **Figure R3**, clearly shows that the reactant (H_2O) and the product (O_2) enter or leave the active Ir_{cus} site from different positions, which is indeed reminiscent of a traditional Walden inversion, although the angle is smaller than 180° .

Figure R3. Initial (IS), transition (TS), and final (FS) states for the water-mediated O_2 desorption from the partly hydroxylated $\text{IrO}_2(110)$ surface. All bond lengths are given in Å.

From my point of view, these results show the following: First, O_2 desorption and H_2O adsorption proceed sequentially and, hence, there is no "Walden" step on $\text{IrO}_2(110)$, which is in agreement to previous findings by Ping et al.

Response: We disagree with this statement. If O_2 desorption and H_2O adsorption occur sequentially, this would lead to the free-energy landscape shown in **Figure R1a**, which is clearly disadvantaged compared to the Walden mechanism shown in **Figure R1b**.

In addition, Ping et al. (**Ref. R1**) has modeled the OER on $\text{IrO}_2(110)$ considering a Walden-type step, and we refer the reviewer to **Figure R2** from Ping's work. The result of Ping's work, namely that the Walden-type step (water-mediated O_2 desorption) is not rate determining, is consistent with our contribution and shows that there is no contradiction with previous work.

Second, neither the conventional O_2 desorption nor the water-assisted ("Walden") steps can proceed at relevant rates at ambient temperature on $\text{IrO}_2(110)$. Instead, the OER must proceed via the alternative pathway with a Ir-OOOO-Ir association step, for which we found a feasible barrier of about 0.34 eV in our previous work (Energy Environ. Sci., 2022, 15, 2519). In the present work, the authors did not consider the barrier of this step.

-Tobias Binninger

Response: We thank Dr. Binninger for pointing out his previous work with Doublet in Energy Environ. Sci. (**Ref. R2**). We cited this work in our originally submitted manuscript because we believe it represents an important contribution to the OER community.

To our opinion, Dr. Binninger and we take a different approach to identify the favored mechanistic description of the OER on $\text{IrO}_2(110)$.

Dr. Binninger refers to the free-energy barrier of chemical reaction steps in the OER cycle and aims to identify the smallest free-energy barrier in the reaction mechanism based on the modeling of selected transition states. We have several reservations about this approach:

i) According to the Butler-Volmer theory (**Ref. R6**), the rate of electrocatalytic processes is not determined by barriers but rather by transition states, combining the thermodynamic and kinetic contributions to the transition state with the highest free energy. Therefore, it is not our aim to discuss barriers in identifying the preferred mechanism or the limiting reaction step.

ii) To identify the transition state with the highest free energy, which determines the kinetic rate equation, the energetics of all transition states along the reaction coordinate must be calculated. We emphasize that no theoretical work has been able to tackle such a daunting task, as it requires the resolution of the transition states of all chemical and electrochemical steps in the OER catalytic cycle. Despite increasing computational power, this still exceeds the capabilities of theoretical studies by far due to the enormous computational effort required to identify transition states in an electrochemical environment. Therefore, we conclude that current approaches in theoretical electrocatalysis for barrier calculations are not sufficient to capture the ‘true kinetic picture’ of the proton-coupled electron transfer steps. To this end, we rely on thermodynamic approaches rather than barrier calculations to infer the preferred mechanistic pathway.

iii) We argue that the discussion of barriers for chemical reaction steps in the OER over IrO₂(110) does not seem to make sense to us if one wants to identify the preferred reaction mechanism. The reason for this is – as explained above – that chemical steps are not rate determining (**Ref. R3-R5**) and are therefore hidden in the kinetic rate equation, which is determined only by the transition state with the highest free energy (**Ref. R6**). Considering that the transition state with the highest free energy in the OER over IrO₂(110) is an electrochemical step, it is counterintuitive to focus on the chemical steps when discussing electrocatalytic activity. Note that the transition states of electrochemical steps in the OER over IrO₂(110) were not resolved in previous computational studies.

iv) Based on the above discussion, we draw conclusions in our manuscript using thermodynamic approaches by making use of the descriptor $G_{\max}(U)$ (**Ref. R7-R8**). This descriptor is a thermodynamic representation of the energetic model (**Ref. R9**) and approximates the transition state with the highest free energy by the free energy of the intermediate states in dependence of the applied electrode potential. The advantage of the $G_{\max}(U)$ descriptor is that, in contrast to barrier calculations, all free-energy spans are analyzed, which are then related to the kinetics via the BEP (Brønsted–Evans–Polanyi) relation. In this context, it was reported that the sensitivity of the $G_{\max}(U)$ descriptor due to the approximation of the transition state with the highest free energy is 0.20 eV (**Ref. R7**) – in other words, if the energetics of two mechanisms differ by at least 0.20 eV, it can be concluded that the mechanism with the smaller $G_{\max}(U)$ value prevails. Although this approach is described on page 6 of our manuscript, we repeat it here to emphasize why we rely on thermodynamic information rather than the energetics of selected transition states when discussing OER on IrO₂(110). Considering the caveats listed above and reservations regarding the discussion of barriers to chemical reaction steps, we believe that this approach is fully consistent – within the BEP approximation – and provides us with new insights into the OER over IrO₂(110).

The criticism of Dr. Binniger urges us to a detailed discussion of the oxide mechanism (which includes an Ir-OOOO-Ir association step) and the Walden-type mechanisms, which is carried out below.

Comparing the thermodynamic free-energy landscapes of the OER on IrO₂(110), **Figure R4** shows that the Walden-type pathway is clearly preferred over the oxide mechanism, as the $G_{\max}(U)$ value is 0.44 eV smaller. Considering the sensitivity of 0.20 eV of the $G_{\max}(U)$ descriptor (**Ref. R7**), it can be reasonably assumed that the Walden mechanism is superior to the oxide description.

Figure R4. a) Thermodynamic free-energy diagram of the OER over the partly hydroxylated IrO₂(110) surface for the Walden-type mechanism with water-assisted O₂ desorption at $U = 1.53 \text{ V}$ vs. RHE. b) Thermodynamic free-energy diagram of the OER over the fully oxygen-covered IrO₂(110) surface for the oxide mechanism with an Ir-OOOO-Ir association step at $U = 1.53 \text{ V}$ vs. RHE.

Figure R5 indicates the free-energy landscape of the OER for the Walden-type and oxide mechanisms considering the transition state for water-assisted O₂ desorption and the Ir-OOOO-Ir association step, respectively. When comparing the barrier heights of the transition states, it becomes clear that the Ir-OOOO-Ir association step is kinetically simpler.

Figure R5. a) Thermodynamic free-energy diagram of the OER over the partly hydroxylated IrO₂(110) surface for the Walden-type mechanism with water-assisted O₂ desorption at $U = 1.53$ V vs. RHE. The calculated activation free energy for the water-assisted desorption of the O₂ molecule has been added to the figure. b) Thermodynamic free-energy diagram of the OER over the fully oxygen-covered IrO₂(110) surface for the oxide mechanism with an Ir-OOOO-Ir association step at $U = 1.53$ V vs. RHE. The calculated activation free energy for the Ir-OOOO-Ir association step is taken from previous work by Binninger and Doublet (Ref. R2).

We argue that the actual barrier height of the water-assisted O₂ desorption or the Ir-OOOO-Ir association step is not relevant because none of these steps constitutes the rate-determining step in the OER over IrO₂(110) (Ref. R3-R5). Therefore, these steps are hidden in the kinetic rate equation, which is determined only by the transition state with the highest free energy (Ref. R6).

By comparing the $G_{\max}(U)$ descriptor, which is directly related to the transition state with the highest free energy, it becomes clear that the oxide mechanism is thermodynamically unfavorable due to the formation of two adjacent *OOH intermediates. Such a scenario is not observed for the Walden-type mechanism, which requires only a single *OOH intermediate. Therefore, we conclude that, despite the kinetically facile Ir-OOOO-Ir association step, the bottleneck of the oxide mechanism is due to the lateral interaction of the *OOH intermediates required for this pathway. For a further contemplation of the limiting factors, it would be required to calculate transition states for the formation of two adjacent *OOH intermediates. However, this is beyond the scope of the present manuscript, which focuses on the thermodynamic information related to the reaction intermediates by using the $G_{\max}(U)$ descriptor.

It should also be noted that, even if the barrier of 1.05 eV for water-assisted O₂ desorption appears large (cf. **Figure R5a**), this barrier can be overcome for industrial operating conditions of 80 – 90 °C. However, we do not believe that a qualitative discussion of the water-assisted O₂ desorption barrier is justified, considering that previous work on the topic reported a significantly smaller barrier (cf. **Figure R6**). Following the previous work by Ping et al. (**Ref. R1**), there are no kinetic limitations for water-assisted O₂ desorption (free-energy barrier of 0.56 eV), even at room temperature.

Figure R6. Free-energy diagram of the OER over the partly hydroxylated IrO₂(110) surface for the Walden-type mechanism with water-assisted O₂ desorption at $U = 1.53 \text{ V}$ vs. RHE. The calculated activation free energy for the water-assisted desorption of the O₂ molecule is taken from previous work by Ping et al. (**Ref. R1**).

To further investigate the correlation of the two reaction pathways, we have performed microkinetic simulations based on the evaluation of the descriptor $G_{\max}(U)$ (**Ref. R7-R8**) to estimate the current density (j) as a function of the applied electrode potential (U) for the oxide mechanism (cf. **Table R1**), which can be compared to the experimentally measured current density by Suntivich and coworkers for an IrO₂(110) model electrode (**Ref. R3-R4**).

In our analysis, we use an applied electrode potential of $U = 1.60$ V vs. RHE (reversible hydrogen electrode) to link experimental and theoretical investigations. Following previous work on the descriptor $G_{\max}(U)$ (Ref. R7-R8), this activity measure allows the determination of the current density using equation (1):

$$j(U) = \frac{4k_B T}{h} e \Gamma_{\text{act}} e^{\left[\frac{-(G_{\max}(U) + \beta)}{k_B T}\right]} \quad (1)$$

In equation (1), e , k_B , T , and h denote the elementary charge, Boltzmann constant, absolute temperature in Kelvin, and Planck constant, respectively, while the density of active surface site (cus sites) amounts to $\Gamma_{\text{act}} = 7 \times 10^{14} \text{ cm}^{-2}$ for $\text{IrO}_2(110)$. β refers to the Brønsted-Evans-Polanyi (BEP) intercept constant, which links the thermodynamic analysis in terms of $G_{\max}(U)$ with the kinetics related to the transition-state free energy. Two β values are selected for the analysis: a) $\beta = 0.60$ eV and b) $G_{\max}(U) + \beta = 0.34$ eV; the latter corresponding to the activation barrier reported for O_2 desorption by the *OO-OO* association step in Ref. R2.

Table R1. Current densities (j) for the oxide mechanism over the fully oxygen-covered $\text{IrO}_2(110)$ surface configuration at $U = 1.60$ V vs. RHE.

Mechanism	BEP intercept constant (β)	$U = 1.60$ V vs. RHE
		Current Density (j) [mA/cm ²]
Oxide	0.60	2.14×10^{-8}
	0.34	4.99×10^6

Given that the experimental benchmark for OER over $\text{IrO}_2(110)$, as reported by Kuo et al. (Ref. R3-R4), is $j \approx 0.01 \text{ mA/cm}^2$ at $U = 1.60$ V vs. RHE, it is evident that the current densities predicted for the oxide mechanism deviate significantly from this benchmark. On the contrary, the current densities derived for the Walden-type pathway approach the experimental current density for single-crystal $\text{IrO}_2(110)$ (cf. section 16 of the supplementary information), thus confirming our conclusion that Walden-like mechanisms are favored in the OER.

Finally, we performed a Bader charge analysis for the oxide mechanism, which is shown in Figure R7. As discussed in section 2.4 of our manuscript, we link the largest charge span in the catalytic cycle – Q_{\max} – to the electrocatalytic activity in terms of $G_{\max}(U)$. Our analysis reveals that Q_{\max} amounts to $+0.23e$ for the oxide description, which is larger than the charge span for the Walden-type pathway of $+0.17e$ (cf. Figure 6b of the main text). Therefore, this approach also confirms that the OER over $\text{IrO}_2(110)$ is controlled by Walden-like mechanisms.

In summary, we have provided a detailed comparison of the oxide mechanism containing the Ir-OOOO-Ir association step – as reported by Dr. Binniger in Ref. R2 – and the Walden-type pathways put forth in the present contribution. Based on the above reasoning, we remain committed that the overarching message of our manuscript – ‘Oxygen evolution reaction on $\text{IrO}_2(110)$ is governed by Walden-type mechanisms’ – is valid.

Figure R7. Charge states (upper panel) and free-energy diagram (lower panel) of the reaction intermediates in the OER over the fully oxygen-covered IrO₂(110) surface for the oxide mechanism at $U = 1.53 \text{ V vs. RHE}$.

Ultimately, however, we must acknowledge that all current works in the field of theoretical electrocatalysis are based on approximations, since not all transition states can be resolved in the free-energy landscape. It can be interpreted as a matter of taste whether the thermodynamics – as done in the present contribution to a large extent – or the kinetics of selected steps is used for a discussion of the electrocatalytic activity. We have pointed out in this reply that the discussion of the transition states for selected steps as an indicator for electrocatalytic activity is subject to bias because it assumes *a priori* which elementary step limits the reaction rate. A further discussion on the topic can be found in **Ref. R10**, where an entire section in the manuscript (section 3.5) is devoted to the question of using thermodynamic approaches in the realm of $G_{\text{max}}(U)$ or kinetic approaches by referring to selected transition states for activity analysis. In the end, one has to be aware that none of these approaches is entirely conclusive, since only knowledge of all transition states, coupled with a degree of rate control analysis (**Ref. R11**), would allow us to draw definite conclusions about the electrocatalytic activity, limiting steps, and reaction mechanisms of proton-coupled electron transfer steps at electrified solid/ liquid interfaces. We believe that the community will continue to evolve in this direction over the next years, provided the determination of transition states in an electrochemical environment is facilitated by machine learning and artificial intelligence approaches. Currently, we believe that discussing the OER over IrO₂(110) using the $G_{\text{max}}(U)$ approach is the best compromise for a consistent and unbiased evaluation of the electrocatalytic activity of a model system.

Changes:

- i) We have added a sentence on page 19 related to the charge span approach for the oxide mechanism.
- ii) We have added a sentence on page 21 related to the modeling of current densities for the oxide mechanism.
- iii) We have added a discussion of Dr. Binninger's previous work and the different approaches to electrocatalytic activity on page 20 of the revised version of our manuscript.
- iv) We have added the above discussion on the oxide and Walden-type pathways to a new section of the SI: section 17 "Comparison with a previous work by Binninger and Doublet". Changes are made on pages S49-S55 of the SI.

Ref. R1. Ping, Y., Nielsen, R. J. & Goddard, W. A. The reaction mechanism with free energy barriers at constant potentials for the oxygen evolution reaction at the IrO₂ (110) surface. *J. Am. Chem. Soc.* 139, 149–155 (2017).

Ref. R2. Binninger, T. & Doublet, M.-L. The Ir–OOOO–Ir transition state and the mechanism of the oxygen evolution reaction on IrO₂(110). *Energy Environ. Sci.* 15, 2519–2528 (2022).

Ref. R3. Kuo, D.-Y. et al. Influence of Surface Adsorption on the Oxygen Evolution Reaction on IrO₂(110). *J. Am. Chem. Soc.* 2017, 139, 3473– 3479.

Ref. R4. Kuo, D.-Y. et al. Measurements of oxygen electroadsorption energies and oxygen evolution reaction on RuO₂(110): A discussion of the sabatier principle and its role in electrocatalysis. *J. Am. Chem. Soc.* 140, 17597–17605 (2018).

Ref. R5. Exner, K. S. & Over, H. Beyond the rate-determining step in the oxygen evolution reaction over a single-crystalline IrO₂(110) model electrode: kinetic scaling relations. *ACS Catal.* 9, 6755–6765 (2019).

Ref. R6. Bockris, J. O'M. Reddy, A. K. N. *Modern Electrochemistry*, Vol.2, Plenum Publishing Corporation, New York (1973).

Ref. R7. Exner, K. S. A universal descriptor for the screening of electrode materials for multiple-electron processes: Beyond the thermodynamic overpotential. *ACS Catal.* 10, 12607–12617 (2020).

Ref. R8. Razzaq, S. & Exner, K. S. Materials screening by the descriptor $G_{\max}(\eta)$: The free-energy span model in electrocatalysis. *ACS Catal.* 13, 1740–1758 (2023).

Ref. R9. Kozuch, S. & Shaik, S. How to Conceptualize Catalytic Cycles? The Energetic Span Model. *Acc. Chem. Res.* 44, 101–110 (2011).

Ref. R10. Exner, K. S. Standard-state entropies and their impact on the potential-dependent apparent activation energy in electrocatalysis. *J. Energy Chem.* **83**, 247–254 (2023).

Ref. R11. Dhaka, K. & Exner, K. S. Degree of span control to determine the impact of different mechanisms and limiting steps: Oxygen evolution reaction over $\text{Co}_3\text{O}_4(001)$ as a case study. *J. Catal.* 443, 115970 (2025).